palaeontology

Dinosauria, Ornithischia, Thyreophora, Kayenta Formation, Jurassic, *Scutellosaurus lawleri*

**Author for correspondence:**
Benjamin T. Breeden III
e-mail: b.breeden@utah.edu

# The anatomy and palaeobiology of the early armoured dinosaur *Scutellosaurus lawleri* (Ornithischia: Thyreophora) from the Kayenta Formation (Lower Jurassic) of Arizona

Benjamin T. Breeden III[1,2,3], Thomas J. Raven[4,5], Richard J. Butler[6], Timothy B. Rowe[3] and Susannah C. R. Maidment[4]

[1]Department of Geology and Geophysics, University of Utah, Salt Lake City, UT, USA
[2]Natural History Museum of Utah, Salt Lake City, UT, USA
[3]Jackson School of Geosciences, The University of Texas at Austin, Austin, TX, USA
[4]Department of Earth Sciences, The Natural History Museum, Cromwell Road, London SW7 5BD, UK
[5]School of Environment and Technology, University of Brighton, Lewes Road, Brighton BN1 4GJ, UK
[6]School of Geography, Earth and Environmental Sciences, University of Birmingham, Edgbaston, Birmingham B15 2TT, UK

BTB, 0000-0001-7352-0013; TJR, 0000-0002-4349-5635;
RJB, 0000-0003-2136-7541; SCRM, 0000-0002-7741-2500

The armoured dinosaurs, Thyreophora, were a diverse clade of ornithischians known from the Early Jurassic to the end of the Cretaceous. During the Middle and Late Jurassic, the thyreophorans radiated to evolve large body size, quadrupedality, and complex chewing mechanisms, and members of the group include some of the most iconic dinosaurs, including the plated *Stegosaurus* and the club-tailed *Ankylosaurus*; however, the early stages of thyreophoran evolution are poorly understood due to a paucity of relatively complete remains from early diverging thyreophoran taxa. *Scutellosaurus lawleri* is generally reconstructed as the earliest-diverging thyreophoran and is known from over 70 specimens from the Lower Jurassic Kayenta Formation of Arizona, USA. Whereas *Scutellosaurus lawleri* is pivotal to our understanding of

character-state changes at the base of Thyreophora that can shed light on the early evolution of the armoured dinosaurs, the taxon has received limited study. Herein, we provide a detailed account of the osteology of *Scutellosaurus lawleri*, figuring many elements for the first time. *Scutellosaurus lawleri* was the only definitive bipedal thyreophoran. Histological studies indicate that it grew slowly throughout its life, possessing lamellar-zonal tissue that was a consequence neither of its small size nor phylogenetic position, but may instead be autapomorphic, and supporting other studies that suggest thyreophorans had lower basal metabolic rates than other ornithischian dinosaurs. Faunal diversity of the Kayenta Formation in comparison with other well-known Early Jurassic-aged dinosaur-bearing formations indicates that there was considerable spatial and/or environmental variation in Early Jurassic dinosaur faunas.

## 1. Introduction

Thyreophoran dinosaurs (the armoured dinosaurs—stegosaurs, ankylosaurs and related forms) were important and diverse terrestrial herbivores from the Late Jurassic to the Late Cretaceous [1–3]. They split from their sister-clade, Neornithischia (including ornithopods, pachycephalosaurs and ceratopsians), by the Early Jurassic and diverged into two major clades (Stegosauria and Ankylosauria, together forming the clade Eurypoda) by the Middle Jurassic [1–6]. However, the paucity of taxa known from these early phases of thyreophoran evolution (Late Triassic–Middle Jurassic) hampers attempts to understand thyreophoran systematics, palaeobiology and evolution.

Owen [7,8] described the early thyreophoran *Scelidosaurus harrisonii* from the Early Jurassic (early Sinemurian–early Pliensbachian; [9]) of England. This taxon is represented by multiple well-preserved skeletons representing all parts of the skeleton; however, until recently very limited anatomical information had been published [10–12]. Other early thyreophorans are generally known from very limited material [13–17].

Colbert [18] described two specimens (MNA.V.175, MNA.V.1752) of a small, armoured ornithischian from the Lower Jurassic Kayenta Formation of Arizona, USA (figure 1) as *Scutellosaurus lawleri*, emphasizing similarities to the early-branching ornithischian *Lesothosaurus diagnosticus* from the Early Jurassic of South Africa. Colbert [18] assigned *Scutellosaurus lawleri* to the family Fabrosauridae but suggested that *Scutellosaurus lawleri* could possibly be a 'remote ancestor' of ankylosaurs and stegosaurs. Recent phylogenetic analyses and taxonomic reviews [1–5,20,25–33] have recovered *Scutellosaurus lawleri* as one of the earliest branching members of Thyreophora, and Fabrosauridae is now considered polyphyletic [1,3,5,26,31,34]. Detailed knowledge of the anatomy of *Scutellosaurus lawleri* is important for several reasons: (i) character polarization during analyses of stegosaurian and ankylosaurian phylogeny and thyreophoran functional evolution. This is especially important given there are several dozen specimens of *Scutellosaurus lawleri*, including several relatively complete postcranial skeletons with associated cranial material, making it one of the best-known early thyreophorans; (ii) determination of the phylogenetic position of Thyreophora within Ornithischia, and of the plesiomorphic character-states for Thyreophora in broad-scale analyses of archosaurian and dinosaurian evolution; and (iii) reconstruction of the character-state changes that occurred at the Thyreophora–Neornithischia split, and during early thyreophoran evolution.

Despite the obvious importance of *Scutellosaurus lawleri*, little further descriptive or systematic work has been carried out since that of Colbert [18] until very recently. Rosenbaum & Padian [29] referred six specimens from the collections of the University of California Museum of Paleontology (UCMP) to *Scutellosaurus lawleri* in 2000 but provided only very brief descriptions of most elements, and Tykoski [35] figured one additional specimen reposited at The University of Texas Vertebrate Paleontology Collections (TMM) in 2005. Furthermore, there are inaccuracies in the anatomical information provided by Colbert [18] and Rosenbaum & Padian [29], the only previous diagnosis [18], did not attempt to distinguish autapomorphies and symplesiomorphies, and many phylogenetically informative features have not been previously discussed. In 2020, Breeden & Rowe [20] described new specimens of *Scutellosaurus lawleri* reposited at TMM preserving features of skeleton not present or poorly preserved in the type or referred Museum of Northern Arizona (MNA) and UCMP specimens but refrained from re-describing any of those latter specimens. The primary aims of this paper are to expand and emend previous anatomical descriptions of the type and referred specimens reposited at MNA and UCMP and to report and describe specimens reposited at the Museum of Comparative

**Figure 1.** (a) Geologic map of the Adeii Eechii Cliffs on Ward Terrace in northern Arizona. General geographic regions in which specimens of *Scutellosaurus lawleri* have been collected from the Kayenta Formation are highlighted. Modified from Clark & Fastovsky [19] and Breeden & Rowe [20]. (b) Idealized outcrop lithostratigraphy and chronostratigraphy of Ward Terrace. Lithostratigraphy after Harshbarger *et al.* [21] and chronostratigraphy after Marsh *et al.* [22], Suarez *et al.* [23] and Marsh [24]. (c) Key to the field seasons that have yielded specimens of *Scutellosaurus lawleri* by the institution in which specimens are reposited and the year in which they were collected.

Zoology (MCZ) for the first time, providing new data on non-eurypodan thyreophoran morphology and a new life reconstruction for *Scutellosaurus lawleri* based on new palaeobiological interpretations.

## 1.1. Institutional abbreviations

AMNH, American Museum of Natural History, New York, NY, USA; CAMSM, Sedgwick Museum of Earth Sciences, University of Cambridge, Cambridge, UK; CMN, Canadian Museum of Nature, Ottawa, Ontario, Canada; DMNH, Denver Museum of Nature & Science, Denver, CO, USA; IVPP, Institute of Vertebrate Paleontology and Paleoanthropology, Beijing, People's Republic of China; MBLUZ, Museo de Biología de La Universidad del Zulia, Maracaibo, Zulia, Venezuela; MCZ,

Museum of Comparative Zoology, Harvard University, Cambridge, MA, USA; MCZ VPRA, the designated prefix for fossil reptile and amphibian specimens at MCZ; ML, Museu da Lourinhã, Portugal; MNA, Museum of Northern Arizona, Flagstaff, AZ, USA; NHMUK, Natural History Museum (formerly NHM, BMNH), London, UK; ROM, Royal Ontario Museum, Toronto, Canada; SAM-PK, Iziko South African Museum, Cape Town, South Africa; SGWG, Greifswalder Geologische Sammlungen, Universität Greifswald (formerly Ernst-Moritz-Arndt-Universität Greifswald), Greifswald, Germany; TMM, Texas Vertebrate Paleontology Collections, The University of Texas at Austin, Austin, TX, USA; UCMP, University of California Museum of Paleontology, Berkeley, CA, USA; UCMP V, designated prefix for UCMP localities; USNM, National Museum of Natural History, Smithsonian Institution, Washington, DC, USA; YPM, Peabody Museum, Yale University, New Haven, CT, USA; ZDM, Zigong Dinosaur Museum, Zigong, Sichuan, People's Republic of China.

## 2. Geologic setting

The specimens of *Scutellosaurus lawleri* described below were all collected from the Lower Jurassic Kayenta Formation on lands of the Navajo Nation in northern Arizona. The Kayenta Formation is one of several formations comprising the Upper Triassic to Lower Jurassic Glen Canyon Group on the Colorado Plateau. Together they comprise a thick assemblage of terrestrial sediments that accumulated in a back-arc basin formed by subduction of the Pacific plate under the western margin of North America [36]. The Kayenta Formation lies unconformably atop the Dinosaur Canyon Member of the Moenave Formation, which is a mixture of aeolian and ephemeral fluvial deposits formed mostly by sheet-wash run-off. Without a sustained fluvial system, the Moenave Formation preserved very few vertebrate fossils [37,38]. The Triassic–Jurassic boundary probably lies near the middle of the Moenave Formation [22,23,38]. The upper boundary of the Kayenta Formation interfingers with the upper Lower Jurassic aeolian Navajo Sandstone [39,40]. The Glen Canyon Group is made up almost entirely of fine, well-sorted sandstones, and it is widely understood to represent an immense, long-lived dune field or erg that accumulated from the latest Triassic until the end of the Early Jurassic. During this time, the region was characterized by an arid climate, with episodic wet intervals that deposited local fluvial and freshwater lacustrine sediments [38–43]. It was during one or more of these episodic wet intervals in which the specimens described below were buried [38].

The Kayenta Formation extends across the southern Colorado Plateau from eastern Arizona to western Utah. Geologists have divided it into a 'typical facies', or 'sandy facies', that forms the immense Vermillion Cliffs, which stretch east to west across the southern Plateau, and a 'silty facies' that is restricted to exposures in eastern Arizona [21,39,44,45]. Over nearly a century of field research, fossils have been discovered throughout exposures of both facies of the Kayenta Formation. However, by far the greatest numbers and taxonomic diversity of vertebrate fossils have been collected from exposures of the silty facies on the Navajo Nation in Arizona, including all the *Scutellosaurus lawleri* material described herein.

The silty facies represents a geographically and temporally localized fluvial interlude in what was otherwise a vast, prolonged aeolian landscape. It is a heterogeneous assemblage of coarse- to fine-grained clastic sediments that accumulated in channels of cross-bedded sands and lenses of overbank mud that represent a multi-channel fluvial system with low sinuosity and frequent flash flood stages that pulsed through sand dunes [39,42,43]. Also present are localized shallow freshwater lake and marsh deposits that produced lenses of freshwater limestone and localized beds of blue mudstone that are rich in tetrapod fossils and that also contain ostracods [46] and carbonized sphenopsids [38]. Complex interfingering along lengths of the Navajo Sandstone and the silty facies of the Kayenta Formation suggests that dominant southeasterly winds blew Navajo erg sands into the Kayenta stream systems [40]. Palaeocurrent analyses indicate that these streams generally flowed to the northwest and were deflated by the strong southeasterly winds. As a result, the fluvial system deposited sand along with the dune fields, and continually recycled sediment from the Navajo to the Kayenta system and back again [36,38,40,47,48].

The major exposures of the silty facies are along the eastern walls of the Little Colorado River Valley. Productive exposures for vertebrate fossils are mostly confined to Moenkopi Wash, near Tuba City, and to a line of cliffs extending to the south that is made up of the Chinle, Moenave, Kayenta and Navajo formations, in ascending order [18,19,35,38,49–51]. This escarpment bears various names, including the 'Adeii Eechii Cliffs' [52,53], the 'Tloi Eechii Cliffs' [54] and the Echo Cliffs [45]. The sources of these names are unknown to us, nor do the local residents use or even recognize them; however, the

name Adeii Eechii Cliffs is most frequently used in the scientific literature (e.g. [18–20,29,35,38,50,52,53], so we use it here.

To the south of Moenkopi Wash, the Adeii Eechii escarpment forms the western edge of the Moenkopi Plateau and the eastern flank of the Little Colorado River valley. Below the Adeii Eechii Cliffs is Ward Terrace, famous for its extensive badlands exposures of the Triassic Moenkopi and Chinle formations that are rich in vertebrate (mostly tetrapod) fossils [37]. From Moenkopi Wash southwards, exposures of the silty facies of the Kayenta Formation are steep and present little surface area for prospecting, although a few fossils have been recovered [55]. However, at the southernmost extent of the Adeii Eechii Cliffs, the underlying Dinosaur Canyon member of the Moenave Formation thickens to form 100 m tall cliffs that broaden considerably to the west above Ward Terrace. Resting on this broad bench are expansive and heavily dissected badlands of the silty facies of the Kayenta Formation from which the *Scutellosaurus lawleri* specimens described here were collected.

Most of the *Scutellosaurus lawleri* specimens were collected from fluvial and blue mudstone layers that lie approximately 20–30 m above the lower boundary of the Kayenta Formation. Among these is the type locality of *Sarahsaurus aurifontanalis* [38,56]. During laboratory preparation of the *Sarahsaurus aurifontanalis* holotype specimen from its encasing sandstone matrix, Adam Marsh extracted a large sample of detrital zircon crystals for dating. Preliminary laser ablation ICP-MS detrital zircon U-Pb results yielded a date of $183.7 \pm 2.7$ Ma [22]. This date establishes this part of the Kayenta Formation (recorded in most field notes as the informal 'middle third' of the Kayenta Formation) in the Pliensbachian and Toarcian stages of the Lower Jurassic. This is a considerably younger date than earlier U-Pb attempts based on smaller zircon samples that estimated a Triassic age for the Kayenta Formation [48]. The Early Jurassic U-Pb date is entirely consistent with Early Jurassic age estimates of the Kayenta Formation based on biostratigraphy [19,46,50,51,57–59].

# 3. History of discovery of the ornithischian dinosaurs of the Kayenta Formation

The first ornithischian dinosaur remains recognized from the Kayenta Formation were discovered in June 1971 by Mr. David Lawler, then a summer intern at the MNA. He collected a nearly complete postcranial skeleton of an armoured ornithischian dinosaur with a few cranial fragments that would become the holotype specimen of *Scutellosaurus lawleri* Colbert, 1981 (MNA.V.175) southeast of the topographic landmark Rock Head at MNA Locality 219-1. A second larger, but less complete, specimen of *Scutellosaurus lawleri* (MNA.V.1752) was discovered along Gold Spring Wash in July 1977 by Mr. William Amaral as part of MCZ in a joint expedition with MNA, under permits issued to MNA and funded by the National Science Foundation. This locality (MNA Locality 291-5) is approximately 16 km to the northwest of the holotype locality (figure 1). These two specimens were described by the late Edwin H. Colbert, who had become a Research Associate of the MNA following his retirement from a long career as Curator of Fossil Reptiles at the American Museum of Natural History [18]. Colbert [18] also referred to a femur from a much larger ornithischian than *Scutellosaurus lawleri* collected during that time; however, he did not cite a specimen number for the femur. We infer that this specimen is MNA.V.109, a large left femur referred to *Dilophosaurus wetherilli* by Gay [60]; however, we agree with Colbert's initial assignment of this specimen to Ornithischia.

Exploration of the silty facies continued during the late 1970s and early 1980s and resulted in the discovery of more ornithischian material by field parties from MNA, MCZ and UCMP (figure 1). In 1981, William W. Amaral of the MCZ collected a partial skeleton of a juvenile heterodontosaurid (MCZ VPRA-9092) from a microvertebrate locality near Gold Spring (MCZ Gold Springs Quarry 1 = TMM 45609), also informally known as the *Eocaecilia* Quarry [20,61]. That heterodontosaurid specimen has been mentioned by several authors [26,62,63] but still awaits a detailed description.

Field efforts by the UCMP during 1981 and 1983 led to the discovery of at least six new specimens of *Scutellosaurus lawleri* (figure 1), which were later described by Rosenbaum & Padian [29] (see Systematic palaeontology for further details about these UCMP specimens). In October 1981, that fieldwork by the UCMP also yielded osteoderms of an animal larger than *Scutellosaurus lawleri*. Similar osteoderms were previously reported by Colbert ([18]; MNA.V.96, MNA.V.136) and attributed to aetosaurs and used in support of his argument of a Triassic age for the Kayenta Formation. However, Padian [59] noted similarity between these osteoderms and those of the Early Jurassic thyreophoran dinosaur *Scelidosaurus harrisonii* from the Charmouth Mudstone Formation of England and referred the osteoderms from the Kayenta Formation to that species. Norman *et al.* [30] suggested a more

conservative assignment of 'Thyreophora indet.' for these osteoderms, which has been adopted by some authors [64] while others have opted for a cautious assignment of 'Scelidosaurus sp.' [35]. Regardless of their taxonomy, the presence of these osteoderms in the Kayenta Formation was used by Padian [59] to support the argument for an Early Jurassic age of the Kayenta Formation, an age assignment that is supported by the presence of both tritylodontids [65,66] and ostracods [46] with Early Jurassic affinities.

The most recent fieldwork in the Kayenta Formation to yield new ornithischian specimens was conducted between 1997 and 2000 by The University of Texas (figure 1), whose vertebrate palaeontological collections were an administrative unit of the Texas Memorial Museum (TMM) at that time but are now part of The Jackson School of Geosciences (figure 1). That work yielded over 40 new specimens of *Scutellosaurus lawleri* that were recently described by Breeden & Rowe [20], additional osteoderms from a larger unnamed thyreophoran like those reported by Padian [59], and other indeterminate ornithischian specimens yet to be fully described [20,67]. The most significant of those specimens are TMM 43663-1 and TMM 43664-1, both of which are partial skeletons of *Scutellosaurus lawleri* that preserve portions of the skull and pelvis not known from any other specimen [20].

# 4. Systematic palaeontology

Dinosauria [68]

Ornithischia [69]

Thyreophora [70] *sensu* [71]

Genus *Scutellosaurus* [18]

*Type Species*: *Scutellosaurus lawleri* Colbert, 1981 by monotypy.

*Scutellosaurus lawleri* [18]

*Etymology*: *Scutellosaurus*, from *scutellum* (Latin, a little shield) and *sauros* (Greek, lizard or reptile); *lawleri*, to honour David Lawler, who discovered and collected the holotype in June, 1971 [18].

*Holotype*: MNA.V.175, relatively complete postcranial skeleton, with some cranial fragments ([18]: figs 6–14, 16–24, 26–34; [28]: figs 20.1C–D, 20.3A,D; [30]: figs 15.1C–D, 15.5A,D). This specimen was first published using the retired catalogue number 'MNA P1.175' [18].

*Holotype locality*: MNA Locality 219-1, Rock Head (*Scutellosaurus lawleri* Locality). Ward Terrace, Coconino County, Navajo Nation, Arizona, USA (figure 1).

*Paratype*: MNA.V.1752, an incomplete postcranial skeleton of an individual slightly larger than the holotype ([18]: figs 15, 25, 31). This specimen was first published using the retired catalogue number 'MNA P1.1752' [18].

*Paratype locality*: MNA Locality 291-5, Gold Spring. Ward Terrace, Coconino County, Navajo Nation, Arizona, USA (figure 1).

*Referred specimens*: MNA.V.3133, MNA.V.3137, MNA.V.12395, UCMP 130580, UCMP 170829, UCMP 130581, UCMP 175166, UCMP 175167, UCMP 175168, MCZ VPRA-8792, MCZ VPRA-8793, MCZ VPRA-8794, MCZ VPRA-8795, MCZ VPRA-8796, MCZ VPRA-8797, MCZ VPRA-8798, MCZ VPRA-8799, MCZ VPRA-8800, MCZ VPRA-8801, MCZ VPRA-8802, MCZ VPRA-8803, MCZ VPRA-8804, MCZ VPRA-8805, MCZ VPRA-8806, MCZ VPRA-8808, MCZ VPRA-8810, MCZ VPRA-8820, TMM 43647-7, TMM 43647-8, TMM 43647-11, TMM 43647-12, TMM 43648-13, TMM 43656-2, TMM 43656-3, TMM 43656-5, TMM 43661-1, TMM 43663-1, TMM 43664-1, TMM 43664-2, TMM 43669-5, TMM 43669-6, TMM 43669-11, TMM 43670-5, TMM 43670-7, TMM 43670-8, TMM 43687-9, TMM 43687-13, TMM 43687-16, TMM 43687-17, TMM 43687-22, TMM 43687-42, TMM 43687-50, TMM 43687-57, TMM 43687-75, TMM 43687-81, TMM 43687-96, TMM 43687-112, TMM 43687-114, TMM 43687-115, TMM 43687-116, TMM 43687-121, TMM 43687-122, TMM 43687-123, TMM 43687-124, TMM 43690-6, TMM 43691-18, TMM 43691-20, TMM 45608-3, TMM 45609-4, TMM 45609-5, TMM 45609-6, TMM 47001-1. See electronic supplementary material, for detailed inventory of each MNA, UCMP and MCZ specimen listed and discussed herein. For a detailed inventory of TMM specimens, see Breeden and Rowe [20].

*Stratigraphic occurrence and provenance*: All known specimens of *Scutellosaurus lawleri* are derived from the 'silty facies' of the Kayenta Formation, Glen Canyon Group (Early Jurassic: Hettangian–Toarcian; [22,24]; figure 1) in Coconino County, Navajo Nation, Arizona, USA. Detailed locality information for all specimens described herein is on file at MCZ, MNA, TMM and UCMP and is available from those institutions to qualified researchers upon request; however, general locality information for each individual specimen is provided within the specimen inventory (electronic supplementary material).

*Revised diagnosis*: *Scutellosaurus lawleri* differs from all other ornithischians by the possession of the following autapomorphies: frontals are exceedingly narrow, maximum anteroposterior length more than 350% of minimum interorbital width; facets for atlantal neural arch extend onto the lateral surface of odontoid process; cervical vertebrae with very prominent and rugose ventral keels; preacetabular process of ilium is narrow dorsoventrally relative to length, dorsal and ventral surfaces are drawn out medially into distinct flanges (modified from [1,20]).

*Comments*: *Scutellosaurus lawleri* is the most abundantly recognized dinosaurian taxon from the Kayenta Formation ([35]; B.T.B. 2020, personal observation); however, in some cases, specimens referred to *Scutellosaurus lawleri* are represented only by small assemblages of associated but disarticulated ornithischian bones and/or bone fragments lacking diagnostic autapomorphies of *Scutellosaurus lawleri* [72,73]. These specimens nearly all preserve osteoderms superficially similar to those of MNA.V.175, and the size and shape of all specimens are more consistent with the type and referred specimens of *Scutellosaurus lawleri* than to the two other unnamed ornithischian taxa that have been recognized from the Kayenta Formation, which at present include a larger thyreophoran (MNA.V.96, MNA.V.136, UCMP 130056: [59]; TMM 45608-1: [35]) and a heterodontosaurid (MCZ VPRA-9092: [63,74]).

Specimens of *Scutellosaurus lawleri* reposited at MNA were initially published by Colbert [18] using the catalogue number format 'MNA P1.####' (e.g. 'MNA P1.175', 'MNA P1.1752'); however, shortly after that publication, MNA changed their catalogue number format for vertebrate fossil specimens to 'MNA V####' (e.g. 'MNA V175', 'MNA V1752'). That change was not published at the time, and as a result, several subsequent publications cited the retired 'MNA P1.####' catalogue numbers used by Colbert [18] for the type specimens of *Scutellosaurus lawleri* [1,16,17,29,34,35,75–77]. Additionally, the figure captions of Colbert [18] erroneously cited MNA.V.175 (='MNA P1.175') and MNA.V.1752 (='MNA P1.1752') as 'MNA Pl. 175' and 'MNA Pl. 1752', respectively, and these erroneous specimen numbers have proliferated in the literature [78–80]. Several additional erroneous specimen numbers have been published for MNA.V.175 and MNA.V.1752, including 'MNA 175' and 'MNA 1752' [81,82]; 'MNA PI.175' [83,84]; 'MNA Pl.175' and 'MNA Pl.1752' [2,79,80,85–88]; and 'MNA.Pl.1752' [89]. These variations are in part a result of the confusion of the numeral *one* ('1') with the uppercase letter 'I' (the first letter in the word 'India') and the lowercase letter '*el*' (the first letter in the word 'lima'), both by researchers and internally by MNA cataloguers.

Subsequently, when MNA palaeontology collection records were first digitized in 2000, representatives of the database provider recommended that collections managers not use spaces in specimen numbers, and the catalogue number format 'MNA.V.####' was adopted for vertebrate fossil specimens. For published specimen numbers to remain consistent with the MNA database, the catalogue number format of 'MNA.V.####' is now preferred for all MNA vertebrate fossil specimens (J. Gillette 2019, personal communication). This format is reflected herein and should be used henceforth when citing these and other MNA vertebrate fossil specimens.

Six specimens were referred to *Scutellosaurus lawleri* by Rosenbaum & Padian [29] and are reposited at the UCMP. These specimens were collected between 1981 and 1983 by field parties led by James M. Clark under permits issued to the MNA (J. Clark 2019, personal communication). All six of these specimens were reported to have been collected from the same locality (UCMP V85010—'Lower Blue') by Rosenbaum & Padian [29]; however, only four of the six specimens (UCMP 130580; UCMP 175166; UCMP 175167; UCMP 175168) were actually collected from that locality. The other two specimens UCMP 170829 and UCMP 130581 were collected from the localities UCMP V85013 ('Gold Springs 1') and UCMP V84235 (Red Knob), respectively. Additionally, UCMP 130580 was sometimes erroneously cited as 'UCMP 130850' by Rosenbaum & Padian [29] in their report of these referred specimens.

All referred specimens of *Scutellosaurus lawleri* that are reposited at MCZ were collected during joint expeditions between MCZ and MNA in 1978 and 1982 under Navajo Nation Antiquities Permits issued to MNA. Interestingly, several MCZ specimens that were collected during the 1978 field season were referred in the field to 'cf. *Kayentasaurus lawleri*', which reflects Colbert's tentative name for *Scutellosaurus lawleri* prior to its eventual description in 1981 (H.-D. Sues 2019, personal communication).

# 5. Osteological description

The following description is based upon the holotype specimen of *Scutellosaurus lawleri* (MNA.V.175) unless otherwise noted. Elements best preserved in previously described TMM specimens are noted

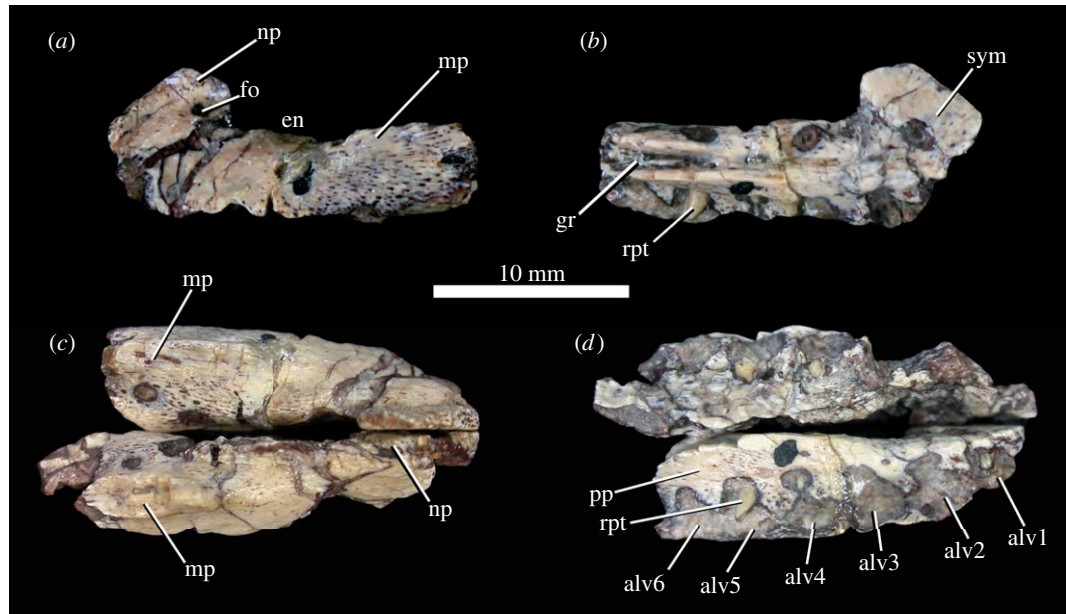

**Figure 2.** *Scutellosaurus lawleri*, holotype (MNA.V.175) left premaxilla in lateral (*a*) and medial (*b*) views, and left and right premaxillae in articulation in dorsal (*c*) and ventral (*d*) views. Abbreviations: alv1–alv6, alveoli; en, external naris; fo, foramen; gr, groove on medial surface of premaxilla for anterior process of maxilla; mp, maxillary process; np, nasal process; pp, premaxillary palate; rpt, replacement tooth; sym, flattened symphysial surface.

but not re-described here except where novel comparisons are made. For further information on those specimens, see Breeden & Rowe [20].

The following specimens were examined for comparison: *Agilisaurus louderbacki* (ZDM 6011 [holotype]); *Ankylosaurus magniventris* (AMNH 5895 [holotype]); *Dacentrurus armatus* (NHMUK PV OR46013 [holotype]); *Dacentrurus sp.* (= *Miragaia longicollum*; ML 433); *Edmontonia longiceps* (AMNH 3076; CMN 8531); *Emausaurus ernsti* (SGWG 85 [holotype]); *Eocursor parvus* (SAM-PK-K8025 [holotype]); *Hesperosaurus mjosi* (DMNH 29431 [cast of holotype]); *Heterodontosaurus tucki* (SAM-PK-K337 [holotype]; SAM-PK-K1332); *Hexinlusaurus multidens* (ZDM T6001 [holotype]); *Huayangosaurus taibaii* (IVPP 6728 [holotype]; ZDM T7001); *Hypsilophodon foxii* (NHMUK PV R197 [holotype]); *Jeholosaurus shangyuanensis* (IVPP V12529 [holotype]; IVPP V12530); *Laquintasaura venezuelae* (MBLUZ P.1396 [holotype]; MBLUZ P.5003; MBLUZ P.5014; MBLUZ P.1443); *Lesothosaurus diagnosticus* (NHMUK PV RUB17 [syntype]); *Panoplosaurus mirus* (CMN 2759; ROM 125); *Paranthodon africanus* (NHMUK PV OR47338 [holotype]); *Polacanthus foxii* (NHMUK PV R175 [holotype]); *Sauropelta edwardsi* (AMNH 3032 [holotype]); *Scelidosaurus harrisonii* (NHMUK PV R1111 [lectotype]; NHMUK PV R5909, NHMUK PV R6704; BRSMG LEGL 0005; BRSMG Ce12785; CAMSM X39256); *Stegosaurus stenops* (USNM 4934 [holotype]; YPM 1853; YPM 1856; YPM 1858 [composite]; NHMUK PV R36730); *Stormbergia dangershoeki* (SAM-PK-K1105 [holotype]); *Yinlong downsi* (IVPP V14530 [holotype]). Comparisons with *Hualianceratops wucaiwanensis* [90] and *Isaberrysaura mollensis* [91] were made based on literature.

## 5.1. Skull

### 5.1.1. Premaxilla

The premaxilla (figure 2) comprises a horizontally aligned main body (beneath the external naris), which is longer anteroposteriorly than wide mediolaterally. Posterodorsally directed maxillary (= 'posterolateral process') and nasal processes arise from this main body. The maxillary process is broken, and only the base is preserved on each side. The nasal process (figure 2*a–c*, np) projects from the anteromedial corner of the element at an angle of approximately 33° to the long axis of the main body, forming the anterior margin of the external naris. There is a sutural surface for the opposing element at the anterior end of the medial surface of each premaxilla (figure 2*b*). Together, the premaxillae form a long, narrow snout with a rounded anterior tip in dorsal view (figure 2*c*). As in other early ornithischians (e.g. *Lesothosaurus diagnosticus*: [34]), the first tooth of the premaxilla is inset a short distance from the anteriormost tip of

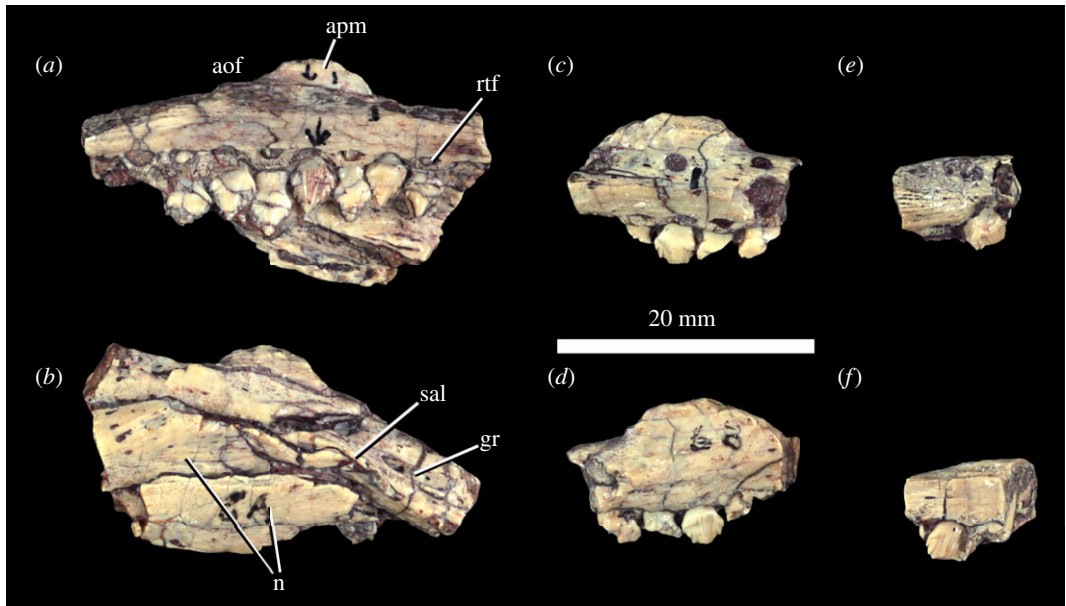

**Figure 3.** *Scutellosaurus lawleri*, holotype (MNA.V.175) maxillae. Partial left maxilla in medial (*a*) and dorsolateral (*b*) views. Fragments of right maxilla in medial (*c,e*) and lateral (*d,f*) views. Abbreviations: aof, ventral border of the antorbital fossa; apm, base of the ascending process of the maxilla; gr, groove on dorsal surface of posterior end of bone, medial to the supra-alveolar lamina; n, nasals appressed to lateral surface of left maxilla; rtf, replacement tooth foramina; sal, supra-alveolar lamina.

the premaxilla; the anteriormost tip has a rugose, pitted texture, indicating the presence of a horny beak. There are six alveoli (figure 2*d*), indicating six premaxillary teeth, although the teeth themselves are missing and only the tips of a few replacement crowns are present. Six premaxillary teeth are also present in *Lesothosaurus diagnosticus* (NHMUK PV RUB17), *Laquintasaura venezuelae* (MBLUZ P.5014; MBLUZ P.5016) and *Jeholosaurus shangyuanensis*. By contrast, some other early ornithischian dinosaurs have three (heterodontosaurids: [63]) or five (e.g. *Agilisaurus louderbacki*: [92]); the early thyreophorans *Emausaurus ernsti* [14] and *Scelidosaurus harrisonii* (BRSMG Ce12785) also have five, and the early diverging stegosaur *Huayangosaurus taibaii* (ZDM T7001; [93]) has seven. The early stegosaur *Isaberrysaura mollensis* has at least six premaxillary teeth, but the anterior end of the premaxilla is not preserved in the holotype and only specimen, so more may have been present [91].

The horizontal premaxillary palate is virtually absent between the two anteriormost teeth, but mediolaterally broadened posteriorly (figure 2*d*, pp), although the palate is comparatively narrow, and like that of other early ornithischians [34,63,76]. A deep and elongate anteriorly tapering groove (figure 2*b*, gr) is present on the medial surface of the premaxilla, which can be inferred to have accommodated a well-developed anterior process of the maxilla. A premaxillary foramen is present in the anteroventral corner of the narial fossa (MNA.V.175, [18]: fig. 9A; figure 2*a*, fo), like the condition in other early ornithischians [34,63,76].

### 5.1.2. Maxilla

The left maxilla (figure 3*a,b*) of the holotype was misidentified by Colbert [18] as the right maxilla, and the two fragments that were identified as the left maxilla by Colbert [18] probably represent the right maxilla (figure 3*c–f*). In the left maxilla of MNA.V.175, there are seven teeth present, as well as three empty alveoli, and the tooth row is slightly inset from the lateral margin of the bone, similar to the condition in *Lesothosaurus diagnosticus* [34], *Laquintasaura venezuelae* (MBLUZ P.5016) and *Emausaurus ernsti*. However, this buccal emargination is not as well developed as that in eurypodans [94,95]. The tips of unerupted replacement teeth are also present medial and dorsal to the tooth row, lying within replacement tooth foramina (figure 3*a*, rtf). The maxillary teeth show an alternating eruption pattern in which every other tooth is more advanced in eruption than the intervening tooth. The tooth row is oriented in a straight line in ventral and medial views. The ascending process is mostly missing, with only the base being preserved.

Owing to incomplete preservation, it is not possible to determine whether a secondary maxillary palate is present. As in most ornithischians [96], a supra-alveolar lamina (figure 3*b*, sal) projects

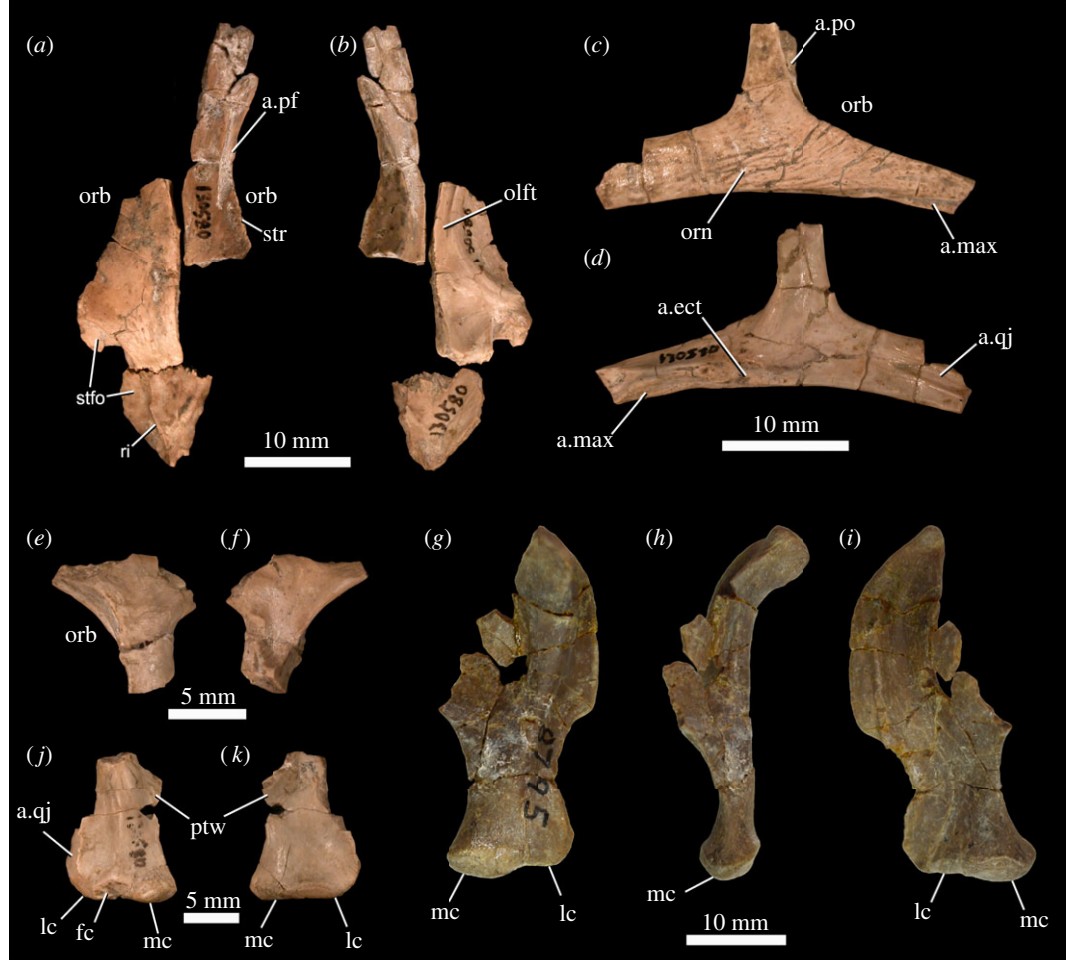

**Figure 4.** *Scutellosaurus lawleri*, referred cranial bones. Frontals and parietal of UCMP 130580 in dorsal (*a*) and ventral (*b*) views. Right jugal of UCMP 130580 in lateral (*c*) and medial (*d*) views. Left postorbital of UCMP 130580 in lateral (*e*) and medial (*f*) views. Left quadrate of UCMP 130580 in posterior (*j*) and anterior (*k*) views. Right quadrate of MCZ VPRA-8795 in posterior (*g*), medial (*h*) and anterior (*i*) views. Abbreviations: a.ect, articular surface for the ectopterygoid; a.max, articular surface for the maxilla; a.pf, articular surface for the prefrontal; a.po, articular surface for the postorbital; a.qj, articular surface for the quadratojugal; fc, facet; lc, lateral condyle; mc, medial condyle; olft, olfactory tract; orb, orbit; orn, ornamentation (cortical remodelling); ptw, pterygoid wing; ri, ridge; stfo, supratemporal fossa; str, striations.

dorsally above the posterior end of the tooth row at the lateral margin of the bone, forming the ventral margin of the antorbital fenestra. Medial to this, a posteriorly opening groove is present on the dorsal surface of the bone, representing the groove for the neurovascular bundle. In *Scelidosaurus harrisonii* (NHMUK PV R1111), this neurovascular groove connects with the row of lateral neurovascular foramina positioned above the tooth row. If such lateral foramina were present in *Scutellosaurus*, as is the normal condition in ornithischians [97], they are obscured by the nasals.

### 5.1.3. Nasal

Although portions of the nasals are preserved in MNA.V.175 (figure 3*b,n*), these are fractured and closely appressed to the left maxilla, so they yield limited anatomical information, and a more complete nasal is preserved in TMM 43664-1 ([20]: fig. 9A–D).

### 5.1.4. Frontal

Although a posterior portion of a left frontal is preserved in MNA.V.175 [20], more complete frontals are preserved in UCMP 130580 (figure 4*a,b*), TMM 43648-13 ([20]: fig. 2), TMM 43663-1 ([20]: fig. 3) and TMM 43664-1 ([20]: figs 10–11). The frontals are much longer anteroposteriorly than wide

mediolaterally, a feature we interpret to be an autapomorphy of *Scutellosaurus lawleri*. Similarly elongate frontals are also present in some neornithischian taxa (e.g. *Hypsilophodon foxii*, *Jeholosaurus shangyuanensis*) but appear unique among thyreophorans. The frontals widen towards their posterior ends and are dorsoventrally thickened where they form a midline contact with one another. They are not extensively sculpted, but there is some weak sculpting of the dorsal surface immediately adjacent to the parietal contact and supratemporal fossa at the posterior end of the element. In *Emausaurus ernsti*, the frontals are similarly unsculpted, except for a small region around the anterior orbital margin. The frontals are gently arched along their length. As in *Lesothosaurus diagnosticus* ([34]: fig. 12B), there is an elongate facet on the anterolateral surface of the frontal for a tapering posterior process of the prefrontal (figure 4*a*, a.pf). As noted by Maidment & Porro [98], there are a series of fine striations along the lateral margin of the frontal of UCMP 130580 (figure 4*a*, str), which were interpreted to represent an osteological correlate for soft tissue, and these striations are also present on the frontals of TMM 43663-1 and TMM 43664-1 [20] and on the frontals of *Emausaurus ernsti*. The frontal formed part of the dorsal rim of the orbit (figure 4*a*,*b*, orb), as in most early ornithischians [34,63,76,92] but differing from the condition in *Scelidosaurus harrisonii* (NHMUK PV R1111), ankylosaurs [99] and stegosaurs [94], in which supraorbital bones were incorporated into the skull roof and separated the frontals from the dorsal rim of the orbit. Medial to the orbital margin on the ventral surface of the frontal, there is an anteroposteriorly extending wide groove for the olfactory tract of the brain (figure 4*b*, olft). The posterior portion of the frontal possesses a prominent supratemporal fossa to accommodate the temporal musculature (figure 4*a*, stfo), as in saurischians [25] and some early ornithischians [34,76,100].

### 5.1.5. Parietal

The parietal of *Scutellosaurus lawleri* is known only from a fragment with many broken surfaces in UCMP 130580 (figure 4*a*,*b*). Anteriorly, the parietal formed a nearly straight, interdigitating, transverse suture with the frontals. On each lateral side of the parietal, there is a sharp ridge (figure 4*a*, ri) that demarcates the relatively large supratemporal fossa and is continuous with the ridge defining the supratemporal fossa (figure 4*a*, stfo) on the frontal. These ridges on the parietal converge posteriorly, and together define a flat, triangular area on the dorsal surface of the bone.

### 5.1.6. Jugal

A small portion of a left jugal is present in MNA.V.175, and a mostly complete right jugal is present in UCMP 130580 (figure 4*c*,*d*), but the most complete jugals are known from TMM 43663-1 and TMM 43664-1 [20]. Sutural surfaces are here identified based upon comparison with the disarticulated holotype skull of *Scelidosaurus harrisonii* (NHMUK PV R1111). The lateral surface of the jugal ventral to the dorsal process and ventral to the ventral margin of the orbit is ornamented with several weak anterodorsal-to-posteroventrally extending, anastomosing ridges and shallow grooves (figure 4*c*, orn). This ornamentation is similar in position and form to that of *Emausaurus ernsti* and *Scelidosaurus harrisonii* [11], although it is less well developed than in the latter. Similar jugal ornamentation is present in the neornithischians *Yinlong downsi*, *Hualianceratops wucaiwanensis* and *Jeholosaurus shangyuanensis*.

The anterior process of the jugal is mediolaterally expanded beneath the orbit (figure 4*c*). This mediolateral expansion does not occur along the entire anterior process but is confined to a short section immediately anterior to the dorsal process of the jugal. A medially and slightly ventrally facing facet for the ectopterygoid is present on the medial surface of this expansion (figure 4*d*, a.ect), like the condition in *Emausaurus ernsti* and *Scelidosaurus harrisonii* (NHMUK PV R1111). Two sutural surfaces are present on the jugal for the maxilla (figure 4*c*,*d*, a.max): an extensive ventrally and slightly medially facing surface, and a shorter and less well-defined ventrally and laterally facing surface. The sutural surface for the lacrimal, positioned on the dorsal surface of the anterior end of the process in *Scelidosaurus harrisonii* (NHMUK PV R1111), may be represented in *Scutellosaurus lawleri* by a small dorsal notch in the extreme end of the anterior process of the jugal of TMM 43664-1. The dorsal process of the jugal is anteroposteriorly narrow and has an elongate narrow groove on the anterior portion of the lateral surface to articulate with the ventral process of the postorbital (figure 4*c*, a.po), as in *Emausaurus ernsti*. The posterior process of the jugal has a V-shaped, anteriorly tapering depression on its medial surface, which articulates with the quadratojugal (figure 4*d*, a.qj); a facet with similar morphology is present in *Scelidosaurus* (BRSMG Ce12785).

### 5.1.7. Postorbital

A small portion of a left postorbital is present in UCMP 130580 (figure 4*e–f*) that was not figured by Rosenbaum & Padian [29], but TMM 43663-1 and TMM 43664-1 possess the best-preserved postorbitals ([20]: figs 3A,B, 10C,D, 11).

### 5.1.8. Squamosal

Although a possible portion of the squamosal in MNA.V.175 was noted by Colbert [18], we could not locate this; however, a complete squamosal is preserved in TMM 43663-1 ([20]: fig. 3K,L).

### 5.1.9. Quadrate

The quadrate is not known from the holotype or paratype specimens but is preserved in a number of other specimens (e.g. UCMP 130580 (figure 4*j,k*), UCMP 175166: [29]; TMM 43647-7, TMM 43687-121, TMM 43663-1, TMM 43664-1: [20]; MCZ VPRA-8795 (figure 4*g–i*); MCZ VPRA-8797). The head is arched posteriorly relative to the shaft (figure 4*h*) and is not fused to the paroccipital process. It is moderately compressed in lateral view, like that of *Stegosaurus* (e.g. NHMUK PV R36730). The pterygoid wing (figure 4*j,k*, ptw) is best preserved in TMM 43664-1 and is subtriangular, dorsoventrally tall, tapers anteromedially and bears a shallow fossa ([20]: fig. 10K,L), like that of *Stegosaurus* (NHMUK PV R36730). The lateral surface of the ventral end of the shaft bears an extensive articular surface for the quadratojugal (figure 4*j*, a.qj), which ventrally would have closely approached the quadrate condyles. As noted by Rosenbaum & Padian [29], the medial condyle is weakly enlarged relative to the lateral and resembles that of *Scelidosaurus harrisonii* (NHMUK PV R1111), and the lateral condyle tapers laterally where it curves anteriorly. A facet is present on the posteroventral surface of the condyles in UCMP 130580 (figure 4*j*, fc), but this is not present in any other specimen that preserves a quadrate (e.g. TMM 43663-1, TMM 43687-121). Nevertheless, it appears to be a natural feature and not a broken surface. Such a facet does not appear to be present in *Scelidosaurus harrisonii* (NHMUK PV R1111), whereas the quadrate is not preserved in *Emausaurus ernsti*.

### 5.1.10. Braincase

Only the basioccipital and the paroccipital process are known from the braincase of *Scutellosaurus lawleri*, and all examples of these elements are preserved disarticulated. None are well enough preserved to determine the exits of cranial nerves.

The basioccipital is preserved in MNA.V.175 (figure 5*a,b*), UCMP 130580 (figure 5*c,d*) and MCZ VPRA-8797 (figure 5*e–h*). A midline ridge is present on the ventral surface of the basioccipital separating the basioccipital recesses (figure 5*f*, bor); a similar ridge is also present in *Lesothosaurus diagnosticus* ([101]: fig. 9K], *Emausaurus ernsti*, *Scelidosaurus harrisonii* (NHMUK PV R1111) and *Agilisaurus louderbacki* (ZDM 6011). Anterior to the basioccipital recesses are rugose basal tubera (figure 5*f,g*, bt). The occipital condyle (figure 5*f,h*, occ) is convex and reniform in posterior view, with a concave dorsal surface demarcating the floor of the foramen magnum (figure 5*h*, fm). On the dorsal surface and lateral to the margin of the foramen magnum are roughened posterolaterally oriented surfaces for articulation with the exoccipitals (figure 5*e,h*, a.ex). The endocranial floor (figure 5*e*, ef) on the dorsal surface of the basioccipital is generally smooth and concave except for a thin median ridge at the anterior end.

Paroccipital processes are preserved in TMM 43663-1 [20] and MCZ VPRA-8797 (figure 5*i,j*). The paroccipital process flares laterally from the main body of the exoccipital-opisthotic. In posterior view, there are notches along the dorsal edge of the main body of the exoccipital-opisthotic presumably for articulation with the supraoccipital (figure 5*i*, a.so) and parietal (figure 5*i*, a.p). In anterior view, there is a rough subtriangular surface that is pierced by a small foramen, and there is a medial notch for articulation with the prootic (figure 5*j*, a.pr). Dorsal to this surface, there is a groove for articulation with the supraoccipital (figure 5*j*, a.so). The distal end of the paroccipital process is dorsoventrally expanded but is not pendant.

### 5.1.11. Dentary

A nearly complete left (figure 6*a,b*) and partial right (figure 6*c,d*) dentary are preserved in the holotype, and several partial dentaries are preserved in other specimens (e.g. MCZ VPRA-8797: figure 7*a–c*). The

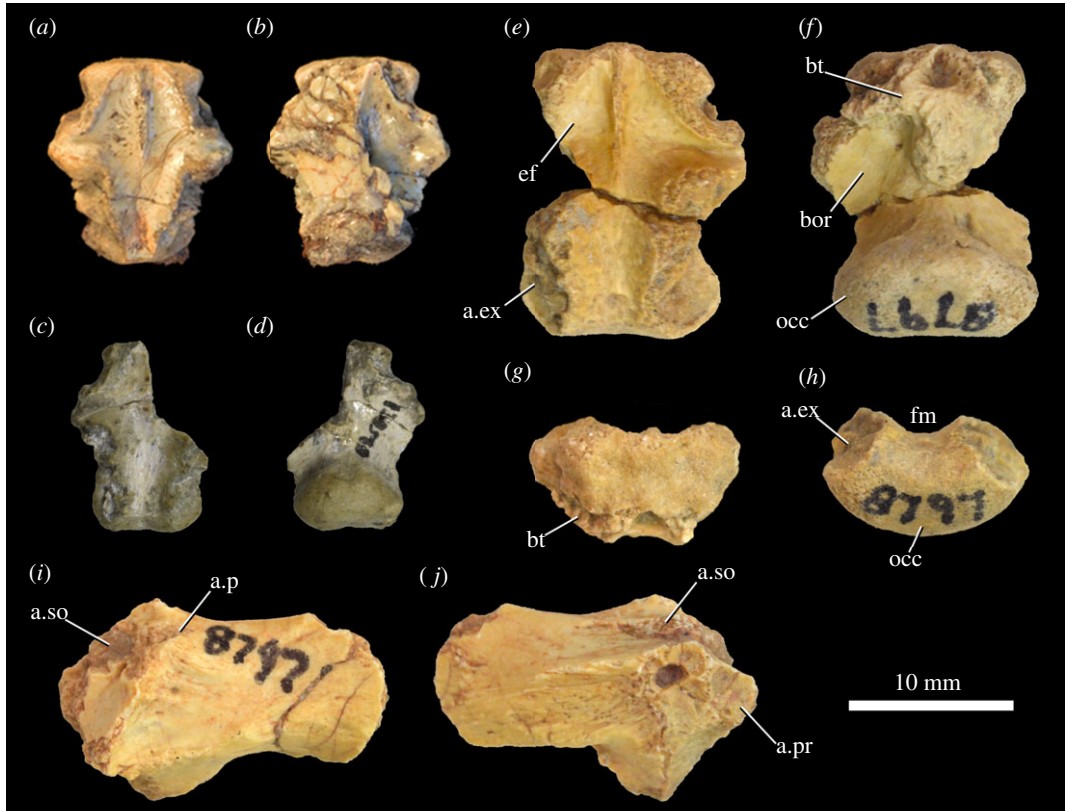

**Figure 5.** *Scutellosaurus lawleri*, braincase elements. Basioccipitals of the holotype (MNA.V.175; *a,b*), UCMP 130580 (*c,d*) and MCZ VPRA-8797 (*e,f*) in dorsal (*a,c,e*), ventral (*b,d,f*), anterior (*g*) and posterior (*h*) views. Paroccipital process of MCZ VPRA-8797 in posterior (*i*) and anterior (*j*) views. Abbreviations: a.ex, articular surface for the exoccipital; a.p, articular surface for the parietal; a.pr, articular surface for the prootic; a.so, articular surface for the supraoccipital; bor, basioccipital recess; bt, basal tubera; ef, endocranial floor; fm, foramen magnum; occ, occipital condyle.

dentary tooth row is slightly sinuous in dorsal view (figure 7*c*), but to a much lesser degree than that of derived nodosaurs (e.g. *Panoplosaurus mirus*, ROM 1215: [102]), and in lateral and medial views, the tooth row is sinuous with the anterior end downturned (figures 6 and 7) as in other thyreophorans (e.g. *Scelidosaurus harrisonii*, NHMUK PV R1111; *Emausaurus ernsti*, SGWG 85), but differing from early ornithischians [34,63,76]. There is a ridge on the dorsal half of the lateral surface that laterally defines a narrow buccal emargination posterior to the 10th tooth position (figures 6*a*, 7*a*, be). The ventral margin of the dentary is relatively straight, in contrast with the sinuous ventral margin in nodosaurs [32]. Colbert [18, p. 11] identified an unusual rugose depression on the ventral surface of the left dentary of MNA.V.175 as a possible pathologic feature (figure 6*a*); a similar depression is also present in an equivalent position of the right dentary, but one is not present on the referred dentary of MCZ VPRA-8797 (figure 7*a–c*).

There is a broad, flat facet on the medial surface of the dentary for the symphysis (figure 6*b*, sym), although this is not 'spout-shaped' as in most other ornithischians [27,34], indicating that the articulation between the lower jaws would have been V-shaped in dorsal view. The tooth row does not extend to the anterior end of the dentary, and there are clear dorsal and ventral facets for articulation with the predentary (figure 7*d,e*, a.pd); however, no predentary is preserved in any specimen. This is similar to *Lesothosaurus diagnosticus*, which also bears facets for the predentary on the edentulous anterior end of the dentary ([34]: fig. 10D,E). By contrast, the dentary of *Emausaurus ernsti* possesses little evidence for a predentary. The maximum dorsoventral depth of the dentary ramus is at the posterior end, and the depth of the dentary at the symphysis is shallower than half the maximum depth of the dentary in lateral view. The Meckelian groove on the medial surface (figures 6 and 7, Mg) does not extend anteriorly as far as the symphysial facet, in contrast with the condition in *Emausaurus ernsti*. Small fragments of the splenial (figure 6*b*, spl) are present in the posterior end of the Meckelian groove of the left dentary of the holotype. Multiple nutrient foramina

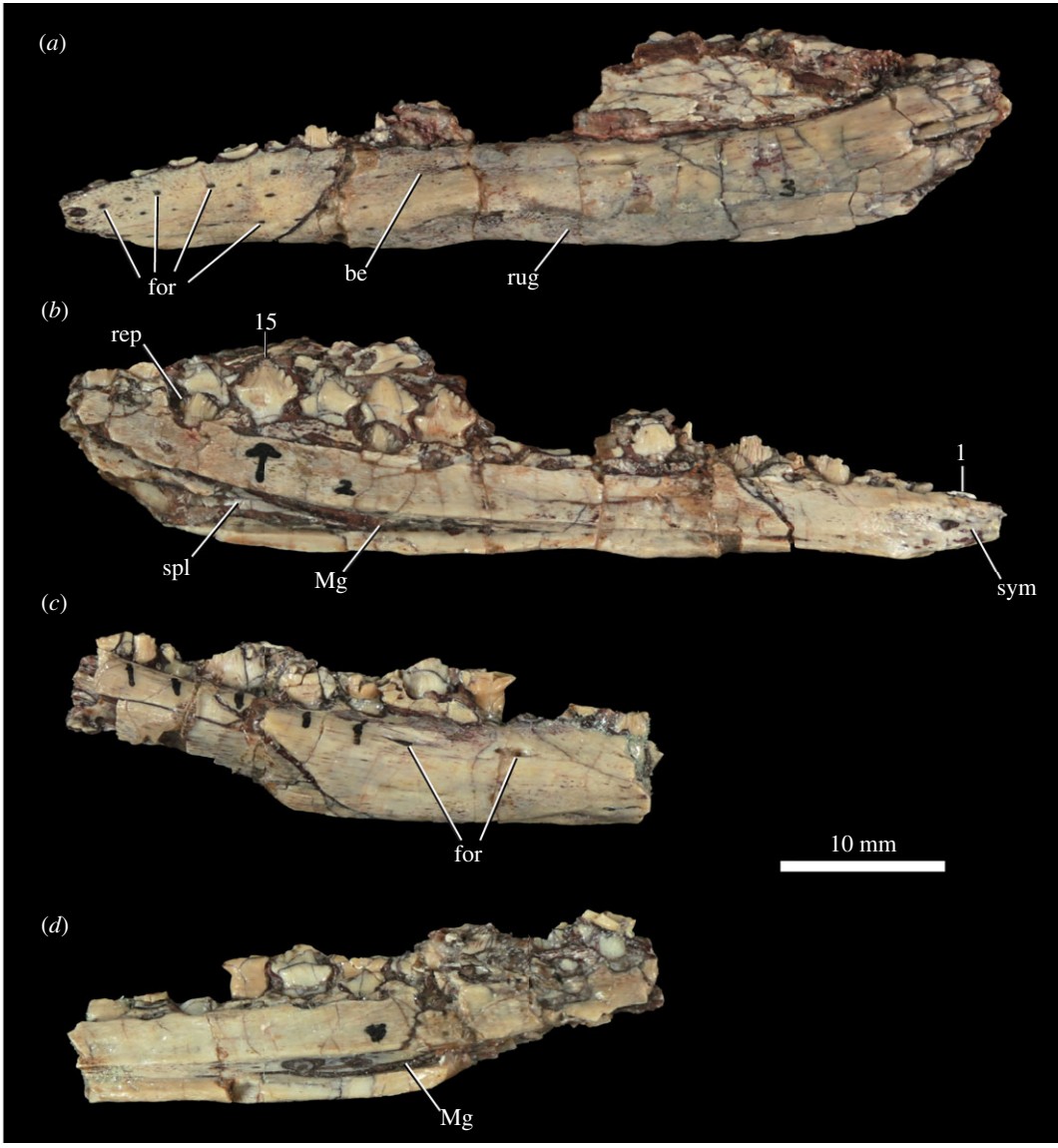

**Figure 6.** *Scutellosaurus lawleri*, holotype (MNA.V.175) dentaries. Left dentary in lateral (*a*) and medial (*b*) views. Right dentary in lateral (*c*) and medial (*d*) views. Abbreviations: be, buccal emargination; for, foramen; Mg, Meckelian groove; rep, replacement tooth; rug; rugose surface; spl, splenial; sym, symphysial surface for the opposing dentary; 1, first tooth position; 15, 15th tooth position.

are present on the lateral surface of the anterior end of the left dentary (figures 6 and 7, for). Similar foramina are also present on the dentaries of *Emausaurus ernsti*.

The size of the dentary tooth crowns increases posteriorly, and in MNA.V.175, the largest dorsoventral crown height is in the 14th tooth position. The tooth alveoli face dorsally, rather than dorsomedially as occurs in some eurypodans (e.g. *Stegosaurus stenops*, NHMUK PV R36730; *Edmontonia longiceps*, CMN 8531), and there is no lateral lamina present to obscure the tooth row in lateral view.

### 5.1.12. Surangular

The surangular is known only from UCMP 130580 (figure 7*f*,*g*). It is a mediolaterally compressed element that is generally convex on its lateral surface, concave on its medial surface and sigmoidal in lateral view. The surangular generally resembles that of *Lesothosaurus diagnosticus* (NHMUK PV RUB17), although it is broken anteriorly and dorsally, so the height of the coronoid process cannot be established with certainty. A prominent anteroposteriorly extending ridge is present on the lateral surface of the surangular (figure 7*f*, lr), anterodorsal to the jaw articulation, as in *Lesothosaurus diagnosticus* ([34]: fig. 13F),

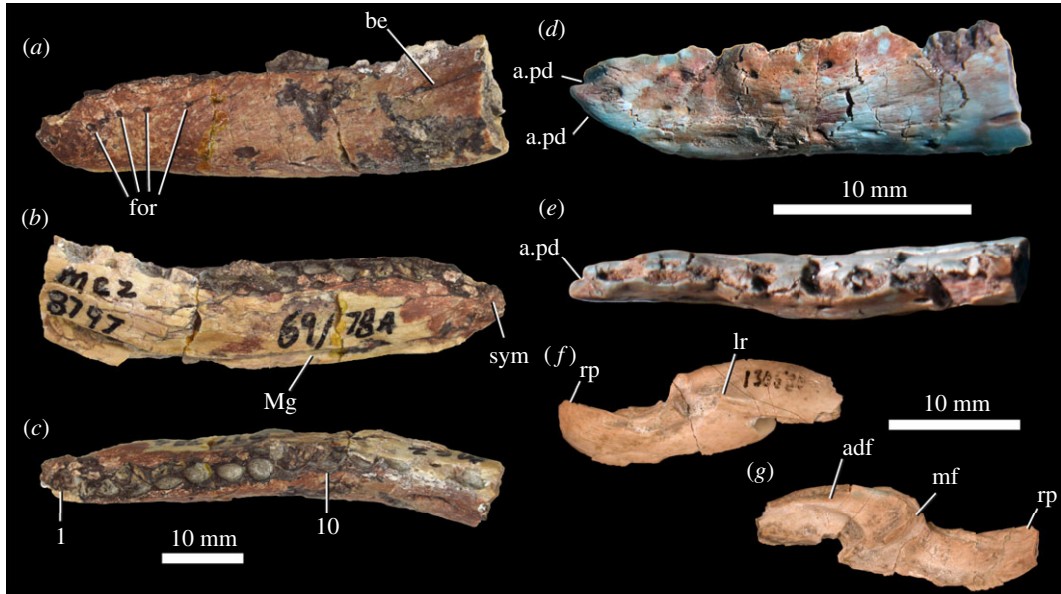

**Figure 7.** *Scutellosaurus lawleri*, referred mandibular bones. Left dentary of MCZ VPRA-8797 in lateral (*a*), medial (*b*) and dorsal (*c*) views. Left dentary of TMM 43647-11 in lateral (*d*) and dorsal (*e*) views. Right surangular of UCMP 130580 in lateral (*f*) and medial (*g*) views. Abbreviations: a.pd, articulation with the predentary; adf, adductor fossa; be, buccal emargination; for, foramen; lr, lateral ridge; mf, medial flange; Mg, Meckelian groove; rp, retroarticular process; sym, symphysis with antimere; 1, first tooth position; 10, 10th tooth position.

*Scelidosaurus harrisonii* (NHMUK PV R1111) and *Emausaurus ernsti*. This feature appears to have been lost in stegosaurs [103] and ankylosaurs [95]. The ridge is damaged posteriorly, and the presence or absence of a surangular foramen cannot be determined. Posteriorly there is a well-developed, slightly upturned, retroarticular process (figure 7*f*,*g*, rp), the lateral surface of which is gently striated. Medially there is a well-developed internal adductor fossa (figure 7*g*, adf), and a medial flange that descends posteroventrally from the inferred position of the glenoid (figure 7*g*, mf).

### 5.1.13. Dentition

The dentition of the holotype was well described by Colbert [18] and that description is supplemented here. The premaxillae contain alveoli for six teeth, but all are empty and only one replacement tooth, the fifth on the left premaxilla of MNA.V.175, is sufficiently erupted to decipher any morphology. It is triangular with a sharp, elongate apex, and fine denticles are present. There are no striations or ridges on the crown surface.

The maxillary teeth (figure 8*a*) are subtriangular, symmetrical and taller dorsoventrally than wide anteroposteriorly. They closely resemble those of *Lesothosaurus diagnosticus* [34] but are very distinct from the apicobasally expanded teeth of *Laquintasaura venezuelae* [104]. There is a labiolingual swelling present at the base of the crown that sits above a constricted tooth root, similar to that in *Emausaurus ernsti* and *Scelidosaurus harrisonii* (NHMUK PV R1111), but not developed into a true cingulum like that of more derived thyreophorans [94,95]. There is a broad central eminence which extends to the crown tip. There are eight marginal denticles present on each tooth, four either side of the central eminence. By contrast, the maxillary teeth of *Emausaurus ernsti* are asymmetrical with the crown tips offset slightly distally. There are four marginal denticles on the shorter distal side in *Emausaurus ernsti*, and six are present on the longer mesial side. In *Scelidosaurus harrisonii* (NHMUK PV R1111), the maxillary tooth crowns are also offset distally, and the marginal denticles are smaller and more numerous [105]. Striations confluent with the denticles in *Scutellosaurus lawleri* are not present on the surfaces of the maxillary crowns. Wear facets are not evident on the maxillary teeth, like *Emausaurus ernsti* but in contrast with *Scelidosaurus harrisonii* [105], although this could be due to poor preservation.

The dentary teeth (figure 8*b*) are similar in morphology to the maxillary teeth. There are six marginal denticles either side of a central apex, although these denticles appear to be larger than on the maxillary teeth, and there is a labiolingual swelling present that is not developed into a true cingulum.

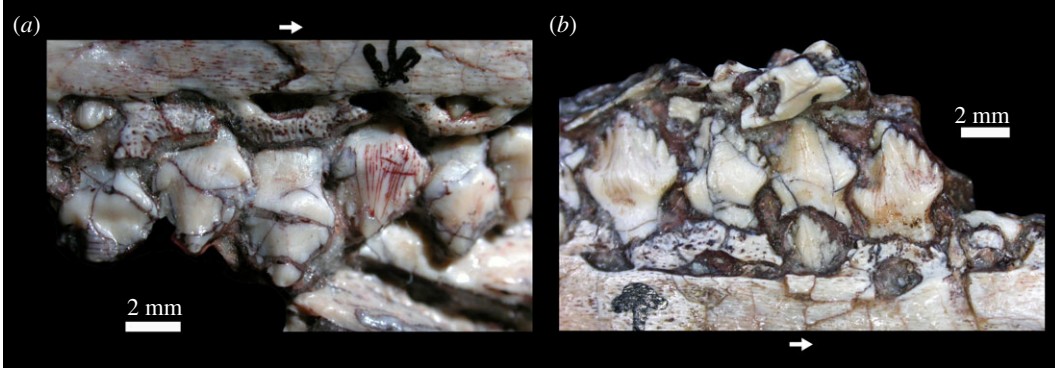

**Figure 8.** *Scutellosaurus lawleri*, holotype (MNA.V.175) maxillary (*a*) and dentary (*b*) teeth, both in lingual view. Arrows point in the anterior direction.

## 5.2. Postcrania

### 5.2.1. Cervical vertebrae

Colbert [18] suggested that there were nine cervical vertebrae in *Scutellosaurus lawleri*, accounting for the atlas and axis and inferring the absence of the fourth cervical vertebra; however, only six postaxial cervical centra are actually preserved (figures 9 and 10). MCZ VPRA-8800 possesses six well-preserved postaxial cervical vertebrae and one postaxial cervical centrum, which is consistent with Colbert's inference of nine total cervical vertebrae. This is typical of other small-bodied ornithischians (e.g. *Heterodontosaurus tucki*, [104]; *Agilisaurus louderbacki*, [92]) but differs from *Scelidosaurus harrisonii* (NHMUK PV R1111, CAMSM X39256; [11]: fig. 10), which possesses eight cervical vertebrae.

Elements from the atlas-axis complex are sparsely preserved among specimens of *Scutellosaurus lawleri*. The intercentrum of the atlas is preserved in UCMP 130580 (figure 9*a,b*), a right atlas neural arch is preserved in MNA.V.175 (figure 9*c*), and an odontoid is preserved in UCMP 130580 (figure 9*d–f*). These elements of the atlas remain discrete and unfused in the specimens in which they are preserved. The neural arches are also unfused to the atlas intercentrum in *Emausaurus ernsti* and *Scelidosaurus harrisonii* (NHMUK PV R1111; CAMSM X39256). In the latter, the odontoid is fused to the axis centrum in larger individuals (NHMUK PV R1111), but not in smaller specimens (CAMSM X39256). The intercentrum is reniform in anterior view with a concave dorsal margin and a convex ventral margin. It possesses an anterodorsal depression for articulation with the basioccipital and a posterodorsal groove to receive the ventral lip of the odontoid. The lateral margins of the intercentrum are rounded as preserved and lack clearly defined facets for the neural arches or an atlantal rib. The neural arch has a short anterior process that includes the prezygapophysis (although this articular facet is not well preserved), whereas the postzygapophysis was presumably on the medial surface of the long posterior process of the neural arch but is not exposed. The odontoid process is wedge-shaped and is notable for the presence of prominent concave facets on its lateral surfaces (figure 9*d, cf*). This feature is not seen in other ornithischians for which there is comparable material, and thus is an autapomorphy of *Scutellosaurus lawleri*.

The axis is known from a posterior portion of a neural arch in UCMP 130580 (figure 9*g–i*) and a centrum in MCZ VPRA-8801 (figure 9*j–l*). The neural spine (figure 9*g*) is prominent and extends posterodorsally beyond the posterior margins of the postzygapophyses, broadening slightly posteriorly, as in *Scelidosaurus harrisonii* (NHMUK PV R1111; [11]: fig. 7). A midline ridge extends along the length of the dorsal surface of the neural spine. Only the right postzygapophysis (figure 9*g,i*, poz) is preserved, and it faces mostly ventrally and slightly laterally. The axis centrum of MCZ VPRA-8801 preserves portions of the base of the neural arch (figure 9, na) that show a circular neural canal (figure 9, nc), and the atlas intercentrum is not co-ossified to its anterior articular surface to form the odontoid. The axis centrum is cylindrical and wider than tall. The anterior articular surface is a mediolaterally wide oval bisected by a prominent dorsoventral ridge. There is a shallow facet on the dorsal half of the anterior articular surface for the odontoid (figure 9, odf) and a smaller but more prominent facet along the ventral edge for articulation with the atlas intercentrum (figure 9, axicf). The parapophyses are smooth and rounded (figure 9: para). The subcircular and concave posterior articular surface is mediolaterally narrower than the anterior surface.

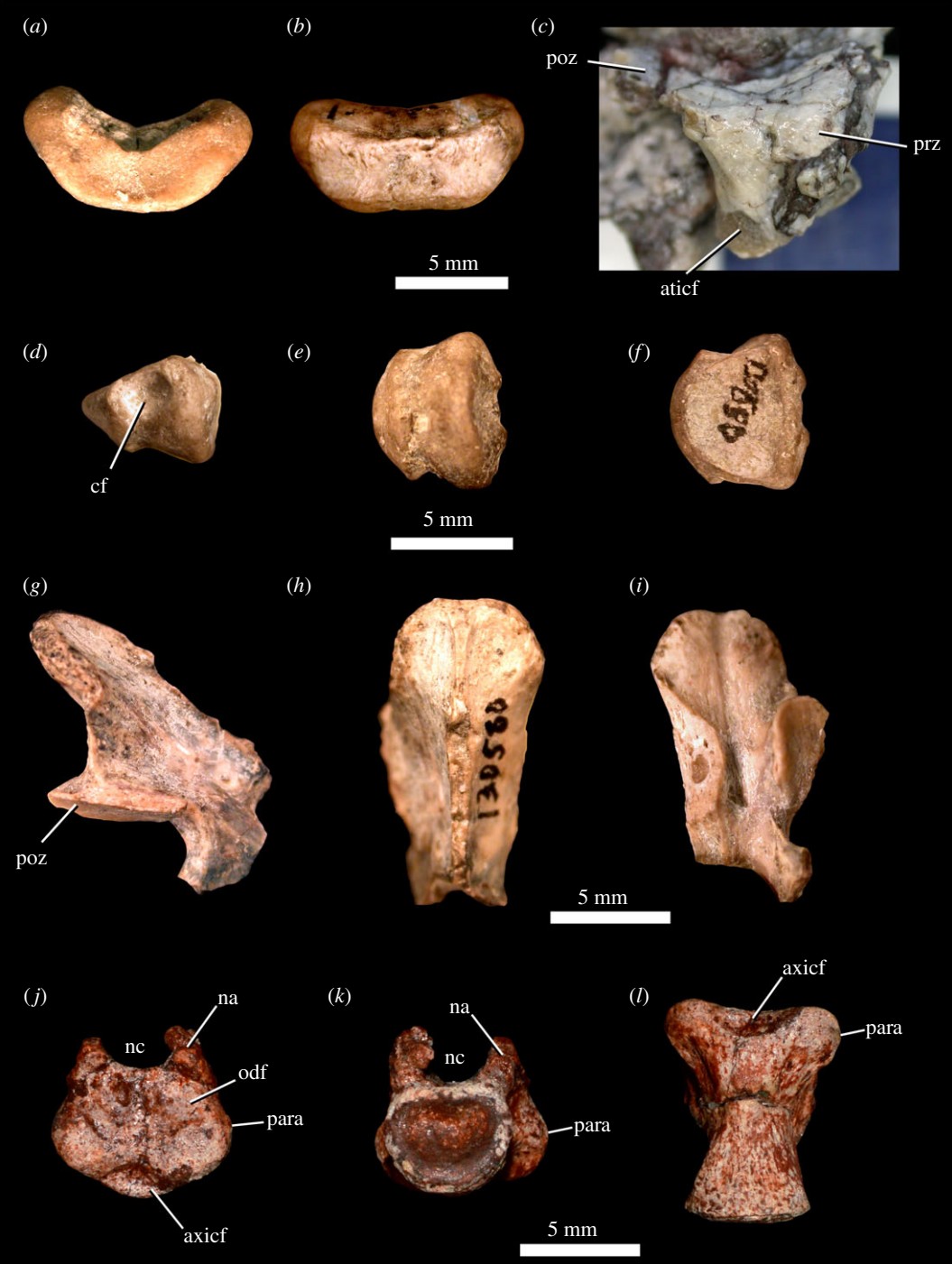

**Figure 9.** *Scutellosaurus lawleri*, atlas and axis. Referred (UCMP 130580) atlas intercentrum in anterior (*a*) and posterior (*b*) views. Holotype (MNA V.175) right atlas neural arch in lateral view (*c*). Odontoid of UCMP 130580 in left lateral (*d*), ventral (*e*) and dorsal (*f*) views. Referred (UCMP 130580) axis neural arch in right lateral (*g*), dorsal (*h*) and ventral (*i*) views. Referred (MCZ VPRA-8801) axis centrum in anterior (*j*), posterior (*k*) and ventral (*l*) views. Abbreviations: aticf, articular facet for the atlas intercentrum on the atlas neural arch; axicf, articular facet for the atlas intercentrum on the axis; cf, concave facet on lateral surface of odontoid, probably for atlas neural arch; na, neural arch; nc, neural canal; odf, facet for odontoid; para, parapophysis; poz, postzygapophysis; prz, prezygapophysis.

Postaxial cervical centra are commonly preserved in specimens of *Scutellosaurus lawleri*, but complete cervical vertebrae preserving neural arches are rare. Posterior cervical centra are shorter anteroposteriorly than anterior centra. The sides of the centra are strongly constricted so that the lateral surfaces are concave dorsoventrally and anteroposteriorly. The ventral margin of the third cervical vertebra is

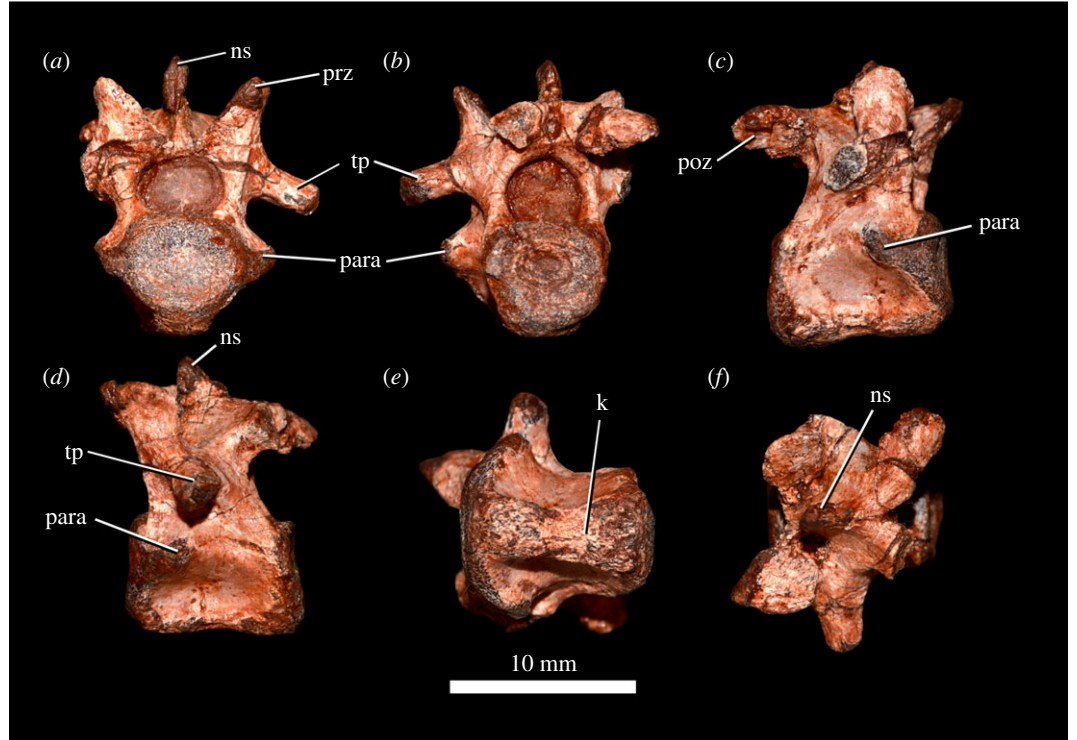

**Figure 10.** *Scutellosaurus lawleri*, referred (MCZ VPRA-8800) cervical vertebra in anterior (*a*), posterior (*b*), right lateral (*c*), left lateral (*d*), ventral (*e*) and dorsal (*f*) views. Abbreviations: k, keel; ns, neural spine; para, parapophysis; poz, postzygapophysis; prz, prezygapophysis; tp, transverse process.

concave upwards in lateral view. An extremely rugose and well-developed keel (figure 10*k*) is present on the mid to posterior cervical vertebrae. This keel is swollen and tubercular towards its anterior end. Similar keels are present on the cervical and dorsal vertebrae of *Scelidosaurus harrisonii* (CAMSM X39256, NHMUK PV R1111), as well as in other early ornithischians (e.g. *Hypsilophodon foxii*, [106]), although these are less prominent that those of *Scutellosaurus lawleri*; the possession of very prominent and rugose ventral keels is therefore considered an autapomorphy of *Scutellosaurus lawleri*.

One nearly complete cervical vertebra was described by Breeden & Rowe ([20]: fig. 17C–G; TMM 45609-6), and the following description is based upon a single well-preserved posterior cervical vertebra (MCZ VPRA-8800; figure 10). The centrum of MCZ VPRA-8800 is weakly amphicoelous, and the anterior and posterior articular facets of the centra are nearly parallel to one another in lateral view. The parapophyseal facet (figure 10, para) is ovoid, with a substantial portion on the dorsolateral margin of the centrum and a small portion on the ventrolateral margin of the neural arch. The neural arch is taller than the centrum, and the neural spine (figure 10, ns) is incomplete. The anterior edge of the neural spine originates between the prezygapophyses (figure 10, prz). The prezygapophyses are anterodorsally directed and widely separated from one another with broad, subcircular articular facets that face dorsomedially, like those in *Stegosaurus* (NHMUK PV R36730). Only the left transverse process is completely preserved (figure 10, tp), and it is ventrolaterally directed, contrary to the dorsolaterally directed transverse processes of the posterior cervical vertebrae of *Scelidosaurus harrisonii* [12, p. 19]. The transverse process (figure 10, tp) narrows distally, terminating in a truncated convex diapophyseal facet. The postzygapophyses are posterolaterally directed with nearly horizontal articular facets. Both the pre- and postzygapophyses extend beyond the articular facets of the centra (figure 10*c*,*d*).

### 5.2.2. Dorsal vertebrae

Fifteen dorsal vertebrae, represented by 14 centra and one neural arch, are present in the holotype, along with one vertebra that is transitional in morphology between the dorsal and sacral vertebrae (a dorsosacral vertebra). Colbert [18] interpreted the series to represent a complete dorsal column. A count of 15 dorsal vertebrae is typical for small ornithischians [92,106], although only 12 are present in *Heterodontosaurus tucki* [107], and there are 16 in *Scelidosaurus harrisonii* (NHMUK PV R1111; CAMSM X39256). In the holotype, the centra and neural arches are disarticulated, and few of the

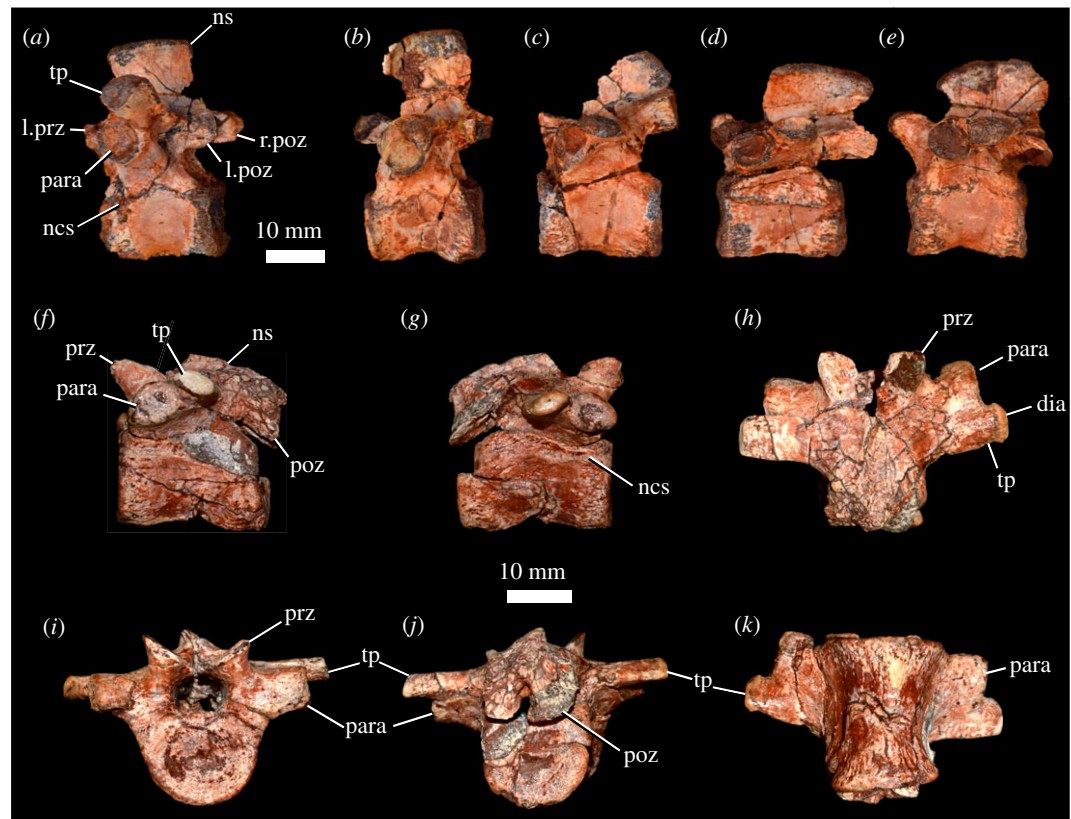

**Figure 11.** *Scutellosaurus lawleri*, referred dorsal vertebrae. (*a–e*) Series of dorsal vertebrae of MCZ VPRA-8800 in lateral view. (*f–k*) Dorsal vertebra of MCZ VPRA-8801 in left lateral (*f*), right lateral (*g*), dorsal (*h*), anterior (*i*), posterior (*j*) and ventral (*k*) views. Abbreviations: dia, diapophysis; l.poz, left postzygapophysis; l.prz, left prezygapophysis; ncs, neurocentral suture; ns, neural spine; para, parapophysis; poz, postzygapophysis; prz, prezygapophysis; r.poz, right postzygapophysis; tp, transverse process.

neural arches are well preserved. The neurocentral sutures appear to have been unfused at time of death, and this is also seen in some other specimens of *Scutellosaurus lawleri*. There exist only a few complete presacral vertebrae with articulated neural arches and centra, although in most cases the neurocentral suture remains visible. Among these, MCZ VPRA-8801 preserves five nearly complete non-sequential dorsal vertebrae (figure 11*a–e*), and MCZ VPRA-8800 preserves a single complete dorsal vertebra (figure 11*f–k*).

The centra are anteroposteriorly longer than they are wide mediolaterally or tall dorsoventrally. There are no longitudinal keels on the ventral surfaces of the centra, and the lateral surfaces are gently concave anteroposteriorly, giving the centra a spool-shape in ventral view. The anterior and posterior articular facets of the centra are flat to slightly concave. Neural arches are dorsoventrally taller than the dorsoventral height of the centra. The neural canal is circular in cross-section. Parapophyses (figure 11, para) are subcircular in outline, concave, smooth and positioned on short stalks. The parapophyses lie anteroventral to the transverse processes on anterior dorsal vertebrae, but extending posteriorly down the column, the parapophyses are progressively more dorsally positioned such that the transverse processes of posterior dorsal vertebrae lie almost directly posterior to the parapophyses. The dorsal surfaces of the stalks of the parapophyses and the transverse processes are confluent on posterior dorsal vertebrae. Transverse processes (figure 11, tp) are dorsoventrally compressed. They project laterally and slightly dorsally in anterior dorsal vertebrae (figure 11*a*), but laterally in more posterior dorsal vertebrae, and they become more dorsoventrally compressed extending down the column (figure 11*e*). The transverse processes bear diapophyses (figure 11, dia) at their distal ends which are more dorsoventrally compressed than the parapophyses and teardrop shaped with the apex pointing posteriorly in anterior dorsal vertebrae. Extending posteriorly down the dorsal vertebral column, the diapophyseal facets become more strongly dorsoventrally compressed. The facets themselves are concave and slightly smaller than the parapophyses. Prezygapophyses (figure 11, prz) extend anteriorly, are separated from each other and face dorsally. Postzygapophyses (figure 11, poz) are either sub-equal in size to the prezygapophyses (in the MCZ specimens; figure 11) or are smaller (in

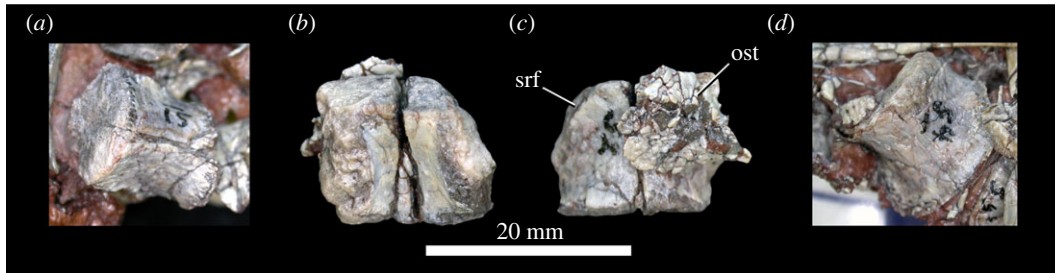

**Figure 12.** *Scutellosaurus lawleri*, holotype (MNA.V.175) sacral vertebrae. Transitional dorsosacral (labelled S1) exposed on a block in ventral view (*a*); true sacral 1 (labelled S2) in dorsal (*b*) and ventral (*c*) views. True sacral 3 (labelled S4) exposed on a block in ventral view (*d*). Abbreviations: ost, osteoderm; scf, sacral rib facet. Scale bar relates only to *b* and *c*; a scale bar is not given for *a* and *d* because the specimens are preserved in blocks of other material and thus the photos are somewhat oblique.

the holotype) and were separated by a deep midline fossa. Both pre- and postzygapophyses overhang the articular facets of the centrum. Neural spines (figure 11, ns) are subrectangular in lateral view. They are dorsoventrally tall and anteroposteriorly narrow in anterior dorsal vertebrae, and become shorter, anteroposteriorly wider and posteriorly positioned on the neural arch in posterior dorsal vertebrae. In the posterior dorsal vertebrae, they are longer anteroposteriorly than they are tall dorsoventrally in lateral view. In *Scelidosaurus harrisonii* (CAMSM X39256), neural spines are dorsoventrally taller than they are long anteroposteriorly.

### 5.2.3. Sacral vertebrae

Colbert [18] identified five sacral vertebrae present in the holotype of *Scutellosaurus lawleri*; they are not fused to one another and only the centra are preserved. The centrum identified as S1 by Colbert [18] appears to be transitional in morphology between the dorsal vertebrae and sacral vertebrae (figure 12*a*); it is longer anteroposteriorly than it is wide mediolaterally, as in the dorsal vertebrae but in contrast with the other sacral vertebrae, and does not appear to have well-developed sacral rib facets on the lateral surfaces of the centrum. We therefore identify this vertebra as a dorsosacral. The lateral surfaces of the centrum are flat anteroposteriorly and gently convex dorsoventrally. The articular facets are sub-quadrate and flat. The sacral vertebrae identified as 2–4 by Colbert ([18]; figure 12*b–d*) are wider mediolaterally than they are dorsoventrally tall or long anteroposteriorly. Their articular facets are reniform in anterior and posterior views. The anterior articular facet of the sacral identified as S2 by Colbert ([18]; figure 12*b*) is flat; the articular facets on the other sacral vertebrae are gently concave (e.g. figure 12*d*). The anterior and posterior articular facets are sub-equal in size in the vertebrae identified by Colbert [18] as S2–3; in S4 the anterior articular facet is larger than the posterior one. Large sacral rib facets occupy the anterolateral surfaces of the centra; it appears that each sacral rib articulated with just one vertebra. Ventrally, the sacral centra lack longitudinal grooves or keels. Unfortunately, the vertebra identified as S5 by Colbert [18] is obscured under other bone fragments and its morphology is unclear. TMM 43664-1 preserves one dorsosacral and four articulated sacral centra, but no other specimen of *Scutellosaurus lawleri* preserves a complete sacrum. The sacrum of *Scutellosaurus lawleri* therefore possessed one dorsosacral vertebra and at least four sacral vertebrae. Five sacral vertebrae are present in early diverging ornithischians such as *Lesothosaurus diagnosticus* [33] and *Agilisaurus louderbacki* [92], whereas there are six in *Heterodontosaurus tucki* [107]. By contrast, there are four sacral vertebrae in *Scelidosaurus harrisonii* (NHMUK PV R1111).

### 5.2.4. Caudal vertebrae

Fifty-eight caudal vertebrae are preserved in the holotype, and this probably represents nearly the whole tail. Caudal vertebrae are commonly preserved among specimens of *Scutellosaurus lawleri*, although anterior caudal vertebrae with complete neural arches (e.g. MCZ VPRA-8801: figure 13*l–p*) are rare. The neural arches are fused to the centra from caudal seven posteriorly in MNA.V.175, and the vertebrae decrease in size posteriorly. The anterior caudal centra are either equidimensional or slightly longer anteroposteriorly than they are wide mediolaterally or tall dorsoventrally (figure 13*a–c*, *l–p*). There are facets for articulation with the haemal arches along the ventral edges of the articular surfaces of the centra, and these are especially well developed on the posterior articular surfaces.

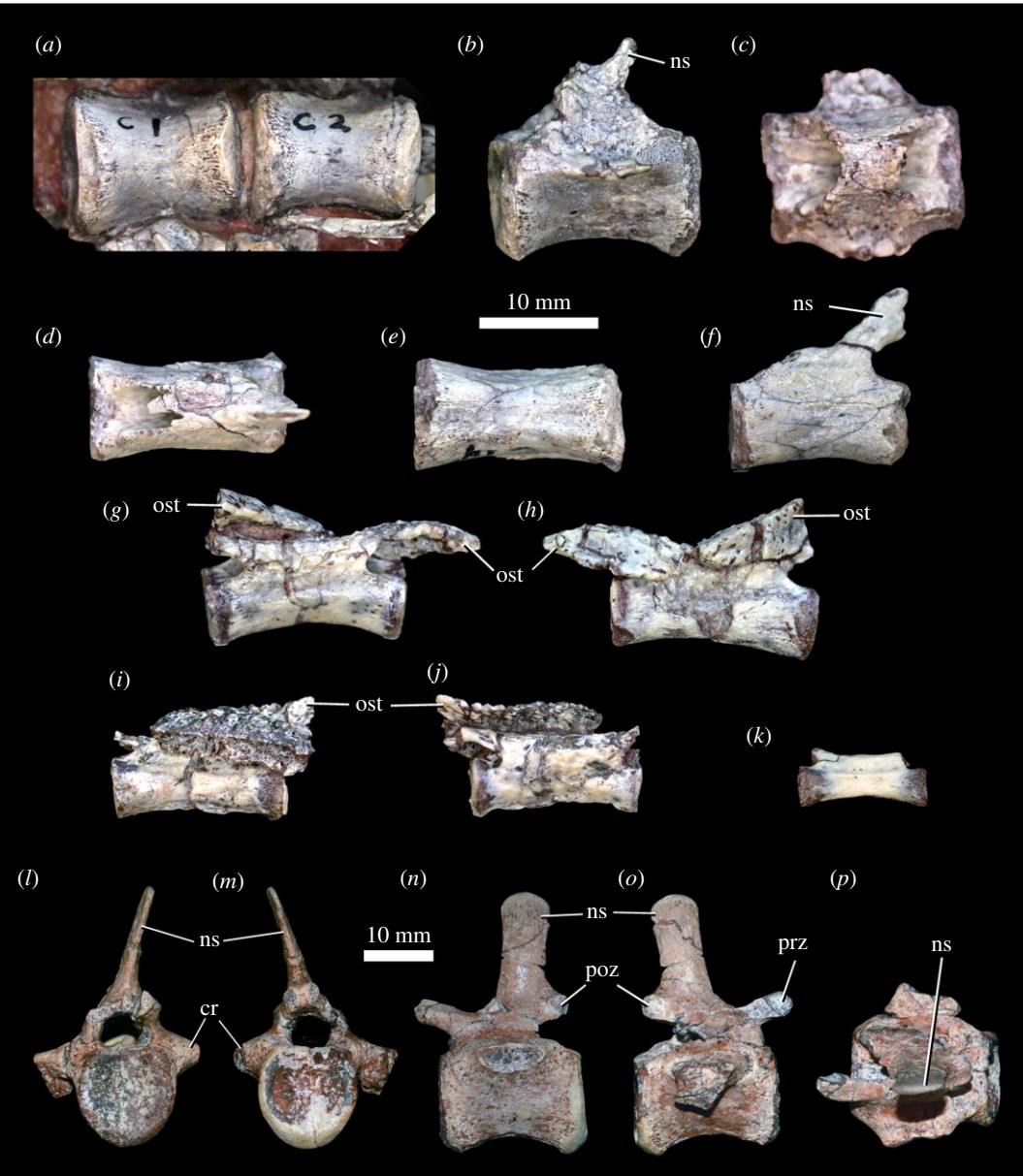

**Figure 13.** *Scutellosaurus lawleri*, caudal vertebrae. (*a–k*) Representative caudal vertebrae of the holotype (MNA.V.175): anterior caudals (labelled caudals 1 and 2) exposed ventrally in a block (*a*); anterior caudal (labelled caudal 10) in left lateral (*b*) and dorsal (*c*) views; posterior caudal (labelled caudal 19) in dorsal (*d*), ventral (*e*) and left lateral (*f*) views; posterior caudal (labelled caudal 30) in left (*g*) and right (*h*) lateral views; caudal 43 in left (*i*) and right (*j*) lateral views; posterior caudal (labelled caudal 50) in left (*k*) lateral view. Caudals 30 and 43 (*g–j*) have osteoderms associated with their neural spines, presumably in close to life position. (*l–p*) Referred anterior caudal vertebra of MCZ VPRA-8801 in anterior (*l*), posterior (*m*), left lateral (*n*), right lateral (*o*) and dorsal (*p*) views. Abbreviations: cr, caudal rib; ns, neural spine; ost, osteoderm; poz, postzygapophysis; prz, prezygapophysis.

Caudal ribs (figure 13, cr) project laterally and are mediolaterally compressed, and these disappear after the 21st caudal vertebra, marking the transition between anterior and posterior caudal vertebrae. The remnants of caudal ribs disappear between caudal vertebrae 22 and 23 in *Scelidosaurus harrisonii* (([11]: fig. 47); NHMUK R1111). The neural spines (figure 13, ns) in the anterior caudal vertebrae of the holotype are generally incomplete, but they are angled strongly posteriorly and appear to lack the bulbous swelling that is present in more derived thyreophorans (e.g. *Scelidosaurus harrisonii*, NHMUK PV R1111; *Dacentrurus armatus*, NHMUK PV OR46013; *Stegosaurus stenops*, YPM 1856; *Hesperosaurus mjosi*, DMNH 29431; *Ankylosaurus magniventris*, AMNH 5895; *Sauropelta edwardsi*, AMNH 3032; *Edmontonia longiceps*, CMN 8531; *Polacanthus foxii*, NHMUK PV R175). The complete neural arch of MCZ VPRA-8801 is taller than the centrum and includes a tall neural spine which is dorsally directed.

The prezygapophyses (figure 13, prz) extend anterodorsally beyond the anterior articular facet of the centrum, whereas the postzygapophyses (figure 13, poz) are shorter and do not extend beyond the posterior articular facet.

In posterior caudal vertebrae (figure 13d–k), the centra are more elongate and are much longer anteroposteriorly than they are wide mediolaterally. After the 28th caudal, chevron facets disappear. Both pre- and postzygapophyses are poorly preserved in the posterior caudal vertebrae but they do not extend substantially beyond the articular facets of the centra.

### 5.2.5. Pectoral girdle

The pectoral girdle is described with the blade of the scapula in a horizontal orientation. The scapula and coracoid are unfused in all known specimens of Scutellosaurus lawleri, as in Lesothosaurus diagnosticus (NHMUK PV RUB17; [33]) and other early diverging thyreophorans (e.g. Scelidosaurus harrisonii, NHMUK PV R1111). The scapulae are nearly complete in the holotype, although the posteriormost portions of the blades are missing. The scapulae of TMM 43663-1 and TMM 43664-1 are more completely preserved; however, in both specimens, the scapulae are mediolaterally compressed taphonomically [20]. The scapula is dorsoventrally expanded at both its anterior and posterior ends with concave dorsal and ventral margins, giving it an asymmetrical hourglass shape in lateral view. The blade of the scapula (figure 14, bl) curves medially, such that the medial surface of the scapular blade is concave along the anteroposterior axis, and the blade flares posteriorly in lateral view. The blade is mediolaterally compressed relative to the proximal plate (figure 14, pp). The proximal plate is small relative to derived thyreophorans such as Stegosaurus stenops (NHMUK PV R36730), and it has a smaller area than the coracoid. The proximal plate is triangular in lateral view, with the anterior margin forming the articular surface for the coracoid, the acromial process (figure 14, ap) extending at approximately 25° from the scapular blade, and a mediolaterally broad ventral process present at the posteroventral margin of the glenoid. The scapula is significantly shorter than the humerus. In MNA.V.175, in which the scapula is incompletely preserved, the scapula length is 63% of the humerus length, and in TMM 43663-1, in which the humerus is incompletely preserved, the scapula length is 70% of the humerus length [20]. This contrasts with some other early ornithischians: for example, the equivalent ratio is 122% in Lesothosaurus diagnosticus (NHMUK PV RUB17; [33]), 117% in Scelidosaurus harrisonii (CAMSM X39256) and 106% in Heterodontosaurus tucki (SAM-PK-K1332; [107]). However, the humerus is also longer than the scapula in Agilisaurus louderbacki (85%; [92]) and Hexinlusaurus consors (78%; [108]).

The holotype preserves both coracoids, which are missing only portions of their dorsal margins. The coracoids are D-shaped elements with a rounded anteroventral surface, a raised posteroventral glenoid surface (figure 14, gl), and no distinct corners. The dorsoventral height is sub-equal to the anteroposterior length. A coracoid foramen (figure 14, cf) is present on the lateral surface near the posterior articular surface for the scapula. There is no anteroventral (sternal) process present and the preserved portion of the ventral margin is rounded in lateral view. The contribution of the coracoid to the glenoid is sub-equal to that of the scapula (figure 14).

### 5.2.6. Humerus

The holotype preserves a complete right humerus (figure 15) and a nearly complete left humerus. The humerus is straight along most of its length in anterior view, but there is a prominent medial projection proximally, the medial tuberosity (figure 15, mt). This projection is less pronounced than in Eocursor parvus [SAM-PK-K8025], but more pronounced than other early ornithischians such as Heterodontosaurus tucki (SAM-PK-K1332) and Lesothosaurus diagnosticus (NHMUK PV RUB17). The proximal articular surface of the humerus is anteroposteriorly compressed and weakly sigmoidal in the proximal view (figure 15c). The proximal articular surface is thickest medially and tapers laterally. The humeral head is not well developed in comparison with Scelidosaurus harrisonii (NHMUK PV R1111), but better developed than in Lesothosaurus diagnosticus (NHMUK PV RUB17). The deltopectoral crest (figure 15, dpc) projects anterolaterally, with an apex occurring at just under 30% of the total shaft length, and distally merges into the shaft just proximal to the midlength of the bone. The deltopectoral crest is a relatively slender protuberance that is triangular in lateral view (figure 15c) with a prominent tubercle present at its apex for the attachment of the m. pectoralis [80]. In Scelidosaurus harrisoni, the deltopectoral crest is considerably larger and projects more strongly anteriorly (CAMSM X39256; [11]). There is no distinct notch between the humeral head and the

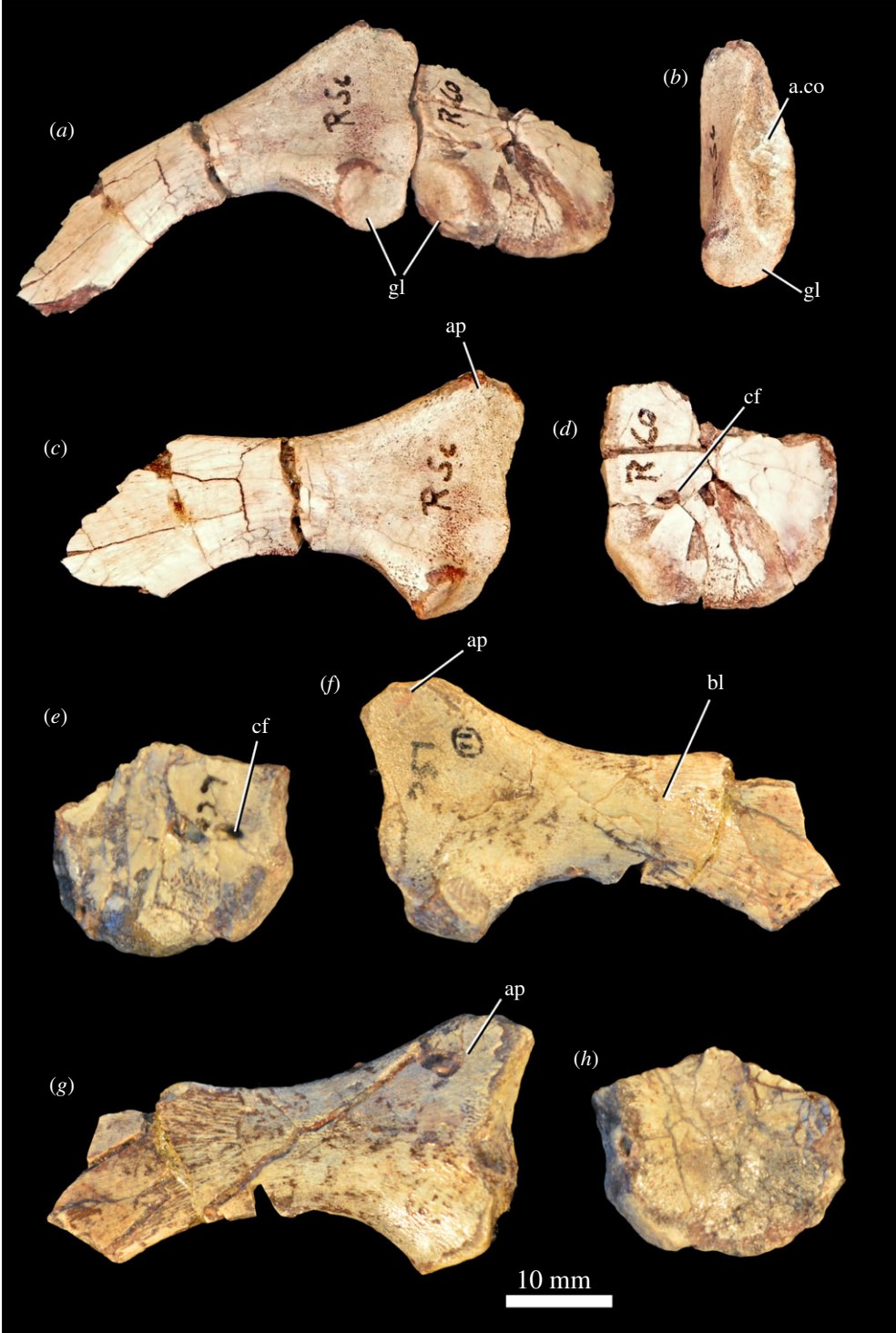

**Figure 14.** *Scutellosaurus lawleri*, holotype (MNA.V.175) scapulae and coracoids. Re-articulated right scapula and coracoid in oblique ventrolateral view (*a*). (*b*,*c*) Right scapula in proximal (*b*) and lateral (*c*) views. Right coracoid in lateral (*d*) view. (*e*,*h*) Left coracoid in lateral (*e*) and medial (*h*) views. (*f*,*g*) Left scapula in lateral (*f*) and medial (*g*) views. Abbreviations: a.co, articulation with the coracoid; ap, acromion process; bl, blade; cf, coracoid foramen; gl, glenoid.

deltopectoral crest or between the humeral head and the medial tuberosity in anterior view. The anterior surface of the proximal end of the humerus is relatively flat between the deltopectoral crest and the medial projection compared with the humerus of *Lesothosaurus diagnosticus* (NHMUK PV RUB17), in which there is a deep anterior depression ([33]: fig. 7A). The shaft of the humerus is ovoid in cross-section, with the long axis oriented mediolaterally. The proximal end of the humerus is sub-equal in transverse width to the distal end. The distal articular surface is figure of eight-shaped in distal view,

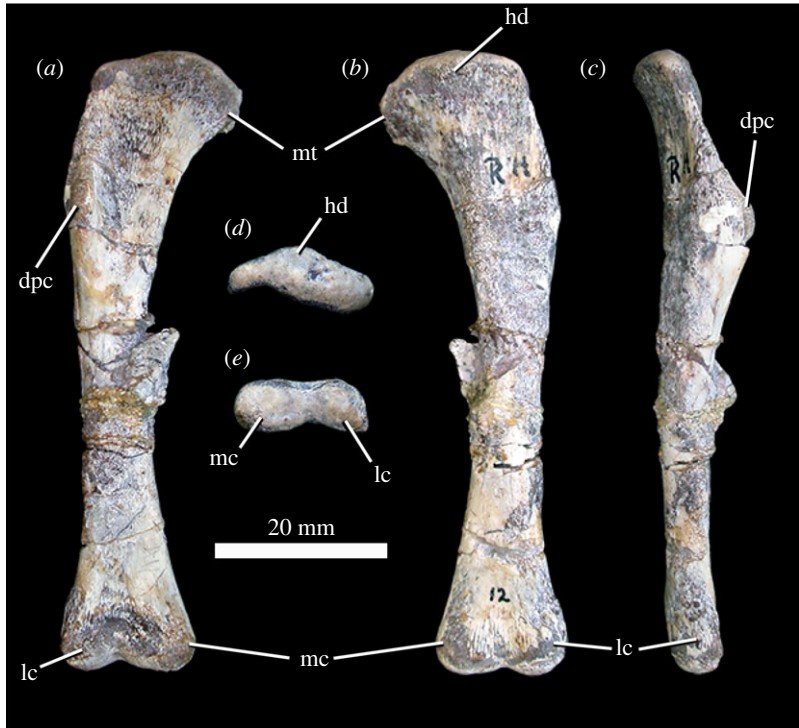

**Figure 15.** *Scutellosaurus lawleri*, holotype (MNA.V.175) right humerus in anterior (*a*), posterior (*b*), lateral (*c*), proximal (*d*) and distal (*e*) views. Abbreviations: dpc, deltopectoral crest; hd, head; lc, lateral condycle; mc, medial condyle; mt, medial tuberosity.

forming a trochlear surface comprising a subovoid lateral (ulnar) condyle (figure 15, lc) and a subquadrangular medial (radial) condyle (figure 15, mc). The medial condyle is slightly wider mediolaterally than the lateral condyle. The anterolateral corner of the lateral condyle forms a point in distal view. The distal end is expanded mediolaterally to approximately twice the width of the midshaft.

### 5.2.7. Radius

Although the holotype preserves fragments of the radius, complete radii are preserved in UCMP 130580, of which the left is better preserved (figure 16*a–d*). The proximal articular surface of the radius is anteroposteriorly compressed with a flat to the concave posterior surface for articulation with the ulna and a convex anterior surface. The shaft is generally straight with a convex anterior margin and is subtriangular in cross-section. There is a well-defined facet on the posterior surface of the distal end of the radius for articulation with the distal end of the ulna. The distal articular surface is convex and is subcircular in distal view. The radius is approximately 60% of the length of the humerus, which is much lower than the 83% estimated by Colbert [18] based on his reconstruction of the radius of MNA.V.175.

### 5.2.8. Ulna

As with the radius, UCMP 130580 includes better-preserved ulnae than the holotype; however, no complete ulna is known (figure 16*e–h*). The ulna is expanded at both the proximal and distal ends in the anterior view. The proximal articular surface of the ulna is anteroposteriorly compressed and subtriangular in the proximal view. The olecranon process (figure 16, ole) is weakly developed, and there is a very weakly developed anterior process. There is a weakly defined articular facet for the radius on the anterior surface of the proximal end of the ulna. The posterior surface of the proximal end of the ulna is generally smooth. The midshaft of the ulna is subovoid in cross-section. The distal end of the ulna is anteroposteriorly compressed and smaller than the proximal end. The anterior surface of the distal end of the ulna is concave to form a facet for the distal end of the radius. The posterior surface of the distal end of the ulna is convex. The distal articular surface is reniform in distal view, with a rounded medial margin and a pointed lateral margin.

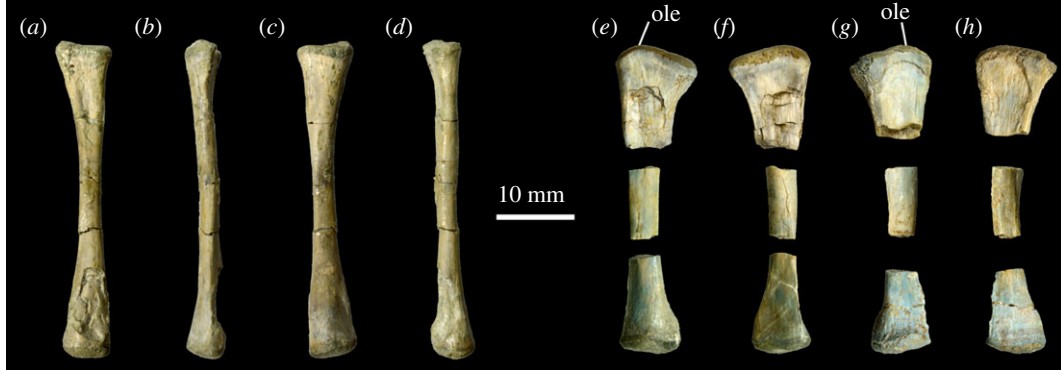

**Figure 16.** *Scutellosaurus lawleri*, referred (UCMP 130580) radius and ulnae. (*a–d*) Left radius in posterior (*a*), lateral (*b*), anterior (*c*) and medial (*d*) views. (*e,f*) Left ulna in posterior (*e*) and anterior (*f*) views. (*g,h*) Right ulna in posterior (*g*) and anterior (*h*) views. Abbreviations: ole, olecranon process.

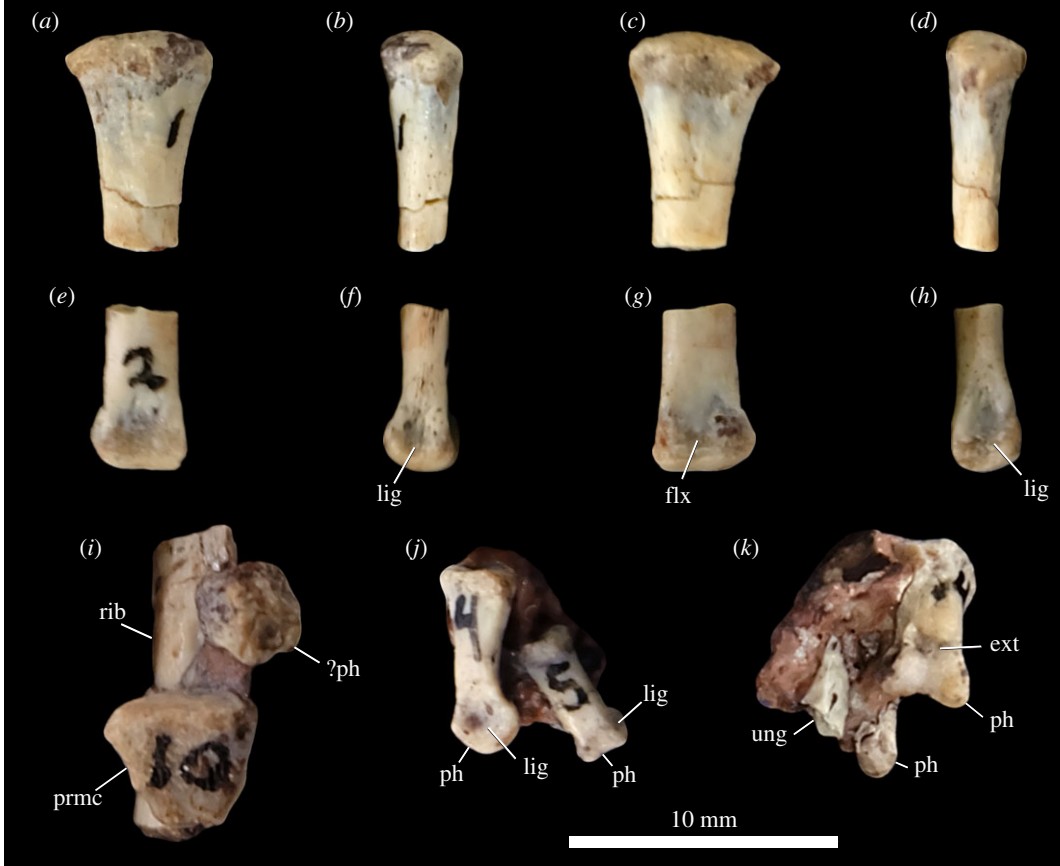

**Figure 17.** *Scutellosaurus lawleri*, holotype (MNA.V.175) manus. (*a–d*) Proximal portion of unknown metacarpal in dorsal (*a*), medial (*b*), ventral (*c*) and lateral (*d*) views; (*e–h*) distal portion of metacarpal in dorsal (*e*), medial (*f*), ventral (*g*) and lateral (*h*) views; (*i*) block containing the proximal portion of an unknown metacarpal exposed in dorsal view; (*j–k*) block containing two manual phalanges and a manual ungual. Abbreviations: ph, possible manual phalanx fragment; ext, extensor pit; flx, flexor pit; lig, ligament pit; ph, manual phalanx; prmc, proximal portion of unknown metacarpal; rib, rib fragment; ung, manual ungual.

### 5.2.9. Manus

The holotype preserves at least two metacarpals (figure 17*a–i*), and one metacarpal is present in UCMP 130580, which was identified by Rosenbaum & Padian [29] as either metacarpal III or IV. The holotype metacarpals were interpreted by Colbert ([18]: fig. 20B) as left metacarpals II and III; however, we cannot confidently confirm these positions. One of the holotype metacarpals is missing its distal end (figure 17*a–d*). The other is missing only a small midshaft portion (figure 17*e–i*), and its proximal end

remains closely appressed to several other bone fragments such that most of it is obscured from view (figure 17*i*). The proximal ends of both metacarpals of the holotype have approximately the same dorsoventral width. The proximal ends of all preserved metacarpals are dorsoventrally expanded relative to their midshafts, as in other early ornithischian dinosaurs (e.g. *Lesothosaurus diagnosticus*, NHMUK PV RUB17; *Scelidosaurus harrisonii*, BRSMG LEGL 0005). The proximal articular surfaces are smooth and flat to gently convex. The dorsal surfaces of the proximal ends of the metacarpals are convex and rounded, whereas the ventral surfaces are nearly flat. All metacarpals have midshafts that are ovoid in cross-section. The distal ends of the metacarpals are dorsoventrally and mediolaterally expanded relative to their shafts and bear a weakly developed flexor pit on their ventral surfaces (figure 17*g*, flx) and well-developed ligament pits on their medial and lateral surfaces (figure 17*f,h*, lig). However, they lack well-developed extensor pits on their dorsal surfaces.

Several manual phalanges are also present in the holotype (e.g. figure 17*j–k*, ph), which were tentatively assigned to digits II and IV in the reconstruction of the manus by Colbert ([18]: fig. 20); however, none of these phalanges was preserved in articulation and their positional identities are uncertain. Non-ungual manual phalanges are dorsoventrally and mediolaterally expanded at both their proximal and distal ends. The proximal articular surfaces are convex with flat ventral margins and round dorsal margins. The distal ends bear prominent flexor and extensor pits (figure 17*k*, ext) on their ventral and dorsal surfaces and well-developed ligament pits (figure 17*j*, lig) on their medial and lateral surfaces. The distal articular surfaces are rounded and spool-shaped. The manual unguals are weakly recurved and taper to a point distally (figure 17*k*, ung). The unguals are triangular in dorsal view and bear parallel grooves along the medial and lateral sides of the dorsal surface like the manual unguals of *Lesothosaurus diagnosticus* ([33]: fig. 9; NHMUK PV RUB17).

Colbert [18] emphasized the relatively large size of the manus of *Scutellosaurus lawleri*, which he suggested may be indicative of facultative bipedalism; however, this claim relies heavily on both his composite reconstruction of the manus based on limited material ([18]: fig. 20B) and his estimated lengths of the radius and ulna ([18]: fig. 20A). As such, the relative size of the manus in *Scutellosaurus lawleri* remains uncertain.

### 5.2.10. Ilium

The right ilium of MNA.V.175 (figures 18 and 19*b*) is incomplete and was erroneously identified as the left ilium by Colbert [18]. It was preserved alongside the partially articulated right hindlimb (figure 18). It is fractured and poorly preserved, however, and little information is available on the morphology of the pubic and ischiadic peduncles, postacetabular process and brevis shelf. Previous reconstructions of these areas are therefore largely speculative (e.g. [18]: fig. 23). The left ilium (figure 19*a*) is preserved together with the left femur and is less complete than the right; it has a triangular outline owing to the loss of the posteromedial and anteromedial parts of the bone. Partial ilia are known from several specimens, including fragments in MNA.V.1752, MCZ VPRA-8792 and UCMP 170829, and portions of both ilia are present in UCMP 130580. A nearly complete right ilium is preserved in TMM 43664-1 [20]; however, this ilium is mediolaterally compressed by taphonomy.

A deep medioventral flange of the ilium (figure 19, mf) partially closes the acetabulum medially, as in many other early diverging ornithischians (*Lesothosaurus diagnosticus*, [34]; *Laquintasaura venezuelae*, MBLUZ P.1443) including the early diverging thyreophoran (*Scelidosaurus harrisonii*, NHMUK PV R1111). A broad lateral expansion (figure 19, le) is present above the acetabulum [18]. This lateral expansion of the dorsal surface of the acetabulum is generally referred to as a 'supraacetabular flange' in early dinosaurs and ornithischians. However, the term 'supraacetabular flange' has also been used to refer to a lateral expansion of the dorsal margin of the ilium above the ischiadic peduncle seen in derived eurypodans, whereas a similar feature in ceratopsians and hadrosaurs has been termed an 'antitrochanter' [2]. As the term 'supraacetabular flange' is used by different authors to refer to non-homologous features within Thyreophora, we avoid using it here to refer to the lateral expansion above the acetabulum of *Scutellosaurus lawleri*. The lateral expansion bounding the acetabulum dorsally is continuous with the lateral edge of the pubic peduncle (figure 19, ppd).

The pubic peduncle is mediolaterally broad and thickest anteriorly, such that the articular surface is subtriangular. In lateral view, the anterodorsal edge of the pubic peduncle is very gently convex, and the ventral articular surface is more strongly convex and rounded. The distal end of the pubic peduncle is swollen and rough, and the articular surface is rugose. The ischial peduncle (figure 19, ispd) is poorly defined relative to some other early ornithischians (e.g. *Eocursor parvus*, SAM-PK-K8025; ([62]: fig. 13A]) but similar to *Scelidosaurus harrisonii* (NHMUK PV R1111; [11]: fig. 68A]). It is a broad swollen

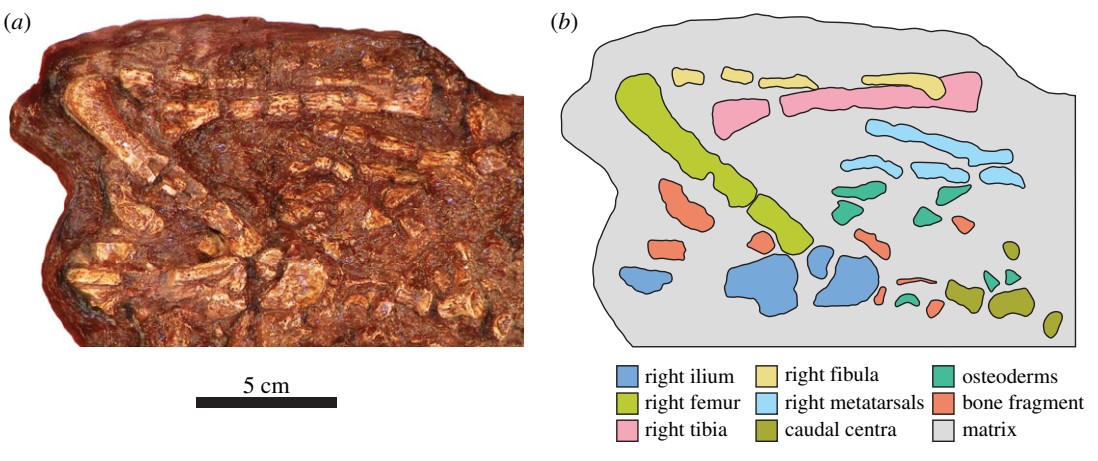

**Figure 18.** *Scutellosaurus lawleri*, preservation of the holotype (MNA.V.175) right hindlimb. Photograph of painted cast of field jacket (*a*) and interpretive line drawing of cast of field jacket (*b*).

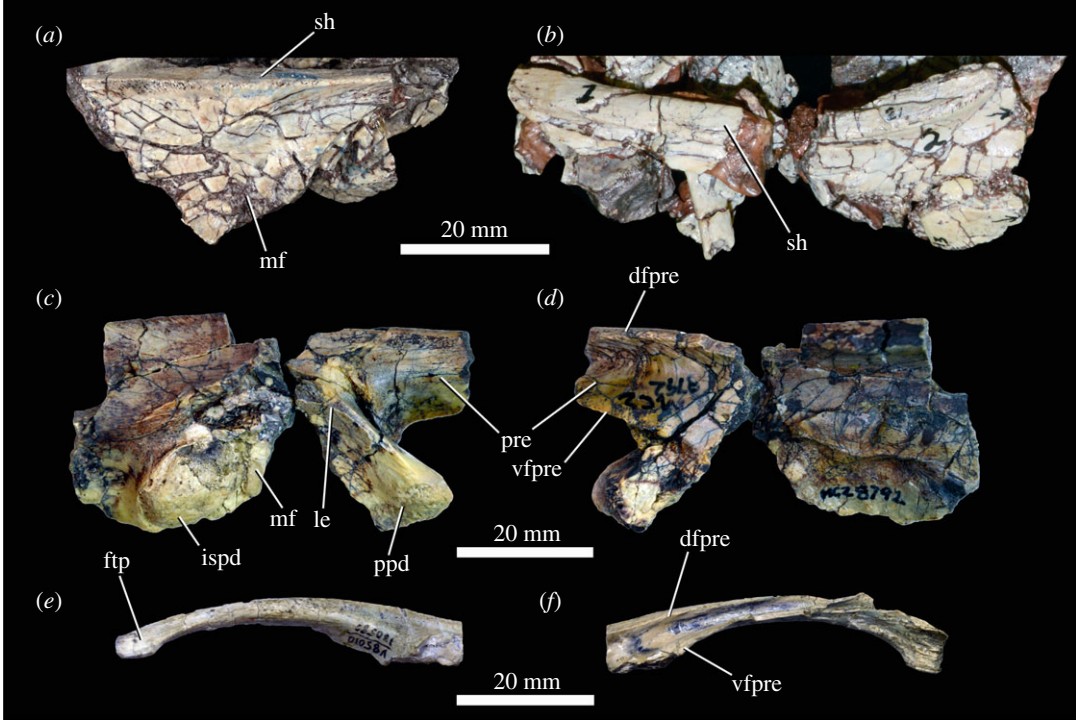

**Figure 19.** *Scutellosaurus lawleri*, ilia. (*a,b*) Holotype (MNA.V.175) left (*a*) and right (*b*) ilia in medial view. (*c–d*) Referred (MCZ VPRA-8792) right ilium in lateral (*c*) and medial (*d*) views. (*e,f*) Referred (UCMP 130580) left ilium, preacetabular process in lateral (*e*) and medial (*f*) views. Abbreviations: dfpre, dorsal flange of the preacetabular process; ftp, flattened tip; ispd, ischial peduncle; le, lateral expansion above the acetabulum; mf, medioventral flange; ppd, pubic peduncle; pre, preacetabular process; sh, shelf; vfpre, ventral flange of the preacetabular process.

surface at the posterior end of the medioventral flange of the acetabulum. The ischial peduncle is anteroposteriorly expanded relative to its mediolateral width.

Rosenbaum & Padian [29] figured, but did not describe in detail, the unusual preacetabular process (figure 19, pre) of the ilium of UCMP 130580. The preacetabular process is elongate and dorsoventrally compressed (figure 19*e–f*). Dorsal and ventral margins of the preacetabular process are drawn out medially into narrow flanges that give it a C-shape in cross-section. Anteriorly the dorsal (figure 19, dfpre) and ventral (figure 19, vfpre) flanges converge upon one another, eventually fusing to form an expanded and dorsolaterally to ventromedially flattened tip (figure 19, ftp). The dorsolateral surface of this expanded tip bears anteroposteriorly extending striations, indicating muscle attachment for the *m. iliotibialis* [80]. The preacetabular process of the ilium of TMM 43664-1 similarly possesses dorsal

and ventral medial flanges, but these converge upon each other anteriorly and fuse to form a more strictly dorsoventrally flattened tip to the preacetabular process, and the process is overall straighter than the bowed process of the ilium of UCMP 130580 [20].

The dorsal margin of the main body of the ilium above the acetabulum is mediolaterally expanded into a narrow shelf (figure 19, sh) that bears prominent vertical striations on its medial surface; however, it is not drawn out medially into a distinct flange, as on the preacetabular process. Mediolateral expansion above the acetabulum in *Scutellosaurus lawleri* exceeds that seen in *Lesothosaurus diagnosticus* (NHMUK PV RU B17) and *Laquintasaura venezuelae* (MBLUZ P.1443), but it is less developed than in other thyreophorans (e.g. *Scelidosaurus harrisonii*, NHMUK PV R1111; stegosaurs [2]; ankylosaurs [95]). The dorsal margin of the ilium above the postacetabular process is not expanded mediolaterally. The postacetabular process is damaged or incomplete in all known ilia, but it appears relatively short and blunt, with a shallow, ventrally facing brevis fossa that is best visualized in TMM 43664-1 ([20]: fig. 13C). One sacral rib impression can be identified (contra [18]) on the partial right ilium of MNA.V.175. This impression is positioned at the base of the preacetabular process and the pubic peduncle; an equivalent rib scar has been identified as the attachment site for the rib of the first true sacral vertebra in *Lesothosaurus diagnosticus* [31],([34]: fig. 9C).

### 5.2.11. Ischium

The holotype preserves the right and left ischia, although both elements are fragmentary. The right ischium, of which the proximal ([18]: fig. 22A; misidentified as the left ischium) and distal ([18]: fig. 22C) ends are preserved, is attached to the pelvic block that also contains the right ilium, right pubis and the proximal end of the right femur, and it is found on the underside of the ilium. The left ischium, which was not figured by Colbert [18] and of which only the distal end is preserved, is found as part of an associated block together with the complete left femur, the left ilium and the left pubis. MCZ VPRA-8801 preserves a nearly complete right ischium (figure 20*a–c*), which is the best-preserved ischium known for *Scutellosaurus lawleri*. UCMP 130580 preserves a nearly complete right ischium (figure 20*d–f*); misidentified as the left ischium by Rosenbaum & Padian ([29]: fig. 3]) and a midshaft portion of the left ischium (figure 20*g,h*).

At the proximal end of the ischium, the pubic (figure 20, ppd) and iliac (figure 20, ilp) peduncles are connected by a mediolaterally compressed and anteriorly concave acetabular region (figure 20, ace). The iliac peduncle of the right ischium of MNA.V.175 and UCMP 130580 partially occludes the acetabulum with an anteriorly extending flange; however, this condition is absent in the right ischium of MCZ VPRA-8801. The ischial shaft is twisted along its length so that the distal end is rotated medially and the distal ischial shaft is mediolaterally expanded; this would have formed an elongate ischial symphysis with the opposing ischium, as in other early diverging ornithischians [31], but in contrast with *Scelidosaurus harrisonii* (NHMUK PV R1111) in which the ischial symphysis was restricted to the distal end of the bone and strong torsion of the shaft is absent. There is a ridge along the dorsal portion of the lateral surface of the shaft (figure 20, lr). Much of the narrow symphysial margin of the ischial shaft is incomplete in the preserved specimens, giving the impression of a discrete, tab-like obturator process ([29]: fig. 2B) and a tab-like obturator process has previously been reconstructed for *Scutellosaurus lawleri* ([18]: fig. 23); however, this is probably an artefact of poor preservation. The right ischium of MCZ VPRA-8801 clearly lacks a prominent obturator process, and the obturator process is not observed on the ischia of other thyreophorans (e.g. *Scelidosaurus harrisonii*, NHMUK PV R1111; stegosaurs [94]; ankylosaurs [95]). A weak groove is present on the dorsal margin of the ischial shaft, and distally passes onto the medial side of the shaft. An identical groove was described for *Lesothosaurus diagnosticus* [34,80,109,110], *Agilisaurus louderbacki* [92] and ornithischian outgroups [111] and probably represents the origin of the *m. adductor femoris* 2 [80].

### 5.2.12. Pubis

The anterior end of the left pubis is present in the holotype (figure 21*a*, pub; [18]: fig. 22B) along with part of the pubic shaft and is preserved underneath the left ilium. We were not able to locate the distal end, despite it being described as present by Colbert [18]. The right pubis, of which only the distal end is preserved ([18]: fig. 22C), is preserved underneath the right pelvic block and next to the distal end of the right ischium. Despite it being described as present by Colbert [18], the anterior end of the right pubis could not be located. No other specimens preserve a pubis.

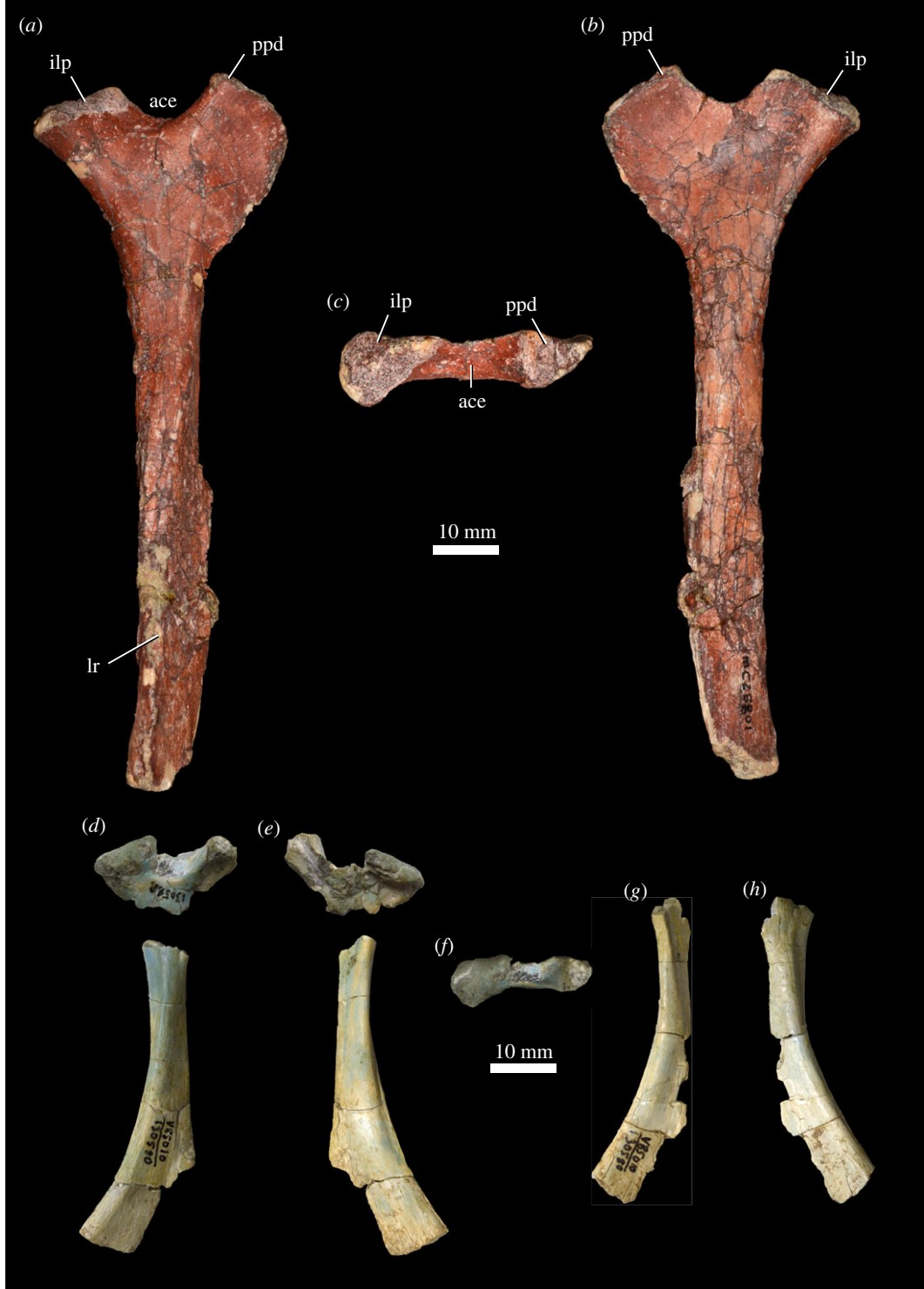

**Figure 20.** *Scutellosaurus lawleri*, referred ischia. (*a–c*) Right ischium of MCZ VPRA-8801 in lateral (*a*), medial (*b*) and proximal (*c*) views. (*d–g*) Right ischium of UCMP 130580 in lateral (*d*), medial (*e*) and proximal (*f*) views. (*g,h*) Shaft of left ischium in lateral (*g*) and medial (*h*) views. Abbreviations: ace, acetabulum; gr, groove; ilp, iliac peduncle; lr, lateral ridge; ppd, pubic peduncle.

There is no evidence to support reconstructions of an elongate prepubis ([18]: fig. 23); the mediolaterally compressed prepubis was probably short as in *Eocursor parvus* [62], *Lesothosaurus diagnosticus* [33,34], *Laquintasaura venezuelae* (MBLUZ P.5008) and *Scelidosaurus harrisonii* (NHMUK PV R1111). The pubic shaft is long, thin and rod-like, with a smaller diameter than the shaft of the ischium, and there does not appear to be a distal expansion, like the condition in *Lesothosaurus diagnosticus* (NHMUK PV RUB17; [33]). At the anterior end, there is a large obturator foramen which

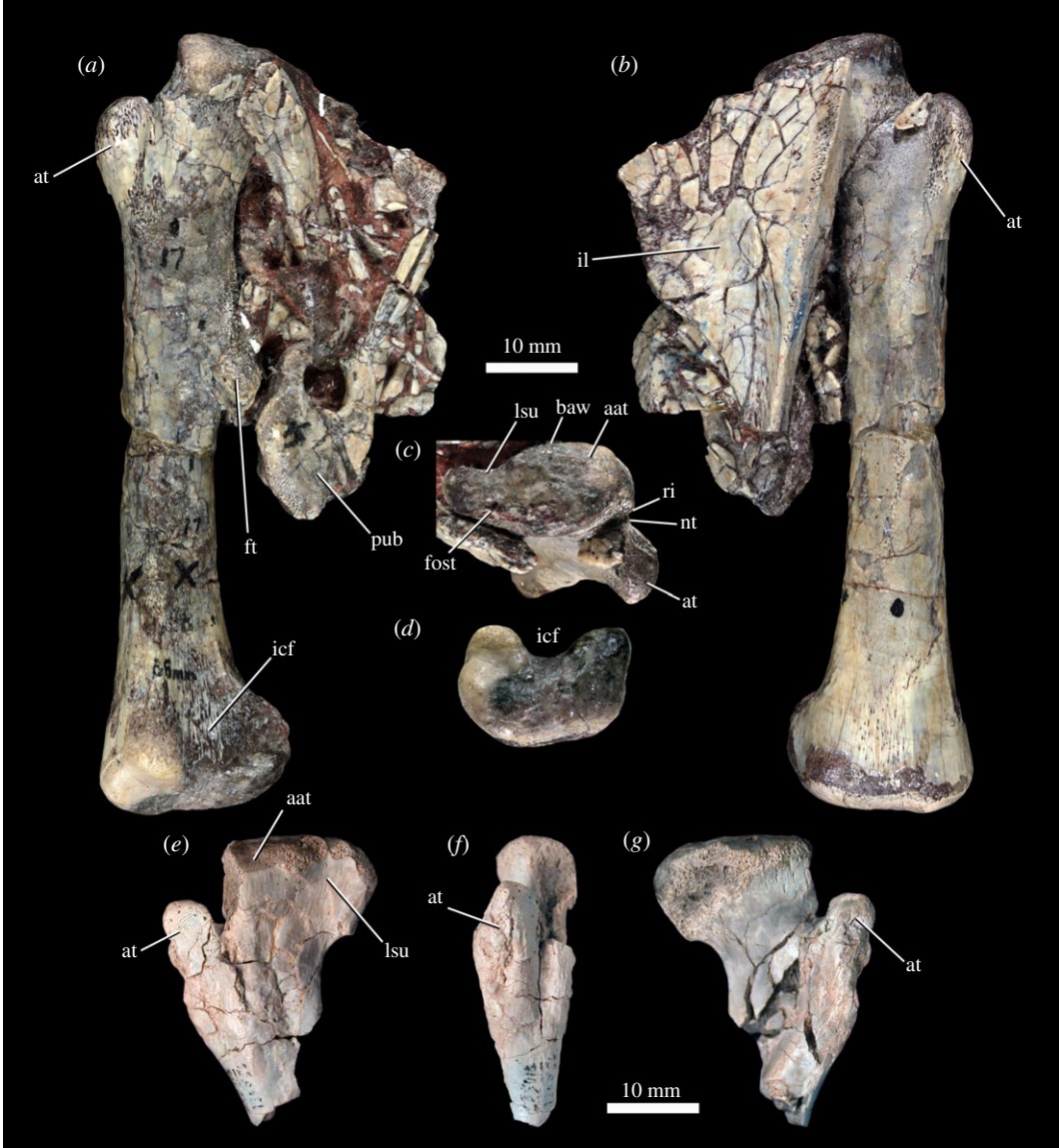

**Figure 21.** *Scutellosaurus lawleri*, femora. (*a*–*d*) Holotype (MNA.V.175) left femur in posterior (*a*), anterior (*b*), proximal (*c*) and distal (*d*) views. (*e*–*g*) Referred (UCMP 130580) left femur, proximal portion in posterior (*e*), lateral (*f*) and anterior (*g*) views. Abbreviations: aat, articularis antitrochanterica; at, anterior trochanter; baw, broadest anteroposterior width; dlt, dorsolateral trochanter; fost, *fossa trochanteris*; ft, fourth trochanter; icf, posterior intercondylar fossa; lsu, ligament groove on the proximal end of the femur; nt, notch separating anterior and dorsolateral trochanters; pub, pubis; r, ridge.

is not closed posteriorly; this is similar to the condition in *Scelidosaurus harrisonii* (NHMUK PV R1111), but contrasts with the condition in some specimens of *Lesothosaurus diagnosticus* (e.g. NHMUK PV RUB17; [33]) and *Laquintasaura venezuelae* (MBLUZ P.5008), in which the obturator foramen is fully enclosed. There is a large facet on the dorsal surface for the acetabulum [18] and at the posterior end of the obturator foramen there is a smaller facet for articulation with the ischium.

### 5.2.13. Femur

Femora are commonly preserved among specimens of *Scutellosaurus lawleri*, and both left (figure 21*a*–*c*) and right (figure 18) femora are nearly complete in the holotype. As noted by Colbert [18], the femur of *Scutellosaurus lawleri* is morphologically like the femora of other early ornithischians such as *Lesothosaurus diagnosticus* [33,34,109], *Laquintasaura venezuelae* (MBLUZ P.5003), and *Scelidosaurus harrisonii* (NHMUK PV R1111). The length of the complete left femur of MNA.V.175 given by Colbert [18] is inaccurate; the correct length is 83.5 mm, and the femur is substantially shorter than the tibia (87% of tibial length), as in

other small-bodied early ornithischians (contra [18]), but it is longer than the humerus (121% of humeral length).

The femur is straight in anterior or posterior view but bowed slightly anteriorly in lateral or medial view. The broadest anteroposterior width of the proximal femoral surface occurs centrally, as in dinosauriforms and early saurischian dinosaurs ([111]: fig. 4C), and in other early ornithischians (*Lesothosaurus diagnosticus* [34]; *Scelidosaurus harrisonii*, NHMUK PV R1111; *Laquintasaura venezuelae*, MBLUZ P.5003), rather than laterally as in most other ornithischians. A weak groove, the *fossa trochanteris*, extends from the anteromedial corner posterolaterally across the proximal surface of the femur, dividing the surface into anterolateral and posteromedial areas. A similar groove is present in most other early dinosaurs [111,112]. This groove is continuous with a distinct depression on the posterolateral corner of the proximal surface (the 'articularis antitrochanterica'), considered to represent the articulation surface for the mediolaterally thickened dorsal margin of the acetabulum of the ilium by Langer [111]. The articularis antitrochanterica is bounded medially by a weak medial tuber (see discussion in [100]). A broad and well-developed ligament sulcus is present medial to the tuberosity.

Many studies have identified the greater trochanter as limited to the posterolateral margin of the proximal femur [18,34]; by contrast, Langer [111] proposed that in early dinosaurs the groove on the proximal surface separates the greater trochanter from the medially facing femoral head. As a result, the greater trochanter occupies the whole of the lateral and anterolateral surface, although there is not a distinct constriction between the greater trochanter and the femoral head. A low proximodistally extending ridge, the dorsolateral trochanter, is present on the lateral surface of the greater trochanter, as in *Scelidosaurus harrisonii* (NHMUK PV R6704) and early saurischian dinosaurs [111,113]. In some other early ornithischians (e.g. *Lesothosaurus diagnosticus* [34,80]; *Laquintasaura venezuelae*, MBLUZ P.5003) the dorsolateral trochanter is extended anteriorly as a distinct flange, and this probably represents the initial stage in the development of the neornithischian femur in which the greater trochanter is anteroposteriorly expanded, limited to the lateral margin of the femur and separated from the femoral head by a distinct constriction.

A distinct cleft separates the dorsolateral trochanter from the large, finger-like anterior trochanter. As in early ornithischians (*Lesothosaurus diagnosticus* [34]; *Scelidosaurus harrisonii*, NHMUK PV R1111), the anterior trochanter is anteroposteriorly broad and positioned rather distally on the shaft relative to the proximal surface of the femoral head.

The attachment of the pendent fourth trochanter (figure 21*a*, ft) is positioned entirely on the proximal half of the femur (contra [28]) and projects ventromedially. A small nutritive foramen is present proximolateral to the base of the trochanter. Distally, a muscle-scar is present on the anterolateral surface of the distal femur, probably for the *m. femorotibialis lateralis* [80]. The angle between the long axis of the femoral head and the long axis of the femoral shaft is roughly 90°. In distal view, the distal end is U-shaped (figure 21*d*), with roughly equal-sized condyles and a large intercondylar fossa, and the fibular epicondyle is not medially inset, unlike the condition in *Laquintasaura venezuelae*, a taxon recovered as sister taxon to *Scutellosaurus lawleri* in some recent phylogenetic analyses [33,110].

### 5.2.14. Tibia

Both tibiae are well preserved in the holotype, and the right tibia (figures 18 and 22) is nearly complete. The tibiae are plesiomorphically longer than the femora, contrary to the condition in *Scelidosaurus harrisonii* (NHMUK PV R1111, [11]) and eurypodans [1,114], but similar to the condition in other early ornithischians (e.g. *Heterodontosaurus tucki* [107]; *Agilisaurus louderbacki* [92]; *Lesothosaurus diagnosticus* [33]). The proximal end of the tibia is expanded anteroposteriorly such that the anteroposterior length is greater than the transverse width; at the distal end, the opposite is true. The proximal articular surface of the tibia comprises a well-developed cnemial crest (figure 22, cnc) and two prominent posteriorly directed condyles. The cnemial crest projects anterolaterally, and the two posterior condyles include the posterolaterally directed fibular condyle (figure 22, fibc) and the larger medial condyle (figure 22, mc). The fibular condyle lies slightly distal to the cnemial crest and the medial condyle. The distal end of the tibia is triangular in cross-section, with a posteriorly directed apex. The anterior surface of the lateral malleolus (figure 22, lm) is flat for articulation with the fibula and is slightly offset anteriorly from the medial malleolus (figure 22, mm), from which it is separated by a ridge. It also extends further ventrally than the medial malleolus but not to the degree seen in *Scelidosaurus harrisonii* (NHMUK PV R1111).

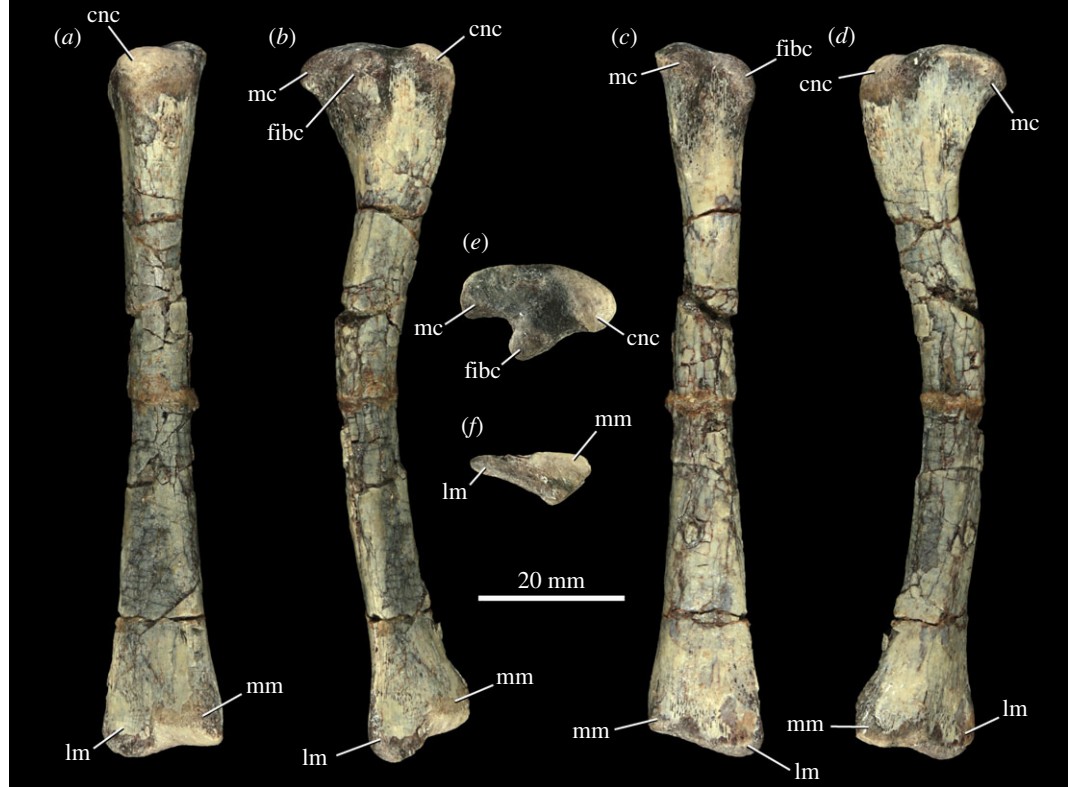

**Figure 22.** *Scutellosaurus lawleri*, holotype (MNA.V.175) right tibia in anterior (*a*), lateral (*b*), posterior (*c*), medial (*d*), proximal (*e*) and distal (*f*) views. Abbreviations: cnc, cnemial crest; fibc, fibular condyle; lm, lateral malleolus; mc, medial condyle; mm, medial malleolus.

### 5.2.15. Fibula

The holotype preserves a nearly complete but badly fragmented left fibula and the proximal and distal ends of the right fibula (figures 18 and 23). The fibula is an elongate, thin element, with an anteroposteriorly and mediolaterally expanded proximal end, although this expansion is not as well developed as that of *Lesothosaurus diagnosticus* (NHMUK PV RUB17; [33]). The proximal end of the fibula is concave medially where it articulates with the tibia (figure 23, a.tib) and convex laterally. The proximal articular surface is smooth, convex and reniform in the proximal view. The shaft does not appear to be bowed anteriorly as it is in *Eocursor parvus* [62], *Lesothosaurus diagnosticus* (NHMUK PV RUB17; [33]) or *Hypsilophodon foxii* [106], although the element is so fragmentary that this could be an artefact of its reconstruction. The distal end of the fibula is both anteroposteriorly and mediolaterally expanded but less so than the proximal end. The anterolateral surface of the distal end is concave and smooth, and there is a flat posterior surface for articulation with the tibia (figure 23, a.tib). The distal articular surface is rough and subquadrangular in the distal view.

### 5.2.16. Tarsus

Both the holotype and paratype include complete and well-preserved astragali. The astragalus (figure 24*a–f*) is subquadrangular in proximal or distal view with a transverse long axis. The proximal surface of the astragalus is smooth and subdivided into two bowl-like concave surfaces for the distal malleoli of the tibia. The medial surface is larger than the lateral surface and would have received the medial malleolus of the tibia (figure 24, a.mm), whereas the distolaterally sloping lateral surface received the medial portion of the lateral malleolus of the tibia (figure 24, a.lm). Novas ([100]: fig. 5D) figured the astragalus of *Scutellosaurus lawleri* in proximal view and misidentified the lateral concave surface as the fibular facet; a well-defined fibular facet is absent from the astragalus of all known ornithischians [27]. The anterior surface of the astragalus is concave in proximal view, with a distinct notch separating the strongly projecting anteromedial corner from the ascending process. In anterior view, the ascending process (figure 24, ap) is low and broad, with the apex positioned laterally; a

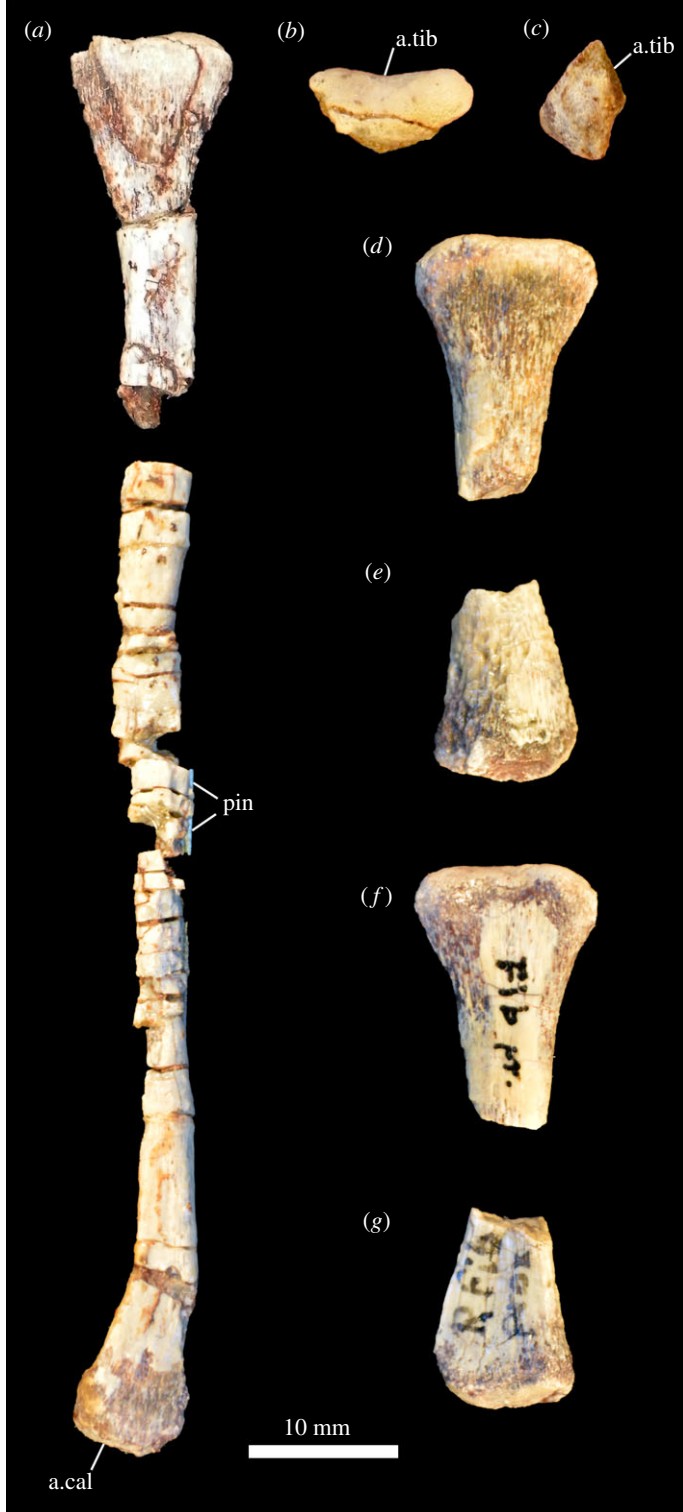

**Figure 23.** *Scutellosaurus lawleri*, holotype (MNA.V.175) fibulae. (*a–c*) Left fibula in lateral (*a*), proximal (*b*) and distal (*c*) views; (*d–g*) right fibula, proximal end in lateral (*d*) and medial (*f*) views, distal end in anterior (*e*) and posterior (*g*) views. Abbreviations: a.cal, articular surface for the calcaneum; a.tib, articular surface for the tibia; pin, metal pin.

small fossa (figure 24, fo) is present on its anterior surface. The lateral surface of the astragalus is concave for articulation with the calcaneum (figure 24, cal). The distal surface of the astragalus is mediolaterally concave and anteroposteriorly convex, forming a trochlear surface.

The calcaneum is partially co-ossified to the astragalus in MNA.V.1752 (figure 24*e,f*) and isolated calcanea are present in several other specimens that represent smaller individuals than MNA.V.1752 (e.g. MNA.V.175, MNA.V.3133, MCZ VPRA-8796). In lateral view, the calcaneum has a rounded distal

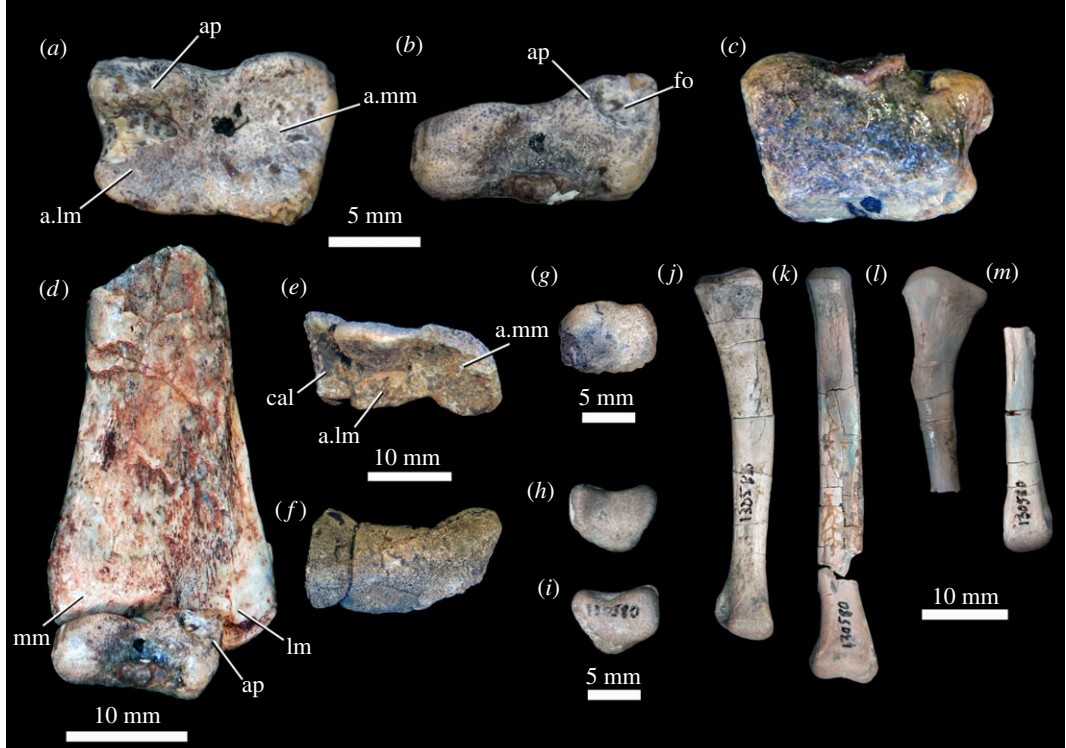

**Figure 24.** *Scutellosaurus lawleri*, tarsus and pes. (*a–c*) Holotype (MNA.V.175) left astragalus in proximal (*a*), anterior (*b*) and distal (*c*) views. (*d*) Holotype (MNA.V.175) left astragalus in articulation with distal end of left tibia in anterior view; (*e,f*), paratype (MNA.V.1752) right astragalus and calcaneum in proximal (*e*) and distal (*f*) views; (*g*) holotype (MNA.V.175) right distal tarsal 3 in proximal view; (*h,i*) referred (UCMP 130580) right distal tarsal 4 in proximal (*h*) and distal (*i*) views; (*j–m*) referred (UCMP 130580) right metatarsus in anterior view, right metatarsal IV (*j*), right metatarsal III (*k*), proximal end of possible metatarsal II (*l*), distal end of metatarsal I (*m*). Abbreviations: a.lm, articular surface for the lateral malleolus of the tibia; a.mm, articular surface for the medial malleolus of the tibia; ap, ascending process; cal, calcaneum; fo, fossa on anterior surface of the ascending process; lm, lateral malleolus; mm, medial malleolus.

margin with one small concave depression (the fibular facet) on the anterior part of the proximal surface to receive the fibula and a second, larger concave depression more posteriorly on the proximal surface to receive the lateral portion of the lateral malleolus of the distal tibia. This larger concave proximal surface is continuous with the concave proximal surfaces of the astragalus, together forming a mediolaterally wide trough-like surface to receive the tibia. The fibular facet is a bowl-shaped concavity that is nearly continuous with the fossa on the anterior portion of the ascending process of the astragalus. The lateral surface of the calcaneum has a pitted, rugose texture.

Colbert [18, p. 31] tentatively noted a 'flattened, somewhat quadrangular element with rounded corners' that might represent a distal tarsal bone in MNA.V.175, which was not figured. We identified an isolated bone matching that description, which we interpret to be the same element (figure 24*g*). Its shape is generally consistent with that of a distal tarsal 3 of *Scelidosaurus harrisonii* figured by Norman ([11]: fig. 87; NHMUK R1111), so we tentatively identify this bone as distal tarsal 3. Distal tarsal 3 is proximodistally thin and wider anteroposteriorly than mediolaterally. The proximal articular surface is slightly concave but nearly flat, whereas the distal articular surface has two shallow concave depressions for articulation with metatarsal III. Distal tarsal 3 is thickest along its lateral margin, which is slightly rough in texture.

An isolated right distal tarsal 4 is preserved in UCMP 130850 (figure 24*h,i*), although this was identified as a distal tarsal 2 by Rosenbaum and Padian ([29]: fig. 5N). Its shape is generally like that of *Scelidosaurus harrisonii* ([11]: fig. 87; NHMUK R1111). The proximal surface is concave, with a convex anterior surface and a concave posterior surface which is wider mediolaterally than the anterior surface. The medial surface is rugose where it contacted distal tarsal 3, and two small facets are present on the anterior and anterolateral surfaces. A concavity is present on the proximal surface of metatarsal IV for articulation with distal tarsal 4; the fact that this concavity does not extend onto the proximal surface of metatarsal III suggests that distal tarsal 4 only distally articulated with metatarsal IV. Distal tarsals are unknown in eurypodan thyreophorans [94,95].

### 5.2.17. Pes

Metatarsals and phalanges are widely preserved among specimens of *Scutellosaurus lawleri*; however, no single specimen preserves a complete pes. There are four metatarsals present in the holotype material. Metatarsals II and IV are missing only their distal ends, whereas metatarsal I only has the distal end preserved, and only the proximal end of metatarsal III is preserved. A partial ungual phalanx of digit I is present, as are phalanges II and III of digit II, phalanges I and IV of digit III, and phalanges I, IV and V of digit IV. No pes material is present in the paratype material, but UCMP 130580 preserves right metatarsals I–IV, of which only III and IV are mostly complete (figure 24*j–m*; mostly identified as left metatarsals by Rosenbaum & Padian [29]); however, these represent the best-preserved associated metatarsus. Rosenbaum & Padian [29] identified the proximal end of a metatarsal as that of the left metatarsal I. However, this identification seems unlikely as the proximal end of this bone is strongly expanded anteroposteriorly; it is possible that this bone is right metatarsal II, but if so, it has undergone post-mortem deformation.

Right metatarsal I has a strongly mediolaterally compressed and splint-like proximal end. The shaft is flattened laterally and would have been closely appressed to metatarsal II along its length. All these features resemble metatarsal I of other early ornithischians, including *Agilisaurus* [92] and *Lesothosaurus diagnosticus* [33,109].

### 5.2.18. Osteoderms

Osteoderms are widely preserved among nearly all specimens of *Scutellosaurus lawleri*. The morphology of the osteoderms of *Scutellosaurus lawleri* was described in detail by Colbert [18] and their histology was studied by Main *et al.* [115]. The holotype of *Scutellosaurus lawleri* contains at least 304 osteoderms, and there are 17 osteoderms in the paratype, although they were found disarticulated in both specimens. Colbert [18] recognized six osteoderm morphotypes but conceding that several of the categories graded into one another other and that the distinctions were subjective. Breeden & Rowe [20] proposed four revised osteoderm morphotypes, including asymmetrical broad, subovoid, flat osteoderms with longitudinal keels that were aligned along the anteroposterior bony axis (Morphotype A; figure 25*a–f*); asymmetrical osteoderms that possess two long sides sloping up to a ridge of varying height and that are deeply concave ventrally, and the basal plate of each osteoderm maintains a fairly uniform thickness (Morphotype B; figure 25*g–j*); symmetrical flat osteoderms that are wider than long, with two longitudinal ridges flanking the midline (Morphotype C; figure 25*k,l*, [31]); and anteroposteriorly long, transversely narrow osteoderms with hollow bases that are approximately as long as the caudal centra (Morphotype D; figures 13*g–j*, 25*m–p*). For a more detailed discussion of these morphotypes and how they compare with other thyreophoran dinosaurs, see Breeden & Rowe [20].

Main *et al.* [115] analysed the histology of osteoderms from several thyreophoran taxa, including *Scutellosaurus lawleri* (UCMP 130580), and concluded that thyreophoran osteoderms are homologous. The histology of the osteoderms of *Scutellosaurus lawleri* suggests that the earliest thyreophoran osteoderms developed from compact dermal bone that was ontogenetically replaced internally.

# 6. Discussion

## 6.1. Ontogeny and growth

Histological thin sections were taken from one radius, one tibia and an osteoderm of UCMP 130580 (an individual somewhat smaller than MNA.V.175) and from one radius and one femur of UCMP 170829 (an individual about 20% larger than UCMP 130580) in order to study the ontogeny of *Scutellosaurus lawleri* by Padian *et al.* ([116]; also [115]). At least, three lines of arrested growth in the right tibia and seven lines of arrested growth in the radius of UCMP 130580 were observed by Padian *et al.* [116], who noted that growth appeared to be ceasing in both bones, which may indicate that UCMP 130580 was nearly fully grown. A slightly larger individual displayed calcified cartilage on the epiphyseal surface of the radius, indicating that the animal was also an adult [116]. Indeed, closure of the neurocentral suture in cervical and dorsal vertebrae is only present in a few of the over 70 known specimens of *Scutellosaurus lawleri*.

Conversely, Tykoski [35] suggested that MNA.V.175 and other specimens of *Scutellosaurus lawleri* were probably juveniles, presumably because of the lack of fusion of the neural arches to the centra of the presacral series of vertebrae. However, although the anterior to posterior sequence of neurocentral

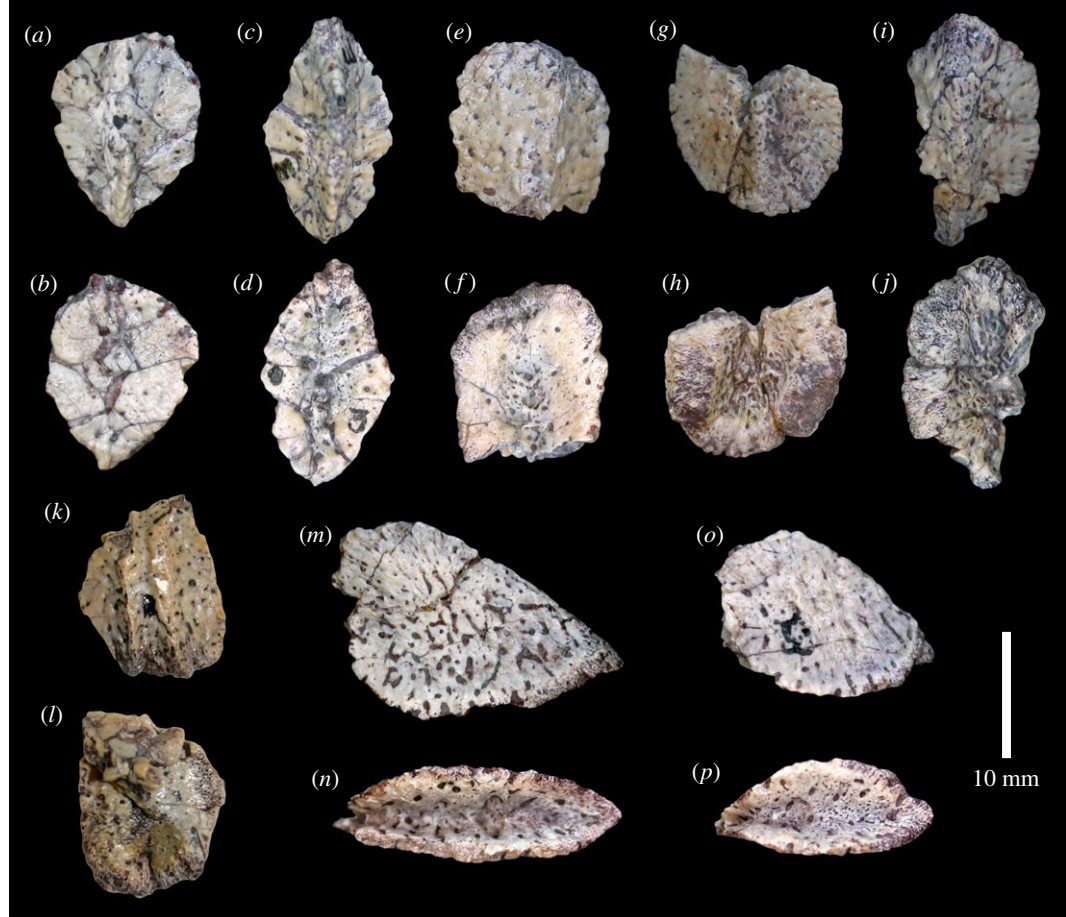

**Figure 25.** *Scutellosaurus lawleri,* holotype (MNA.V.175) osteoderms. (*a–f*) Flat-based osteoderms (Morphotype A) in dorsal (*a,c,e*) and ventral (*b,d,f*) views. (*g–j*) Concave-based osteoderms (Morphotype B) in dorsal (*g,h*) and ventral (*i,j*) views. (*k,l*) Two-keeled osteoderm (Morphotype C) in dorsal (*k*) and ventral (*l*) views. (*m–p*) Hollow-based osteoderms (Morphotype D) in lateral (*m,o*) and ventral (*n,p*) views.

suture closure during ontogeny noted in crocodylians by Brochu [117] is sometimes invoked to interpret the ontogeny of dinosaurs [118–120], its utility as a proxy for skeletal maturity in dinosaurs was called into question by Irmis [121]. Irmis concluded that whereas phytosaurs share the crocodylian state of neurocentral suture closure and that this was plesiomorphic for Pseudosuchia, some dinosaurian taxa (e.g. Ceratopsia, *Camarasaurus* and crown-group Aves) definitely lack the crocodylian pattern of closure and that it was unwise to apply the crocodylian pattern to other archosaurian taxa *a priori*. The neurocentral sutures remain at least partially open in all but the posterior caudal vertebrae in the type and referred specimens of *Scutellosaurus lawleri*, and the lack of neurocentral suture closure is common among early ornithischians (e.g. *Lesothosaurus diagnosticus*, NHMUK PV RUB17; *Stormbergia dangershoeki*, SAM-PK-K1105). However, the relationship between neural arch closure, size and ontogeny in early ornithischian dinosaurs requires further examination.

Histological examination of the limb elements of *Scutellosaurus lawleri* by Padian *et al.* [116] revealed very poorly vascularized and/or lamellar-zonal bone throughout the cortex of all specimens examined. Dinosaurs typically possess more highly vascularized fibrolamellar bone [116,122], which led Padian *et al.* [116] to conclude that *Scutellosaurus lawleri* grew slowly throughout its life as a function of its small body size. However, they also noted that the small-bodied early diverging ornithopod *Orodromeus* possessed bone tissue that was more highly vascularized than that of *Scutellosaurus*. The histology of the small early branching ornithischian *Lesothosaurus diagnosticus* has subsequently been examined [123], and it possesses fibrolamellar bone at early growth stages, indicating a more typical dinosaurian pattern of rapid early growth, followed by a decline in growth rate through ontogeny [116,124,125]. *Lesothosaurus diagnosticus* is variably reconstructed phylogenetically as a non-genasaurian ornithischian [1], an early diverging thyreophoran [3] or an early diverging

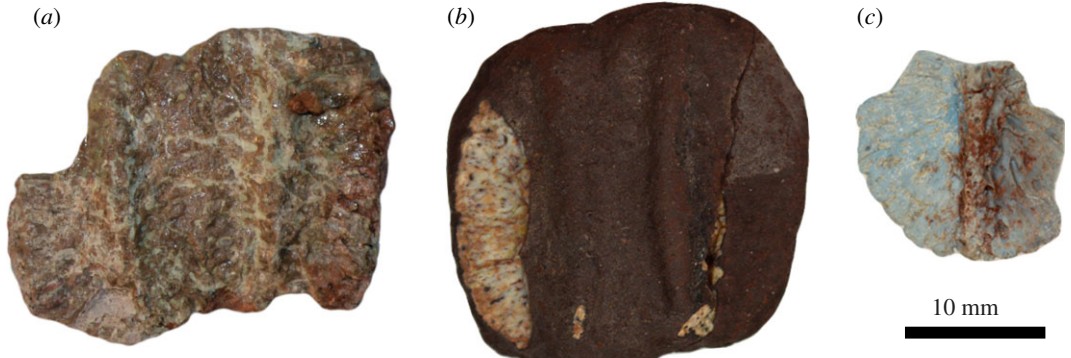

**Figure 26.** *Scutellosaurus lawleri*, referred two-keeled osteoderms (Morphotype C) in dorsal view. (*a*) TMM 43664-1. (*b*) MCZ VPRA-8792. (*c*) MCZ VPRA-8799.

neornithischian [75], but wherever it fits, it is clear that it is a very early branching member of the ornithischian lineage which, like *Scutellosaurus lawleri*, had a small adult body size [34]. The presence of fibrolamellar bone in *Lesothosaurus diagnosticus*, therefore, suggests that lamellar-zonal bone in *Scutellosaurus lawleri* is not necessarily a consequence of either small body size or its phylogenetic position close to the base of Ornithischia, but instead may be unique to either *Scutellosaurus* or to Thyreophora.

The histology of the limb elements of both stegosaurs [124,125] and ankylosaurs [122] has been investigated. Adults of both clades possess poorly vascularized fibrolamellar bone, which becomes avascular and lamellar-zonal in the final stages of growth [122,124,125]. In ankylosaurs, very strong secondary remodelling is observed [122]. The bone tissues of these derived thyreophorans are less well-vascularized than those of other dinosaurs [122,125] and indicate slower growth rates and perhaps lower metabolic rates [125] than in other dinosaurian clades. It seems most likely, therefore, that slow growth rates are a derived characteristic of Thyreophora (also noted in [125]), and the possession of lamellar-zonal bone throughout the cortex may be unique to *Scutellosaurus lawleri* among known ornithischians. Larger-bodied thyreophorans, the ankylosaurs and stegosaurs, may have evolved fibrolamellar bone because of higher growth rates relative to small, early diverging thyreophorans, needed to attain large size (figure 26).

## 6.2. Body proportions and locomotion

Maidment & Barrett [114] included *Scutellosaurus lawleri* in a study of osteological correlates for quadrupedal stance in ornithischian dinosaurs. Of the five anatomical features identified in that study as robust indicators of quadrupedality, *Scutellosaurus lawleri* could only be scored for three, all of which indicate that it was an obligate biped. This conflicts with the interpretation of Colbert [18] of *Scutellosaurus lawleri* as a facultative quadruped based on its limb proportions relative to the obligate biped *Lesothosaurus diagnosticus*. By contrast, Maidment & Barrett [114] interpreted the early thyreophoran *Scelidosaurus harrisonii* as predominantly quadrupedal while retaining some vestiges of its bipedal ancestry, and all eurypodan thyreophorans as obligate quadrupeds. Colbert [18] suggested that the early ancestors of the armoured stegosaurs and ankylosaurs may have been facultative quadrupeds in order to support the additional weight of their bony armour. Maidment *et al*. [86] tested the hypothesis that the additional mass of osteoderms forced the centre of mass to move anteriorly in early thyreophorans, causing them to support their body weight on their forelimbs as well as their hind limbs, and they used *Scutellosaurus lawleri* as a model early thyreophoran. The centre of mass in *Scutellosaurus lawleri* was computed without armour, with its own armour, with the armour of *Stegosaurus* and with the armour of *Euoplocephalus*. The addition of dermal armour to *Scutellosaurus* caused the centre of mass to move slightly posteriorly in all cases, because the centre of mass of the armour alone is posterior to that of the body without armour. The addition of dermal armour did not, therefore, cause early thyreophorans to require forelimb support for their body weight, and Colbert's [18] hypothesis that the weight of dermal armour resulted in quadrupedality can be rejected. Among non-eurypodan thyreophorans, *Scelidosaurus harrisonii* was definitely quadrupedal [126], but the postcranial skeleton of the only specimen of *Emausaurus ernsti* is too

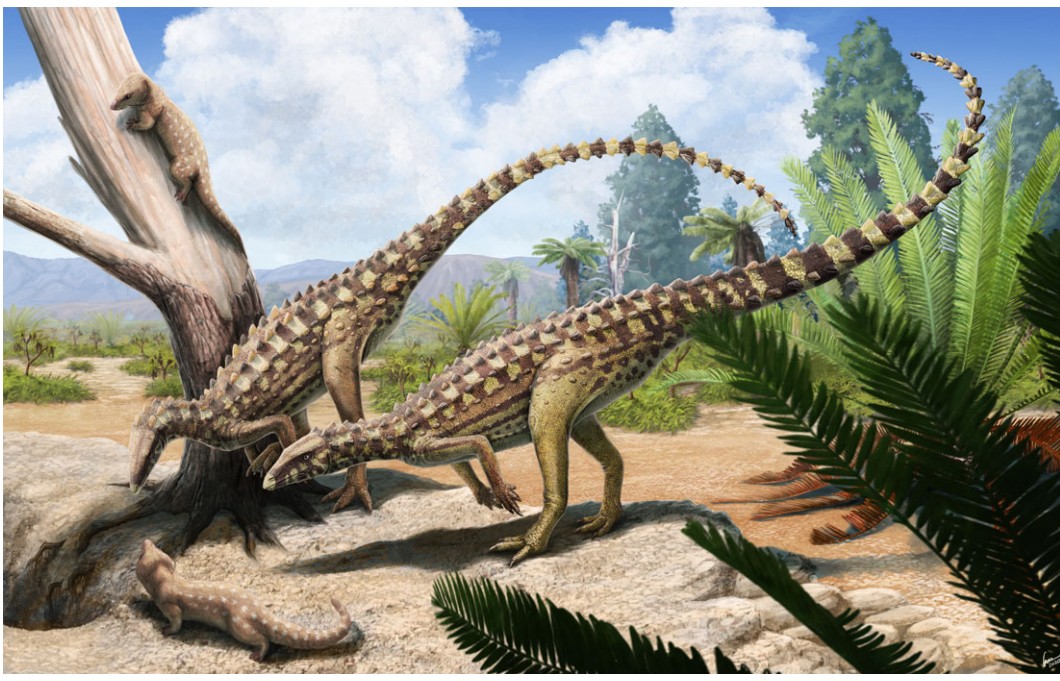

**Figure 27.** Life reconstruction of the thyreophoran ornithischian dinosaur *Scutellosaurus lawleri* from the Lower Jurassic Kayenta Formation rendered as an obligate biped with a speculative osteoderm arrangement. Artwork used with permission by Gabriel Ugueto, who retains the copyright (gabrielugueto.com).

incomplete to infer a mode of locomotion. Current evidence suggests that *Scutellosaurus lawleri* was the only bipedal member of Thyreophora (figure 27), and that more derived members of the clade acquired quadrupedality for reasons other than the acquisition of hypertrophied dermal armour.

## 6.3. Phylogenetic position and character evolution

*Scutellosaurus lawleri* was originally described as a member of Fabrosauridae by Colbert [18], although it was suggested it could be ancestral to ankylosaurs and stegosaurs. Fabrosauridae is now considered polyphyletic [1,3,5,26,31,34], and *Scutellosaurus lawleri* is generally considered an early diverging member of Thyreophora [1–5,20,25–33]. This phylogenetic position is supported by numerous aspects of morphology and subsequently numerous phylogenetic characters. In the ornithischian phylogeny of Butler *et al.* [1], *Scutellosaurus lawleri* is scored as containing four of the six thyreophoran synapomorphies, including characters 46 (state 1, the absence of a forked posterior ramus of the jugal), 101 (state 0, an absent or weak coronoid process), 106 (state 1, the presence of a strong, anteroposteriorly extended ridge on the lateral surface of the surangular) and 112 (state 0, the presence of six premaxillary teeth). Additionally, *Scutellosaurus lawleri* is also scored as possessing the synapomorphy of the clade Thyreophora excluding *Lesothosaurus diagnosticus*, character 89 (state 1, the presence of cortical remodelling on the surface of the skull bones). Similarly, in the ornithischian phylogeny of Boyd [3], *Scutellosaurus lawleri* is scored as possessing the five thyreophoran synapomorphies identified in the analysis, including characters 52 (state 0, the presence of horizontally oriented distal condyles of the quadrates), 86 (state 1, the presence of a strong, anteroposteriorly extended ridge on the lateral surface of the surangular), 112 (state 0, the presence of six premaxillary teeth), 122 (state 0, a concave lingual surface of the maxillary teeth) and 131 (state 0, maxillary teeth positioned near the lateral margin). *Scutellosaurus lawleri* is, therefore, unequivocally a thyreophoran. Within Thyreophora, *Scutellosaurus lawleri* is also excluded from Thyreophoroidea (Eurypoda + *Scelidosaurus harrisonii*) by lacking numerous morphological features. In the ornithischian phylogeny of Butler *et al.* [1], *Scutellosaurus lawleri* contains none of the 16 synapomorphies of Thyreophoroidea identified, and in the ornithischian phylogeny of Boyd [3], *Scutellosaurus lawleri* contains none of the two synapomorphies of Thyreophoroidea identified. Therefore, there is strong evidence for a phylogenetic placement of *Scutellosaurus lawleri* within Thyreophora but outside of Thyreophoroidea, alongside the fragmentary taxon *Emausaurus ernsti*.

## 6.4. Ornithischian dinosaur diversity in the Kayenta Formation

To date, three ornithischian taxa have been identified from the Kayenta Formation: *Scutellosaurus lawleri*, an unnamed larger thyreophoran taxon known from rare osteoderms and postcranial elements [18,59], and an undescribed heterodontosaurid known from a single partial skeleton [26,63,75]. Of these taxa, *Scutellosaurus lawleri* is overwhelmingly the most abundant, with approximately 80 specimens known. Indeed, *Scutellosaurus lawleri* is the most common tetrapod recovered from the Kayenta Formation and is much more common than remains of theropod [127,128] and sauropodomorph [38] dinosaurs. This represents an unusual situation for Early Jurassic dinosaur faunal assemblages: in the well-sampled upper Elliot Formation of South Africa ornithischian remains are relatively abundant but considerably less common than those of sauropodomorphs [129], whereas ornithischian fossils are exceptionally scarce in the Lufeng Formation of China in comparison with sauropodomorphs [16,17,130]. Other Lower Jurassic formations that have yielded ornithischian dinosaur material are generally not sufficiently well sampled to assess relative abundances of ornithischians versus other dinosaurs [10–12,14,104,126]. Unambiguous members of Thyreophora from the Early Jurassic have only been identified from the northern palaeo-hemisphere to date, from the USA (*Scutellosaurus lawleri*) and Europe (*Scelidosaurus harrisonii*, *Emausaurus ernsti*), as well as probably from China ('*Bienosaurus lufengensis*', '*Tatisaurus oehleri*'). The possible thyreophoran affinities of *Lesothosaurus diagnosticus* (southern Africa) and *Laquintasaura venezuelae* (Venezuela) remain unclear. By contrast, the only Early Jurassic heterodontosaurid known to date from the northern palaeo-hemisphere is the single undescribed Kayenta specimen, whereas heterodontosaurids are moderately abundant (*ca* 20 known specimens) in the upper Elliot Formation of southern Africa [63,76,131]. These patterns suggest that there was considerable spatial and/or environmental variation in Early Jurassic ornithischian faunas globally, the nature of which warrants further study.

Ethics. We describe fossil specimens collected from the lands of the Navajo Nation, which require the written permission from the Navajo Nation Minerals Department prior to publication. We sent a draft of our manuscript to the Navajo Nation Minerals Department for approval to publish on 14 August 2020, and we received approval in writing dated 17 August 2020.

Data accessibility. A detailed inventory of referred specimens is available as electronic supplementary material.

Authors' contributions. B.T.B., R.J.B. and S.C.R.M. conceived of the study; B.T.B., T.J.R., R.J.B. and S.C.R.M. wrote the primary draft of the manuscript and assembled the figures. All authors edited and approved the final version of the manuscript.

Competing interests. We declare we have no competing interests.

Funding. B.T.B. was funded for this work by the Ernest L. and Judith W. Lundelius Endowment in Vertebrate Paleontology and the Jackson School of Geosciences at The University of Texas at Austin and the Doris O. and Samuel P. Welles Research Fund at the University of California Museum of Paleontology; T.J.R. was funded by a University of Brighton Science Scholarship; R.J.B. was funded by a NERC PhD studentship during the early stages of this work; T.B.R. was funded by National Science Foundation grants EAR 1258878 and IIS-9874781; and S.C.R.M. was funded by a University of Cambridge Domestic Research Studentship during the early stages of this work.

Acknowledgements. First and foremost, we acknowledge the people of the Navajo Nation for facilitating continued palaeontological research on their land. The specimens described herein were collected by the MNA, UCMP, and MCZ between the years of 1971 and 1983 during fieldwork on the lands of the Navajo Nation under permits issued to MNA. We thank Akhtar Zaman, Bradley Nesemeier, Richard Carlton and Rowena Cheromiah of the Navajo Nation Minerals Department for their assistance and support of this research. Any persons wishing to conduct geologic investigations on the Navajo Nation must first apply for and receive a permit from the Navajo Nation Minerals Department, P.O. Box 1910, Window Rock, Arizona 86515 and phone number +1 (928) 871-6588.

Thanks to David Gillette and Janet Gillette (MNA); Mark Goodwin, Patricia Holroyd and Kevin Padian (UCMP); Christina Byrd, Jessica Cundiff and Stephanie Pierce (MCZ); and Matthew Brown and Christopher Sagebiel (TMM) for providing access to specimens, locality data, historical archives, photographs, and correspondence in their respective museum collections. We acknowledge and thank William Amaral, James Clark, Emily CoBabe, William Downs, Farish Jenkins, David Lawler, Charles Schaff, B. Schubert, Kathleen Smith and Hans-Dieter Sues for discovering the specimens described herein. We also acknowledge and thank William Downs, Ann Johnson, Randy Johnson, David Lawler, Larkin McCormack, Nova Young and any other preparators whose hard work went undocumented for the preparation of the specimens described herein. Joao Vasco-Leite provided photographs of the holotype manus used in figure 17, and Matthew Brown provided the photograph of TMM 43647-11 used in figure 7*e*. Thanks to Gabriel Ugueto for giving permission to use his artwork. We thank Matthew Baron, Paul Barrett, Shannon Barrios, Christopher Bell, Tylor Birthisel, Laura Brenskelle, Matthew Brown, James Clark, Lauren English, Christopher Griffin, Randall Irmis, Joshua Lively, Adam Marsh, Keegan Melstrom, Sterling Nesbitt, David Norman, Kevin Padian, Hans-Dieter Sues, Ronald Tykoski and Zackery Wistort for helpful discussions and advice. We thank the editors, Julia Desojo and Kevin Padian, and three anonymous reviewers for comments that improved this paper.

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
