## [Peer Review File · Royal Society Open Science]

Review History

RSOS-201676.R0 (Original submission)

Review form: Reviewer 1 (Marcos Becerra)

Is the manuscript scientifically sound in its present form?

Yes

Are the interpretations and conclusions justified by the results?

Yes

Is the language acceptable?

Yes

Do you have any ethical concerns with this paper?

No

Have you any concerns about statistical analyses in this paper?

No

Recommendation?

Accept as is

Comments to the Author(s)

No comments are needed. The manuscript is well-written and I enjoyed reading it. It is informative and the bibliographic revision is comprehensive and exhaustive. The actualization of the description of Colbert was needed, and I'm happy for being selected to do it first hand.

Review form: Reviewer 2

Is the manuscript scientifically sound in its present form?

No

Are the interpretations and conclusions justified by the results?

Yes

Is the language acceptable?

Yes

Do you have any ethical concerns with this paper?

No

Have you any concerns about statistical analyses in this paper?

No

Recommendation?

Major revision is needed (please make suggestions in comments)

Comments to the Author(s)

This study provided a detailed description of the early armoured dinosaur *Scutellosaurus lawleri* from the Kayenta Formation of Arizona. This will help for improving the early evolution of ornithischians. Both the description and figures are good. But there are still some things that need to make it clear before publication. Here are my comments as follows:

1. I notice that the authors have already provided a detailed description of *Scutellosaurus* published in JVP recently, although the latter only contains specimens from TMM. This MS cited many times of that paper, and I think it is a little redundant. So I believe it is more appropriate that this paper is a complementary description for *Scutellosaurus* that no need to describe all the bones, but just contains new data and comparisons. This will make the paper more concise.
2. The authors mainly compared *Scutellosaurus* with basal thyreophorans. I hope the author adds more comparison with other ornithischians that have similar size or feature, such as *Eocursor*, *Agilisaurus*, *Hexinlusaurus*, *Jeholosaurus*, and also some primitive stegosaurs including *Huayangosaurus*. *Isaberrysaura* was original considered to a basal neornithischian, but re-put basal stegosaur in Han et al (2018). It should also be added for comparison. In addition, I notice that the dentary of *Scutellosaurus* is quite long and shallow dorsoventrally. The ratio of length to depth is also worth comparing with other ornithischians.

3. In the comparison, I notice many other specimens do not have references. Do you check this by yourself or from references? If you check by yourself, please mention that in the "material and method" part.

4. Page 48. The authors state that among ornithischians only Scutellosaurus have lamellar-zonal bone. This seems unlikely. In Padian (2004), we can see that Scutellosaurus still has abundant vascular canals in the inner region of a tibia (Padian 2004, Figure 3D), unlike the condition in modern crocodiles and lizards, but is similar to Jeholosaurus (Han et al 2020, bone histology of Jeholosaurus). Padian (2004) did mention that a radius shows lamellar-zonal pattern (Fig. 3A), but this is not the femur, tibia or humerus that we usually sampled. In fact, limb bones can show different bone patterns in different parts, the fibulae and ribs usually have less vascular canals that show parallel-fibered or lamellar bone.

5. As the author provided new features of Scutellosaurus, a new phylogenetic analysis is necessary. I find that the authors have provided phylogenetic analysis in another paper (JVP online), but they did not mention it here. Furthermore, I strongly suggest that they can do a new phylogenetic analysis of basal ornithischians based on newly published data, such as Boyd 2015 or Han et al 2018, and add some new related taxa. The matrix of Butler et al (2010) has already changed more or less.

I have more comments in a marked version of the pdf (see Appendix A). Thank you.

Review form: Reviewer 3

Is the manuscript scientifically sound in its present form?

Yes

Are the interpretations and conclusions justified by the results?

Yes

Is the language acceptable?

Yes

Do you have any ethical concerns with this paper?

No

Have you any concerns about statistical analyses in this paper?

No

Recommendation?

Accept with minor revision (please list in comments)

Comments to the Author(s)

This is a well-written and highly detailed anatomical description of the key early dinosaur Scutellosaurus, and represents a very valuable addition to the literature that will no doubt be well-cited in the future. In its position at the base of the diverse and successful armoured dinosaur radiation, Scutellosaurus is a key taxon and the data contained in this manuscript will be invaluable for future research in taxonomy, cladistics, evolutionary dynamics and palaeobiology.

I have only a very few suggested changes for this manuscript prior to publication:

1. For several elements - the nasal, postorbital, squamosal and others, see annotated PDF - it is acknowledged in the text that better preserved and more complete elements exist for some specimens and indeed the text description is based on these best preserved examples. Yet they are not figured; instead, more poorly preserved specimens are figured. Why not figure the very best examples for each bone on which verbal descriptions are based?
2. I do not think Figure 17 is ever cited in the main text and it does not appear between references to Figures 16 and 18.
3. The authors make the very interesting observation on Page 50, Line 24 that different Early Jurassic localities featured different dinosaur faunas, in terms of composition of herbivores (sauropodomorphs, heterodontosaurids, thyreophorans). Any speculation on what was driving these variations?
4. Some minor errors in the references cited (see PDF), please check these carefully prior to submission.
5. Also a few, very minor errors with figure captions, particularly abbreviations. Again, please check carefully. Depending on the final size of the figure, you may want to consider increasing the size of labels on the figures (abbreviations) as they are currently very small and can be difficult to discern. It would be very useful for Figure 8 to indicate which directions are anterior and posterior.

Lastly, there are some minor grammatical errors and a few places where the meaning of sentences could be clarified - I have made suggested changes in the attached annotated PDF (see Appendix B).

Overall, a major piece of work and again a very valuable addition to the literature, superb effort by the authors on this!

Decision letter (RSOS-201676.R0)

Dear Mr Breeden,

The Editors assigned to your paper RSOS-201676 "The anatomy and palaeobiology of the early armoured dinosaur *Scutellosaurus lawleri* from the Kayenta Formation (Lower Jurassic) of Arizona" have now received comments from reviewers and would like you to revise the paper in accordance with the reviewer comments and any comments from the Editors. Please note this decision does not guarantee eventual acceptance.

We do not generally allow multiple rounds of revision so we urge you to make every effort to fully address all of the comments at this stage. If deemed necessary by the Editors, your

manuscript will be sent back to one or more of the original reviewers for assessment. If the original reviewers are not available, we may invite new reviewers.

Please submit your revised manuscript and required files (see below) no later than 21 days from today's (ie 19-Feb-2021) date. Note: the ScholarOne system will 'lock' if submission of the revision is attempted 21 or more days after the deadline. If you do not think you will be able to meet this deadline please contact the editorial office immediately.

on behalf of Dr Julia Brenda Desojo (Associate Editor) and Kevin Padian (Subject Editor)
openscience@royalsociety.org

Subject Editor comments to Author:

The reviewers are happy to see this paper, and are generally enthusiastic about its publication. One raises what should be carefully considered so as to dispel any doubts about "shingling": it would be important to show how this paper differs from the JVP one, and to repeat only necessary details (although this is sometimes difficult). A new phylogeny is probably needed, but this should not be difficult. A wider inclusion of taxa is also recommended. The other reviewer asks why you are figuring sub-optimal specimens rather than better ones, and if this because the better specimens were already figured in the JVP paper please make that clear. Finally, some clarity about the histological features may be needed. It is common in virtually all non-avian dinosaurs that fibro-lamellar bone should be expressed at least in early growth and a transition made either to FL tissue that is less vascularized OR to lamellar-zonal bone before skeletal growth effectively ceases and the EFS is deposited. Smaller taxa tend to grow more slowly, but of course any given section represents its ontogenetic age, so perhaps you could clarify? I don't mean at all to argue with your position, only to note that the outgroups to Dinosauria already grew very quickly and featured copious FL bone, yet they were not larger than *Scutellosaurus*. So the hypothesis that the larger, later Thyreophora evolved (not "developed," please) FL bone as a strategy for more rapid growth to larger size could be called into question. I will be interested in your perspective on this.

Also, I noted that on line 23 of your MS page 46, you have an extra "closure"

Overall, very well done and I look forward to your revisions. Because this is a large MS you may need more time to revise than our standard three weeks, and if so please just inform the office. Thanks for submitting this.

Reviewer comments to Author:

Reviewer: 1

Comments to the Author(s)

No comments are needed. The manuscript is well-written and I enjoyed reading it. It is informative and the bibliographic revision is comprehensive and exhaustive. The actualization of the description of Colbert was needed, and I'm happy for being selected to do it first hand.

Reviewer: 2

Comments to the Author(s)

This study provided a detailed description of the early armoured dinosaur *Scutellosaurus lawleri* from the Kayenta Formation of Arizona. This will help for improving the early evolution of ornithischians. Both the description and figures are good. But there are still some things that need to make it clear before publication. Here are my comments as follows:

1. I notice that the authors have already provided a detailed description of *Scutellosaurus* published in JVP recently, although the latter only contains specimens from TMM. This MS cited many times of that paper, and I think it is a little redundant. So I believe it is more appropriate that this paper is a complementary description for *Scutellosaurus* that no need to describe all the bones, but just contains new data and comparisons. This will make the paper more concise.
2. The authors mainly compared *Scutellosaurus* with basal thyreophorans. I hope the author adds more comparison with other ornithischians that have similar size or feature, such as *Eocursor*, *Agilisaurus*, *Hexinlusaurus*, *Jeholosaurus*, and also some primitive stegosaurs including *Huayangosaurus*. *Isaberrysaura* was original considered to a basal neornithischian, but re-put basal stegosaur in Han et al (2018). It should also be added for comparison. In addition, I notice that the dentary of *Scutellosaurus* is quite long and shallow dorsoventrally. The ratio of length to depth is also worth comparing with other ornithischians.
3. In the comparison, I notice many other specimens do not have references. Do you check this by yourself or from references? If you check by yourself, please mention that in the "material and method" part.
4. Page 48. The authors state that among ornithischians only *Scutellosaurus* have lamellar-zonal bone. This seems unlikely. In Padian (2004), we can see that *Scutellosaurus* still has abundant vascular canals in the inner region of a tibia (Padian 2004, Figure 3D), unlike the condition in modern crocodiles and lizards, but is similar to *Jeholosaurus* (Han et al 2020, bone histology of *Jeholosaurus*). Padian (2004) did mention that a radius shows lamellar-zonal pattern (Fig. 3A), but this is not the femur, tibia or humerus that we usually sampled. In fact, limb bones can show different bone patterns in different parts, the fibulae and ribs usually have less vascular canals that show parallel-fibered or lamellar bone.
5. As the author provided new features of *Scutellosaurus*, a new phylogenetic analysis is necessary. I find that the authors have provided phylogenetic analysis in another paper (JVP online), but they did not mention it here. Furthermore, I strongly suggest that they can do a new phylogenetic analysis of basal ornithischians based on newly published data, such as Boyd 2015 or Han et al 2018, and add some new related taxa. The matrix of Butler et al (2010) has already changed more or less.

I have more comments in a marked version of the pdf. Please also see that. Thank you.

Reviewer: 3

Comments to the Author(s)

This is a well-written and highly detailed anatomical description of the key early dinosaur *Scutellosaurus*, and represents a very valuable addition to the literature that will no doubt be well-cited in the future. In its position at the base of the diverse and successful armoured dinosaur radiation, *Scutellosaurus* is a key taxon and the data contained in this manuscript will be invaluable for future research in taxonomy, cladistics, evolutionary dynamics and palaeobiology.

I have only a very few suggested changes for this manuscript prior to publication:

1. For several elements - the nasal, postorbital, squamosal and others, see annotated PDF - it is acknowledged in the text that better preserved and more complete elements exist for some specimens and indeed the text description is based on these best preserved examples. Yet they are not figured; instead, more poorly preserved specimens are figured. Why not figure the very best examples for each bone on which verbal descriptions are based?
2. I do not think Figure 17 is ever cited in the main text and it does not appear between references to Figures 16 and 18.
3. The authors make the very interesting observation on Page 50, Line 24 that different Early Jurassic localities featured different dinosaur faunas, in terms of composition of herbivores (sauropodomorphs, heterodontosaurids, thyreophorans). Any speculation on what was driving these variations?
4. Some minor errors in the references cited (see PDF), please check these carefully prior to submission.
5. Also a few, very minor errors with figure captions, particularly abbreviations. Again, please check carefully. Depending on the final size of the figure, you may want to consider increasing the size of labels on the figures (abbreviations) as they are currently very small and can be difficult to discern. It would be very useful for Figure 8 to indicate which directions are anterior and posterior.

Lastly, there are some minor grammatical errors and a few places where the meaning of sentences could be clarified - I have made suggested changes in the attached annotated PDF.

Overall, a major piece of work and again a very valuable addition to the literature, superb effort by the authors on this!

===PREPARING YOUR MANUSCRIPT===

===PREPARING YOUR REVISION IN SCHOLARONE===

Author's Response to Decision Letter for (RSOS-201676.R0)

See Appendix C.

RSOS-201676.R1 (Revision)

Review form: Reviewer 2

Is the manuscript scientifically sound in its present form?

No

Are the interpretations and conclusions justified by the results?

No

Is the language acceptable?

Yes

Do you have any ethical concerns with this paper?

No

Have you any concerns about statistical analyses in this paper?

No

Recommendation?

Accept with minor revision (please list in comments)

Comments to the Author(s)

I still have some questions about the bone histology part. The original paper (Padian 2004) only mentioned the bone section of radius to be best characterized as lamellar-zonal. For the bone tissue of femur, he wrote: "Woven bone tissue of periosteal origin surrounds the numerous primary osteons, most of which are parallel and oriented longitudinally (P556), ----No LAGs are observed". This is more like fibrolamellar bone. The original paper did mention that "the bone is parallel-fibered and not highly vascularized compared to the bones of most dinosaurs". However, I am afraid this is also not an "autapomorphy" mentioned in their revised MS. For example, a noosaurids theropod Masiakasaurus also shows parallel-fibered bone type (see Lee and Connor, 2013, Bone histology confirms determinate growth and small body size in the noosaurid theropod Masiakasaurus knopfleri). In sum, I agree that Scutellosaurus may have slower growth rates than most dinosaurs but we need be careful to say it to be autapomorphic in dinosaurs.

Decision letter (RSOS-201676.R1)

Dear Mr Breeden III

On behalf of the Editors, we are pleased to inform you that your Manuscript RSOS-201676.R1 "The anatomy and palaeobiology of the early armoured dinosaur *Scutellosaurus lawleri* from the Kayenta Formation (Lower Jurassic) of Arizona" has been accepted for publication in Royal Society Open Science subject to minor revision in accordance with the referees' reports. Please find the referees' comments along with any feedback from the Editors below my signature.

Please submit your revised manuscript and required files (see below) no later than 7 days from today's (ie 01-Jun-2021) date. Note: the ScholarOne system will 'lock' if submission of the revision is attempted 7 or more days after the deadline. If you do not think you will be able to meet this deadline please contact the editorial office immediately.

Best regards,
Lianne Parkhouse

Editorial Coordinator
 Royal Society Open Science
 openscience@royalsociety.org

on behalf of Professor Kevin Padian (Subject Editor)
 openscience@royalsociety.org

Subject Editor comments to Author:

Dear Ben et al., thanks for your revisions on this. I tend to agree with the reviewer on the histology; we didn't sample a lot, but recall that small taxa grow more slowly than larger related taxa, and our specimen seemed quite typical of small ornithomirans, so I'm not sure this is a synapomorphy of anything, given the mixed tissue signal. Anyway, best wishes for revisions and I look forward to the finished product.

Reviewer comments to Author:

Reviewer: 2
 Comments to the Author(s)

I still have some questions about the bone histology part. The original paper (Padian 2004) only mentioned the bone section of radius to be best characterized as lamellar-zonal. For the bone tissue of femur, he wrote: "Woven bone tissue of periosteal origin surrounds the numerous primary osteons, most of which are parallel and oriented longitudinally (P556), ----No LAGs are observed". This is more like fibrolamellar bone. The original paper did mention that "the bone is parallel-fibered and not highly vascularized compared to the bones of most dinosaurs". However, I am afraid this is also not an "autapomorphy" mentioned in their revised MS. For example, a noasaurids theropod Masiakasaurus also shows parallel-fibered bone type (see Lee and Connor, 2013, Bone histology confirms determinate growth and small body size in the noasaurid theropod Masiakasaurus knopfleri). In sum, I agree that Scutellosaurus may have slower growth rates than most dinosaurs but we need be careful to say it to be autapomorphic in dinosaurs.

===PREPARING YOUR MANUSCRIPT===

Your revised paper should include the changes requested by the referees and Editors of your manuscript. You should provide two versions of this manuscript and both versions must be provided in an editable format:
 one version identifying all the changes that have been made (for instance, in coloured highlight, in bold text, or tracked changes);
 a 'clean' version of the new manuscript that incorporates the changes made, but does not highlight them. This version will be used for typesetting.
 Please ensure that any equations included in the paper are editable text and not embedded images.

While not essential, it will speed up the preparation of your manuscript proof if you format your references/bibliography in Vancouver style (please see

<https://royalsociety.org/journals/authors/author-guidelines/#formatting>). You should include DOIs for as many of the references as possible.

===PREPARING YOUR REVISION IN SCHOLARONE===

<https://royalsociety.org/journals/authors/author-guidelines/#data>. You should ensure that you cite the dataset in your reference list. If you have deposited data etc in the Dryad repository,

please only include the 'For publication' link at this stage. You should remove the 'For review' link.

Author's Response to Decision Letter for (RSOS-201676.R1)

See Appendix D.

Decision letter (RSOS-201676.R2)

Dear Mr Breeden III,

I am pleased to inform you that your manuscript entitled "The anatomy and palaeobiology of the early armoured dinosaur *Scutellostaurus lawleri* from the Kayenta Formation (Lower Jurassic) of Arizona" is now accepted for publication in Royal Society Open Science.

Please see the Royal Society Publishing guidance on how you may share your accepted author manuscript at <https://royalsociety.org/journals/ethics-policies/media-embargo/>. After publication, some additional ways to effectively promote your article can also be found here

<https://royalsociety.org/blog/2020/07/promoting-your-latest-paper-and-tracking-your-results/>.

on behalf of Prof Kevin Padian (Subject Editor)
openscience@royalsociety.org

Appendix A**ROYAL SOCIETY
OPEN SCIENCE****The anatomy and palaeobiology of the early armoured
dinosaur *Scutellosaurus lawleri* from the Kayenta Formation
(Lower Jurassic) of Arizona**

Journal:	Royal Society Open Science
Manuscript ID	RSOS-201676
Article Type:	Research
Date Submitted by the Author:	18-Sep-2020
Complete List of Authors:	Breeden III, Benjamin; The University of Utah, Geology & Geophysics; Natural History Museum of Utah Raven, Thomas; Natural History Museum Butler, Richard; University of Birmingham, School of Geography and Earth Sciences Rowe, Timothy; The University of Texas at Austin, Jackson School of Geosciences Maidment, Susannah; Natural History Museum
Subject:	Palaeontology < EARTH SCIENCES
Keywords:	Dinosauria, Ornithischia, Thyreophora, Kayenta Formation, Jurassic, Scutellosaurus lawleri
Subject Category:	Organismal and Evolutionary Biology

Author-supplied statements

Relevant information will appear here if provided.

Ethics

Does your article include research that required ethical approval or permits?:

Yes

Statement (if applicable):

We describe fossil specimens collected from the lands of the Navajo Nation, which require the written permission from the Navajo Nation Minerals Department prior to publication. We sent a draft of our manuscript to the Navajo Nation Minerals Department for approval to publish on 14 August 2020, and we received approval in writing dated 17 August 2020.

Data

It is a condition of publication that data, code and materials supporting your paper are made publicly available. Does your paper present new data?:

Yes

Statement (if applicable):

Detailed data regarding referred specimens are available as electronic supplementary material.

Conflict of interest

I/We declare we have no competing interests

Statement (if applicable):

CUST_STATE_CONFLICT :No data available.

Authors' contributions

This paper has multiple authors and our individual contributions were as below

Statement (if applicable):

B.T.B., T.J.R., R.J.B., and S.C.R.M. conceived of the study and wrote the manuscript; T.B.R. revised the manuscript. All authors gave final approval for publication and agree to be held accountable for the work performed therein.

[revised manuscript text omitted]

Despite the obvious importance of *Scutellosaurus lawleri*, little further descriptive or systematic
work has been carried out since that of Colbert [18]. Rosenbaum & Padian [23] referred six specimens
from the collections of the University of California Museum of Paleontology (UCMP) to *Scutellosaurus*
*lawleri* but provided only very brief descriptions of most elements. Tykoski [29] figured one additional
specimen repositated at The University of Texas Vertebrate Paleontology Collections (TMM), and Breeden
& Rowe [30] described several additional specimens from TMM. Furthermore, there are inaccuracies in
the anatomical information provided by Colbert [18] and Rosenbaum & Padian [23], the only previous
diagnosis [18] did not attempt to distinguish autapomorphies and symplesiomorphies, and many
phylogenetically informative features have not been previously discussed. The primary aim of this paper
is to expand and emend previous anatomical descriptions, providing new data on non-eurypodan
thyreophoran morphology and new life reconstruction for *Scutellosaurus lawleri*.

41 42 43 **1.1. Institutional abbreviations**

AMNH, American Museum of Natural History, New York, NY, USA; CAMSM, Sedgwick Museum of
Earth Sciences, University of Cambridge, Cambridge, UK; CMN, Canadian Museum of Nature, Ottawa,
Ontario, Canada; MCZ, Museum of Comparative Zoology, Harvard University, Cambridge, MA, USA;
MCZ VPRA, the designated prefix for fossil reptile and amphibian specimens at MCZ; MNA, Museum of
Northern Arizona, Flagstaff, AZ, USA; NHMUK, Natural History Museum (formerly NHM, BMNH),
London, UK; ROM, Royal Ontario Museum, Toronto, Canada; SAM-PK, Iziko South African Museum,

Cape Town, South Africa; SGWG, Greifswalder Geologische Sammlungen, Universität Greifswald
(formerly Ernst-Moritz-Arndt-Universität Greifswald), Greifswald, Germany; TMM, The University of
Texas Vertebrate Paleontology Collections (historically part of the Texas Memorial Museum, formerly
referred to as Vertebrate Paleontology Laboratory), Austin, TX, USA; UCMP, University of California
Museum of Paleontology, Berkeley, CA, USA; UCMP V, designated prefix for UCMP localities; UTCT,
High-Resolution X-ray CT Facility at The University of Texas at Austin; ZDM, Zigong Dinosaur
Museum, Zigong, Sichuan, People's Republic of China.

**2. Geologic setting**

[revised manuscript text omitted]

R1111). A well-developed midline depression is absent, in contrast to many other early ornithischians
(e.g., *Agilisaurus louderbacki*, ZDM 6011; *Heterodontosaurus tucki*, SAM-PK-K1332; *Hexinlusaurus*
*multidens*, ZDM T6001), and the nasals articulate medially along a nearly straight contact. The dorsal
surface of the nasal exhibits cortical remodelling in TMM 43664-1, as in *Scelidosaurus harrisonii*
(NHMUK PV R1111) and stegosaurs (e.g., Stegosauria indet., CMNH 106; *Miragaia longicollum*, ML
433), and the ventral surface is relatively smooth. There are no anteroposteriorly-extending ridges on the
dorsal surface of the nasal, unlike the condition in *Paranthodon africanus* (NHMUK PV OR47338; [4])
and *Hesperosaurus mjosi* (DMNH 29431; [95]).

21 22 **5.1.4. Frontal (Fig. 4A–B)**

Although a posterior portion of a left frontal is preserved in MNA.V.175 [30], more complete frontals are
preserved in UCMP 130580 (Fig. 4A–B), TMM 43663-1, and TMM 43664-1 [30]. The frontals are much
longer anteroposteriorly than wide mediolaterally, a feature we interpret to be an autapomorphy of
*Scutellosaurus*. The frontals widen towards their posterior ends and are dorsoventrally thickened where
they form a midline contact with one another. They are not extensively sculpted, but there is some weak
sculpting of the dorsal surface immediately adjacent to the parietal contact and supratemporal fossa at the
posterior end of the element. In *Emausaurus ernsti* (SGWG 85) the frontals are similarly unsculpted,
except for a small region around the anterior orbital margin. The frontals are gently arched along their
length. As in *Lesothosaurus diagnosticus* [28: fig. 12B], there is an elongate facet on the anterolateral
surface of the frontal for a tapering posterior process of the prefrontal (Fig. 4A, a.pf). As noted by
Maidment & Porro [96], there are a series of fine striations along the lateral margin of the frontal of
UCMP 130580 (Fig. 4A, str), which were interpreted to represent an osteological correlate for soft tissue,
and these striations are also present on the frontals of TMM 43663-1 and TMM 43664-1 [30] and on the
frontals of *Emausaurus ernsti* (SGWG 85). The frontal formed part of the dorsal rim of the orbit (Fig.
4A–B, orb), as in most early ornithischians [28,63,76,90] but differing from the condition in
*Scelidosaurus harrisonii* (NHMUK PV R1111), ankylosaurs [97], and stegosaurs [92], in which

supraorbital bones were incorporated into the skull roof and separated the frontals from the dorsal rim of
the orbit. Medial to the orbital margin on the ventral surface of the frontal, there is an anteroposteriorly
extending wide groove for the olfactory tract of the brain (Fig. 4B, olft). The posterior portion of the
frontal possesses a prominent supratemporal fossa to accommodate the temporal musculature (Fig. 4A,
stfo), as in saurischians [19] and some early ornithischians [28,76,98].

**5.1.5. Parietal (Fig. 4A–B)**

The parietal of *Scutellosaurus lawleri* is known only from a fragment in UCMP 130580 (Fig. 4A–B).
Anteriorly, the parietal formed a nearly straight, interdigitating, transverse suture with the frontals. On
each side of the parietal there is a sharp ridge (Fig. 4A, ri) that demarcates the supratemporal fossa and is
continuous with the ridge defining the supratemporal fossa (Fig. 4A, stfo) on the frontal. These ridges on
the parietal converge posteriorly, and together define a flat, triangular area on the dorsal surface of the
bone.

**5.1.6. Lacrimal**

The lacrimal of *Scutellosaurus lawleri* is known only from TMM 43664-1, which preserves both right
and left lacrimals [30]. The lacrimal comprises a subquadrangular main body with a curved and tapering
posteroventral process, similar in morphology to that of *Lesothosaurus diagnosticus* [28] and *Emausaurus*
*ernsti* (SGWG 85). The lateral surface of the body is cortically remodelled, in contrast to that of
*Emausaurus ernsti* (SGWG 85), where it is smooth. The antorbital fossa extends onto the anteroventral
portion of the main body of the lacrimal of *Scutellosaurus lawleri*, and the posteroventral process of the
lacrimal forms the posterodorsal margin of the antorbital fenestra. It is unclear whether the lacrimal
contacted the prefrontal.

**5.1.7. Jugal (Fig. 4C–D)**

A small portion of a left jugal is present in MNA.V.175, and a mostly complete right jugal is present in

UCMP 130580 (Fig. 4C–D), but the most complete jugals are known from TMM 43663-1 and TMM
43664-1 [30]. Sutural surfaces are here identified based upon comparison with the disarticulated holotype
skull of *Scelidosaurus harrisonii* (NHMUK PV R1111). The lateral surface of the jugal ventral to the
dorsal process and ventral to the ventral margin of the orbit is ornamented with several weak
anterodorsal-to-posteroventrally extending, anastomosing ridges and shallow grooves (Fig. 4C, orn). This
ornamentation is similar in position and form to that of *Emausaurus ernsti* (SGWG 85) and *Scelidosaurus*
*harrisonii* [11], although it is less well developed than in the latter.

The anterior process of the jugal is mediolaterally expanded beneath the orbit (Fig. 4C). This
mediolateral expansion does not occur along the entire anterior process but is confined to a short section
immediately anterior to the dorsal process of the jugal. A medially and slightly ventrally facing facet for
the ectopterygoid is present on the medial surface of this expansion (Fig. 4D, a.ect), like the condition in
*Emausaurus ernsti* (SGWG 85) and *Scelidosaurus harrisonii* (NHMUK PV R1111). Two sutural surfaces
are present on the jugal for the maxilla (Fig. 4C–D, a.max): an extensive ventrally and slightly medially
facing surface, and a shorter and less well defined ventrally and laterally facing surface. The sutural
surface for the lacrimal, positioned on the dorsal surface of the anterior end of the process in
*Scelidosaurus harrisonii* (NHMUK PV R1111), may be represented in *Scutellosaurus lawleri* by a small
dorsal notch in the extreme end of the anterior process of the jugal of TMM 43664-1. The dorsal process
of the jugal is anteroposteriorly narrow and has an elongate narrow groove on the anterior portion of the
lateral surface to articulate with the ventral process of the postorbital (Fig. 4C, a.po), as in *Emausaurus*
*ernsti* (SGWG 85). The posterior process of the jugal has a V-shaped, anteriorly tapering depression on its
medial surface, which articulates with the quadratojugal (Fig. 4D, a.qj); a facet with similar morphology
is present in *Scelidosaurus* (BRSMG Ce12785).

5.1.8 Postorbital (Fig. 4E–F)

54 TMM 43663-1 and TMM 43664-1 possess the best preserved postorbitals [30], but a small portion of a
55
56
57

left postorbital is present in UCMP 130580 (Fig. 4E–F). The postorbital is a triradiate element with
anteromedial, posterior, and ventral processes, which articulate with the frontal, squamosal, and jugal,
respectively. The anterior margin of the postorbital between the anteromedial and ventral processes is
rounded and forms the posterodorsal border of the orbit (Fig. 4E, orb). The anterior process is the shortest
of the three processes and bifurcates anteriorly [30]. The posterior and ventral processes are both slender
[30], similar to those of *Lesothosaurus diagnosticus* [28,99], heterodontosaurids [63,76], and *Emausaurus*
*ernsti* (SGWG 85), but in contrast to that of *Scelidosaurus harrisonii* (NHMUK PV R1111) in which the
main body of the postorbital and the posterior and ventral processes are more robust. The posterior and
ventral processes taper towards their tips, as in other early thyreophorans (*Emausaurus ernstii*, SGWG
85; *Scelidosaurus harrisonii*, NHMUK PV R1111).

23 24 25 26 **5.1.9 Quadratojugal**

The quadratojugal is only known from TMM 43664-1 [30]. The quadratojugal is mediolaterally
compressed and overlies the lateral ramus of the quadrate. There is a prominent foramen on the medial
surface of the quadratojugal that does not extend through to the lateral surface [30]. *Scelidosaurus*
*harrisonii* (NHMUK PV R1111) possesses a dorsal process of the quadratojugal, but this part of the bone
is not preserved in TMM 43664-1 so its presence or absence in *Scutellosaurus lawleri* cannot be
determined.

35 36 37 38 39 40 41 42 43 **5.1.10 Squamosal**

Although a possible portion of the squamosal in MNA.V.175 was noted by Colbert [18], we could not
locate this; however, a complete squamosal is preserved in TMM 43663-1 [30]. This is a triradiate
element comprising anterior, prequadratic, and postquadratic processes. The lateral surface of the
squamosal is deeply excavated by a smooth concavity. The anterior process bifurcates into dorsal and
ventral projections that flank the posterior process of the postorbital to form the upper temporal bar, and
the prequadratic process is anteroposteriorly thin and poorly preserved. The postquadratic process is

anteroposteriorly broader than the prequadratic process and is the shortest of the three processes.

8 **5.1.11 Quadrate (Fig 4G–K)**

The quadrate is not known from the holotype or paratype specimens but is preserved in a number of other
specimens (e.g., UCMP 130580 [Fig. 4J–K], UCMP 175166: [23]; TMM 43647-7, TMM 43687-121,
TMM 43663-1, TMM 43664-1:[30]; MCZ VPRA-8795 [Fig. 4G–I]; MCZ VPRA-8797). The head is
arched posteriorly relative to the shaft (Fig. 4H) and is not fused to the paroccipital process. It is
moderately compressed in lateral view, like that of *Stegosaurus* (e.g., NHMUK PV R36730). The
pterygoid wing (Fig. 4J–K, ptw) is best preserved in TMM 43664-1, and is subtriangular, dorsoventrally
tall, tapers anteromedially, and bears a shallow fossa [30: fig. 10K–L], like that of *Stegosaurus* (NHMUK
PV R36730). The lateral surface of the ventral end of the shaft bears an extensive articular surface for the
quadratojugal (Fig. 4J, a.qj), which ventrally would have closely approached the quadrate condyles. As
noted by Rosenbaum & Padian [23], the medial condyle is weakly enlarged relative to the lateral and
resembles that of *Scelidosaurus harrisonii* (NHMUK PV R1111), and the lateral condyle tapers laterally
where it curves anteriorly. A facet is present on the posteroventral surface of the condyles in UCMP
130580 (Fig. 4J, fc), but this is not present in any other specimen that preserves a quadrate (e.g., TMM
43663-1, TMM 43687-121). Such a facet does not appear to be present in *Scelidosaurus harrisonii*
(NHMUK PV R1111) whereas the quadrate is not preserved in *Emausaurus ernsti* (SGWG 85).

**5.1.12 Braincase (Fig. 5)**

Only the basioccipital and the paroccipital process are known from the braincase of *Scutellosaurus*
*lawleri*, and all examples of these elements are preserved disarticulated. None are well enough preserved
to determine the exits of cranial nerves.

The basioccipital is preserved in MNA.V.175 (Fig. 5A–B), UCMP 130580 (Fig. 5C–D), and
MCZ VPRA-8797 (Fig. 5E–H). A midline ridge is present on the ventral surface of the basioccipital

separating the basioccipital recesses (Fig. 5F, bor); a similar ridge is also present in *Lesothosaurus*
*diagnosticus* [99], *Emausaurus ernsti* (SGWG 85) and *Scelidosaurus harrisonii* (NHMUK PV R1111).
Anterior to the basioccipital recesses are rugose basal tubera (Fig. 5F–G, bt). The occipital condyle (Fig.
5F, H, occ) is convex and reniform in posterior view, with a concave dorsal surface demarcating the floor
of the foramen magnum (Fig. 5H, fm). On the dorsal surface and lateral to the margin of the foramen
magnum are roughened posterolaterally-oriented surfaces for articulation with the exoccipitals (Fig. 5E,
H, a.ex). The endocranial floor (Fig. 5E, ef) on the dorsal surface of the basioccipital is generally smooth
and concave except for a thin median ridge at the anterior end.

Paroccipital processes are preserved in TMM 43663-1 [30] and MCZ VPRA-8797 (Fig. 5I–J).
The paraoccipital process flares laterally from the main body of the exoccipital-opisthotic. In posterior
view, there are notches along the dorsal edge of the main body of the exoccipital-opisthotic presumably
for articulation with the supraoccipital (Fig. 5I, a.so) and parietal (Fig. 5I, a.p). In anterior view, there is a
rough subtriangular surface that is pierced by a small foramen, and there is a medial notch for articulation
with the prootic (Fig. 5J, a.pr). Dorsal to this surface, there is a groove for articulation with the
supraoccipital (Fig. 5J, a.so). The distal end of the paroccipital process is dorsoventrally expanded but is
not pendant.

**5.1.13 Dentary (Figs 6, 7A–C)**

A nearly complete left (Fig. 6A–B) and partial right (Fig. 6C–D) dentary are preserved in the holotype,
and several partial dentaries are preserved in other specimens (e.g., MCZ VPRA-8797: Fig. 7A–C). The
dentary tooth row is slightly sinuous in dorsal view (Fig. 7C), but to a much lesser degree than that of
derived nodosaurs (e.g., *Panoplosaurus mirus*, ROM 1215: [100]), and in lateral and medial views, the
tooth row is sinuous with the anterior end downturned (Figs. 6–7), as in other thyreophorans (e.g.,
*Scelidosaurus harrisonii*, NHMUK PV R1111; *Emausaurus ernsti*, SGWG 85; [21]), but differing from
early ornithischians [28,63,76]. There is a ridge on the dorsal half of the lateral surface that laterally

defines a narrow buccal emargination posterior to the tenth tooth position (Figs. 6A, 7A, be). The ventral
margin of the dentary is relatively straight, in contrast to the sinuous ventral margin in nodosaurs [26].
Colbert [18: 11] identified an unusual rugose depression on the ventral surface of the left dentary of
MNA.V.175 as a possible pathologic feature (Fig. 6A); a similar depression is also present in an
equivalent position of the right dentary, but is not present in the referred dentary of MCZ VPRA-8797
(Fig. 7A–C).

There is a broad, flat facet on the medial surface of the dentary for the symphysis (Fig. 6B, sym),
although this is not ‘spout-shaped’ as in most other ornithischians [21,28], indicating that the articulation
between the lower jaws would have been V-shaped in dorsal view. The tooth row appears to extend to the
anterior end of the dentary, and there is no clear prementary facet on the dorsal surface. A small facet
ventrally on the anterior dentary may be for a prementary, suggesting that if it existed at all, the prementary
was very small. The dentary of *Emausaurus ernsti* (SGWG 85) is very similar, also possessing little
evidence for a prementary. The maximum dorsoventral depth of the dentary ramus is at the posterior end
and the depth of the dentary at the symphysis is shallower than half the maximum depth of the dentary in
lateral view. The Meckelian groove on the medial surface (Figs. 6–7, Mg) does not extend anteriorly as
far as the symphyseal facet, in contrast to the condition in *Emausaurus ernsti* (SGWG 85). Small

[revised manuscript text omitted]

centrum in MCZ VPRA-8801 (Fig. 9J–L). The neural spine (Fig. 9G) is prominent and extends
posterodorsally beyond the posterior margins of the postzygapophyses, broadening slightly posteriorly, as
in *Scelidosaurus harrisonii* (NHMUK R1111; [11]: fig. 7). A midline ridge extends along the length of
the dorsal surface of the neural spine. Only the right postzygapophysis (Fig. 9G, I, poz) is preserved, and
it faces mostly ventrally and slightly laterally. The axis centrum of MCZ VPRA-8801 preserves portions
of the base of the neural arch (Fig. 9, na) that show a circular neural canal (Fig. 9, nc), and the atlas

[revised manuscript text omitted]

**68. Owen R. 1842.** Report on British fossil reptiles. *Report of the British Association for the*
*Advancement of Science* 11: 60–204.

**69. Seeley HG. 1887.** On the classification of the fossil animals commonly named Dinosauria.
*Proceedings of the Royal Society of London* 43: 165–171.
**70. Nopcsa F. 1915.** Die Dinosaurier der Siebenbürgischen Landesteile Ungarns. *Mitteilungen aus*
*dem Jahrbuche der Königlich Ungarischen Geologischen Anstalt* 21: 1–26.
**71. Sereno PC. 1998.** A rationale for phylogenetic definitions, with application to the higher-level
taxonomy of Dinosauria. *Neues Jahrbuch für Geologie und Paläontologie Abhandlungen* 210:
41–83.
**72. Marsh AD. 2018.** Anatomy, systematics, and geochronology of *Dilophosaurus wetherilli*. In:
Thomson TJ, Harris J, Milner AR, Kirkland J, eds. Western Association of Vertebrate
Paleontologists Annual Meeting. *PaleoBios*, 35: 10–11.
**73. Bell CJ, Head JJ, Mead JI. 2004.** Synopsis of the herpetofauna from Porcupine Cave. pp. 117–
126 in Barnosky, A. D. (ed.), *Biodiversity Response to Climate Change in the Middle*
*Pleistocene: The Porcupine Cave Fauna from Colorado*. University of California Press,
Berkeley.
**74. Nesbitt SJ, Stocker MR. 2008.** The vertebrate assemblage of the Late Triassic Canjilon Quarry
(northern New Mexico, USA), and the importance of apomorphy-based assemblage
Comparisons. *Journal of Vertebrate Paleontology* 28: 1063–1072.
**75. Butler RJ, Galton PM, Porro LB, Chiappe LM, Henderson DM, Erickson GM. 2010.** Lower
limits of ornithischian dinosaur body size inferred from a new Upper Jurassic heterodontosaurid
from North America. *Proceedings of the Royal Society of London B* 277: 375–381.
**76. Norman DB, Crompton AW, Butler RJ, Porro LB, Charig AJ. 2011.** The Lower Jurassic
ornithischian dinosaur *Heterodontosaurus tucki* Crompton & Charig, 1962: cranial anatomy,
functional morphology, taxonomy, and relationships. *Zoological Journal of the Linnean Society*
163: 182–276.
**77. Kirkland JI, Alcalá L, Loewen MA, Espílez E, Mampel L, Wiersma JP. 2013.** The basal
nodosaurid ankylosaur *Europelta carbonensis* n. gen., n. sp. from the Lower Cretaceous (lower

Albian) Escucha Formation of northeastern Spain. *PLoS ONE* 8(12).

<https://doi.org/10.1371/journal.pone.0080405>

**78. Carpenter K, DiCroce T, Kinneer B, Simon R. 2013.** Pelvis of *Gargoyleosaurus* (Dinosauria:
Ankylosauria) and the origin and evolution of the ankylosaur pelvis. *PLoS ONE* 8(11).

<https://doi.org/10.1371/journal.pone.0079887>

**79. Maidment SCR, Norman DB, Wei G. 2006.** Re-description of the postcranial skeleton of the
middle Jurassic stegosaur *Huayangosaurus taibaii*. *Journal of Vertebrate Paleontology* 26(4):

944–956. [https://doi.org/10.1671/0272-4634\(2006\)26\[944:ROTPSO\]2.0.CO;2](https://doi.org/10.1671/0272-4634(2006)26[944:ROTPSO]2.0.CO;2)

**80. Maidment SCR, Barrett PM. 2011.** The locomotor musculature of basal ornithischian
dinosaurs. *Journal of Vertebrate Paleontology* 31: 1265–1291.

**81. Nesbitt SJ. 2011.** The Early Evolution of Archosaurs: Relationships and the Origin of Major
Clades. *Bulletin of the American Museum of Natural History* 352: 1–292.

<https://doi.org/10.1206/352.1>

[revised manuscript text omitted]

Journal:	Royal Society Open Science
Manuscript ID	RSOS-201676
Article Type:	Research
Date Submitted by the Author:	18-Sep-2020
Complete List of Authors:	Breeden III, Benjamin; The University of Utah, Geology & Geophysics; Natural History Museum of Utah Raven, Thomas; Natural History Museum Butler, Richard; University of Birmingham, School of Geography and Earth Sciences Rowe, Timothy; The University of Texas at Austin, Jackson School of Geosciences Maidment, Susannah; Natural History Museum
Subject:	Palaeontology < EARTH SCIENCES
Keywords:	Dinosauria, Ornithischia, Thyreophora, Kayenta Formation, Jurassic, Scutellosaurus lawleri
Subject Category:	Organismal and Evolutionary Biology

Author-supplied statements

Relevant information will appear here if provided.

Ethics

Does your article include research that required ethical approval or permits?:

Yes

Statement (if applicable):

We describe fossil specimens collected from the lands of the Navajo Nation, which require the written permission from the Navajo Nation Minerals Department prior to publication. We sent a draft of our manuscript to the Navajo Nation Minerals Department for approval to publish on 14 August 2020, and we received approval in writing dated 17 August 2020.

Data

It is a condition of publication that data, code and materials supporting your paper are made publicly available. Does your paper present new data?:

Yes

Statement (if applicable):

Detailed data regarding referred specimens are available as electronic supplementary material.

Conflict of interest

I/We declare we have no competing interests

Statement (if applicable):

CUST_STATE_CONFLICT :No data available.

Authors' contributions

This paper has multiple authors and our individual contributions were as below

Statement (if applicable):

B.T.B., T.J.R., R.J.B., and S.C.R.M. conceived of the study and wrote the manuscript; T.B.R. revised the manuscript. All authors gave final approval for publication and agree to be held accountable for the work performed therein.

[revised manuscript text omitted]

Despite the obvious importance of *Scutellosaurus lawleri*, little further descriptive or systematic
work has been carried out since that of Colbert [18]. Rosenbaum & Padian [23] referred six specimens
from the collections of the University of California Museum of Paleontology (UCMP) to *Scutellosaurus*
*lawleri* but provided only very brief descriptions of most elements. Tykoski [29] figured one additional
specimen repositated at The University of Texas Vertebrate Paleontology Collections (TMM), and Breeden
& Rowe [30] described several additional specimens from TMM. Furthermore, there are inaccuracies in
the anatomical information provided by Colbert [18] and Rosenbaum & Padian [23], the only previous
diagnosis [18] did not attempt to distinguish autapomorphies and symplesiomorphies, and many
phylogenetically informative features have not been previously discussed. The primary aim of this paper
is to expand and emend previous anatomical descriptions, providing new data on non-eurypodan
thyreophoran morphology and new life reconstruction for *Scutellosaurus lawleri*.

41 42 43 **1.1. Institutional abbreviations**

AMNH, American Museum of Natural History, New York, NY, USA; CAMSM, Sedgwick Museum of
Earth Sciences, University of Cambridge, Cambridge, UK; CMN, Canadian Museum of Nature, Ottawa,
Ontario, Canada; MCZ, Museum of Comparative Zoology, Harvard University, Cambridge, MA, USA;
MCZ VPRA, the designated prefix for fossil reptile and amphibian specimens at MCZ; MNA, Museum of
Northern Arizona, Flagstaff, AZ, USA; NHMUK, Natural History Museum (formerly NHM, BMNH),
London, UK; ROM, Royal Ontario Museum, Toronto, Canada; SAM-PK, Iziko South African Museum,

Cape Town, South Africa; SGWG, Greifswalder Geologische Sammlungen, Universität Greifswald
(formerly Ernst-Moritz-Arndt-Universität Greifswald), Greifswald, Germany; TMM, The University of
Texas Vertebrate Paleontology Collections (historically part of the Texas Memorial Museum, formerly
referred to as Vertebrate Paleontology Laboratory), Austin, TX, USA; UCMP, University of California
Museum of Paleontology, Berkeley, CA, USA; UCMP V, designated prefix for UCMP localities; UTCT,
High-Resolution X-ray CT Facility at The University of Texas at Austin; ZDM, Zigong Dinosaur
Museum, Zigong, Sichuan, People's Republic of China.

**2. Geologic setting**

[revised manuscript text omitted]

R1111). A well-developed midline depression is absent, in contrast to many other early ornithischians
(e.g., *Agilisaurus louderbacki*, ZDM 6011; *Heterodontosaurus tucki*, SAM-PK-K1332; *Hexinlusaurus*
*multidens*, ZDM T6001), and the nasals articulate medially along a nearly straight contact. The dorsal
surface of the nasal exhibits cortical remodelling in TMM 43664-1, as in *Scelidosaurus harrisonii*
(NHMUK PV R1111) and stegosaurs (e.g., Stegosauria indet., CMNH 106; *Miragaia longicollum*, ML
433), and the ventral surface is relatively smooth. There are no anteroposteriorly-extending ridges on the
dorsal surface of the nasal, unlike the condition in *Paranthodon africanus* (NHMUK PV OR47338; [4])
and *Hesperosaurus mjosi* (DMNH 29431; [95]).

21 22 **5.1.4. Frontal (Fig. 4A–B)**

Although a posterior portion of a left frontal is preserved in MNA.V.175 [30], more complete frontals are
preserved in UCMP 130580 (Fig. 4A–B), TMM 43663-1, and TMM 43664-1 [30]. The frontals are much
longer anteroposteriorly than wide mediolaterally, a feature we interpret to be an autapomorphy of
*Scutellosaurus*. The frontals widen towards their posterior ends and are dorsoventrally thickened where
they form a midline contact with one another. They are not extensively sculpted, but there is some weak
sculpting of the dorsal surface immediately adjacent to the parietal contact and supratemporal fossa at the
posterior end of the element. In *Emausaurus ernsti* (SGWG 85) the frontals are similarly unsculpted,
except for a small region around the anterior orbital margin. The frontals are gently arched along their
length. As in *Lesothosaurus diagnosticus* [28: fig. 12B], there is an elongate facet on the anterolateral
surface of the frontal for a tapering posterior process of the prefrontal (Fig. 4A, a.pf). As noted by
Maidment & Porro [96], there are a series of fine striations along the lateral margin of the frontal of
UCMP 130580 (Fig. 4A, str), which were interpreted to represent an osteological correlate for soft tissue,
and these striations are also present on the frontals of TMM 43663-1 and TMM 43664-1 [30] and on the
frontals of *Emausaurus ernsti* (SGWG 85). The frontal formed part of the dorsal rim of the orbit (Fig.
4A–B, orb), as in most early ornithischians [28,63,76,90] but differing from the condition in
*Scelidosaurus harrisonii* (NHMUK PV R1111), ankylosaurs [97], and stegosaurs [92], in which

supraorbital bones were incorporated into the skull roof and separated the frontals from the dorsal rim of
the orbit. Medial to the orbital margin on the ventral surface of the frontal, there is an anteroposteriorly
extending wide groove for the olfactory tract of the brain (Fig. 4B, olft). The posterior portion of the
frontal possesses a prominent supratemporal fossa to accommodate the temporal musculature (Fig. 4A,
stfo), as in saurischians [19] and some early ornithischians [28,76,98].

14 15 16 **5.1.5. Parietal (Fig. 4A–B)**

The parietal of *Scutellosaurus lawleri* is known only from a fragment in UCMP 130580 (Fig. 4A–B).
Anteriorly, the parietal formed a nearly straight, interdigitating, transverse suture with the frontals. On
each side of the parietal there is a sharp ridge (Fig. 4A, ri) that demarcates the supratemporal fossa and is
continuous with the ridge defining the supratemporal fossa (Fig. 4A, stfo) on the frontal. These ridges on
the parietal converge posteriorly, and together define a flat, triangular area on the dorsal surface of the
bone.

33 **5.1.6. Lacrimal**

The lacrimal of *Scutellosaurus lawleri* is known only from TMM 43664-1, which preserves both right
and left lacrimals [30]. The lacrimal comprises a subquadrangular main body with a curved and tapering
posteroventral process, similar in morphology to that of *Lesothosaurus diagnosticus* [28] and *Emausaurus*
*ernsti* (SGWG 85). The lateral surface of the body is cortically remodelled, in contrast to that of
*Emausaurus ernsti* (SGWG 85), where it is smooth. The antorbital fossa extends onto the anteroventral
portion of the main body of the lacrimal of *Scutellosaurus lawleri*, and the posteroventral process of the
lacrimal forms the posterodorsal margin of the antorbital fenestra. It is unclear whether the lacrimal
contacted the prefrontal.

53 54 **5.1.7. Jugal (Fig. 4C–D)**

55 A small portion of a left jugal is present in MNA.V.175, and a mostly complete right jugal is present in
56
57

UCMP 130580 (Fig. 4C–D), but the most complete jugals are known from TMM 43663-1 and TMM
43664-1 [30]. Sutural surfaces are here identified based upon comparison with the disarticulated holotype
skull of *Scelidosaurus harrisonii* (NHMUK PV R1111). The lateral surface of the jugal ventral to the
dorsal process and ventral to the ventral margin of the orbit is ornamented with several weak
anterodorsal-to-posteroventrally extending, anastomosing ridges and shallow grooves (Fig. 4C, orn). This
ornamentation is similar in position and form to that of *Emausaurus ernsti* (SGWG 85) and *Scelidosaurus*
*harrisonii* [11], although it is less well developed than in the latter.

The anterior process of the jugal is mediolaterally expanded beneath the orbit (Fig. 4C). This
mediolateral expansion does not occur along the entire anterior process but is confined to a short section
immediately anterior to the dorsal process of the jugal. A medially and slightly ventrally facing facet for
the ectopterygoid is present on the medial surface of this expansion (Fig. 4D, a.ect), like the condition in
*Emausaurus ernsti* (SGWG 85) and *Scelidosaurus harrisonii* (NHMUK PV R1111). Two sutural surfaces
are present on the jugal for the maxilla (Fig. 4C–D, a.max): an extensive ventrally and slightly medially
facing surface, and a shorter and less well defined ventrally and laterally facing surface. The sutural
surface for the lacrimal, positioned on the dorsal surface of the anterior end of the process in
*Scelidosaurus harrisonii* (NHMUK PV R1111), may be represented in *Scutellosaurus lawleri* by a small
dorsal notch in the extreme end of the anterior process of the jugal of TMM 43664-1. The dorsal process
of the jugal is anteroposteriorly narrow and has an elongate narrow groove on the anterior portion of the
lateral surface to articulate with the ventral process of the postorbital (Fig. 4C, a.po), as in *Emausaurus*
*ernsti* (SGWG 85). The posterior process of the jugal has a V-shaped, anteriorly tapering depression on its
medial surface, which articulates with the quadratojugal (Fig. 4D, a.qj); a facet with similar morphology
is present in *Scelidosaurus* (BRSMG Ce12785).

5.1.8 Postorbital (Fig. 4E–F)

54 TMM 43663-1 and TMM 43664-1 possess the best preserved postorbitals [30], but a small portion of a

left postorbital is present in UCMP 130580 (Fig. 4E–F). The postorbital is a triradiate element with
anteromedial, posterior, and ventral processes, which articulate with the frontal, squamosal, and jugal,
respectively. The anterior margin of the postorbital between the anteromedial and ventral processes is
rounded and forms the posterodorsal border of the orbit (Fig. 4E, orb). The anterior process is the shortest
of the three processes and bifurcates anteriorly [30]. The posterior and ventral processes are both slender
[30], similar to those of *Lesothosaurus diagnosticus* [28,99], heterodontosaurids [63,76], and *Emausaurus*
*ernsti* (SGWG 85), but in contrast to that of *Scelidosaurus harrisonii* (NHMUK PV R1111) in which the
main body of the postorbital and the posterior and ventral processes are more robust. The posterior and
ventral processes taper towards their tips, as in other early thyreophorans (*Emausaurus ernstii*, SGWG
85; *Scelidosaurus harrisonii*, NHMUK PV R1111).

23 24 25 26 **5.1.9 Quadratojugal**

The quadratojugal is only known from TMM 43664-1 [30]. The quadratojugal is mediolaterally
compressed and overlies the lateral ramus of the quadrate. There is a prominent foramen on the medial
surface of the quadratojugal that does not extend through to the lateral surface [30]. *Scelidosaurus*
*harrisonii* (NHMUK PV R1111) possesses a dorsal process of the quadratojugal, but this part of the bone
is not preserved in TMM 43664-1 so its presence or absence in *Scutellosaurus lawleri* cannot be
determined.

35 36 37 38 39 40 41 42 43 **5.1.10 Squamosal**

Although a possible portion of the squamosal in MNA.V.175 was noted by Colbert [18], we could not
locate this; however, a complete squamosal is preserved in TMM 43663-1 [30]. This is a triradiate
element comprising anterior, prequadratic, and postquadratic processes. The lateral surface of the
squamosal is deeply excavated by a smooth concavity. The anterior process bifurcates into dorsal and
ventral projections that flank the posterior process of the postorbital to form the upper temporal bar, and
the prequadratic process is anteroposteriorly thin and poorly preserved. The postquadratic process is

anteroposteriorly broader than the prequadratic process and is the shortest of the three processes.

8 **5.1.11 Quadrate (Fig 4G–K)**

The quadrate is not known from the holotype or paratype specimens but is preserved in a number of other
specimens (e.g., UCMP 130580 [Fig. 4J–K], UCMP 175166: [23]; TMM 43647-7, TMM 43687-121,
TMM 43663-1, TMM 43664-1:[30]; MCZ VPRA-8795 [Fig. 4G–I]; MCZ VPRA-8797). The head is
arched posteriorly relative to the shaft (Fig. 4H) and is not fused to the paroccipital process. It is
moderately compressed in lateral view, like that of *Stegosaurus* (e.g., NHMUK PV R36730). The
pterygoid wing (Fig. 4J–K, ptw) is best preserved in TMM 43664-1, and is subtriangular, dorsoventrally
tall, tapers anteromedially, and bears a shallow fossa [30: fig. 10K–L], like that of *Stegosaurus* (NHMUK
PV R36730). The lateral surface of the ventral end of the shaft bears an extensive articular surface for the
quadratojugal (Fig. 4J, a.qj), which ventrally would have closely approached the quadrate condyles. As
noted by Rosenbaum & Padian [23], the medial condyle is weakly enlarged relative to the lateral and
resembles that of *Scelidosaurus harrisonii* (NHMUK PV R1111), and the lateral condyle tapers laterally
where it curves anteriorly. A facet is present on the posteroventral surface of the condyles in UCMP
130580 (Fig. 4J, fc), but this is not present in any other specimen that preserves a quadrate (e.g., TMM
43663-1, TMM 43687-121). Such a facet does not appear to be present in *Scelidosaurus harrisonii*
(NHMUK PV R1111) whereas the quadrate is not preserved in *Emausaurus ernsti* (SGWG 85).

44 **5.1.12 Braincase (Fig. 5)**

Only the basioccipital and the paroccipital process are known from the braincase of *Scutellosaurus*
*lawleri*, and all examples of these elements are preserved disarticulated. None are well enough preserved
to determine the exits of cranial nerves.

The basioccipital is preserved in MNA.V.175 (Fig. 5A–B), UCMP 130580 (Fig. 5C–D), and
MCZ VPRA-8797 (Fig. 5E–H). A midline ridge is present on the ventral surface of the basioccipital

separating the basioccipital recesses (Fig. 5F, bor); a similar ridge is also present in *Lesothosaurus*
*diagnosticus* [99], *Emausaurus ernsti* (SGWG 85) and *Scelidosaurus harrisonii* (NHMUK PV R1111).
Anterior to the basioccipital recesses are rugose basal tubera (Fig. 5F–G, bt). The occipital condyle (Fig.
5F, H, occ) is convex and reniform in posterior view, with a concave dorsal surface demarcating the floor
of the foramen magnum (Fig. 5H, fm). On the dorsal surface and lateral to the margin of the foramen
magnum are roughened posterolaterally-oriented surfaces for articulation with the exoccipitals (Fig. 5E,
H, a.ex). The endocranial floor (Fig. 5E, ef) on the dorsal surface of the basioccipital is generally smooth
and concave except for a thin median ridge at the anterior end.

Paroccipital processes are preserved in TMM 43663-1 [30] and MCZ VPRA-8797 (Fig. 5I–J).
The paraoccipital process flares laterally from the main body of the exoccipital-opisthotic. In posterior
view, there are notches along the dorsal edge of the main body of the exoccipital-opisthotic presumably
for articulation with the supraoccipital (Fig. 5I, a.so) and parietal (Fig. 5I, a.p). In anterior view, there is a
rough subtriangular surface that is pierced by a small foramen, and there is a medial notch for articulation
with the prootic (Fig. 5J, a.pr). Dorsal to this surface, there is a groove for articulation with the
supraoccipital (Fig. 5J, a.so). The distal end of the paroccipital process is dorsoventrally expanded but is
not pendant.

**5.1.13 Dentary (Figs 6, 7A–C)**

A nearly complete left (Fig. 6A–B) and partial right (Fig. 6C–D) dentary are preserved in the holotype,
and several partial dentaries are preserved in other specimens (e.g., MCZ VPRA-8797: Fig. 7A–C). The
dentary tooth row is slightly sinuous in dorsal view (Fig. 7C), but to a much lesser degree than that of
derived nodosaurs (e.g., *Panoplosaurus mirus*, ROM 1215: [100]), and in lateral and medial views, the
tooth row is sinuous with the anterior end downturned (Figs. 6–7); as in other thyreophorans (e.g.,
*Scelidosaurus harrisonii*, NHMUK PV R1111; *Emausaurus ernsti*, SGWG 85; [21]), but differing from
early ornithischians [28,63,76]. There is a ridge on the dorsal half of the lateral surface that laterally

defines a narrow buccal emargination posterior to the tenth tooth position (Figs. 6A, 7A, be). The ventral
margin of the dentary is relatively straight, in contrast to the sinuous ventral margin in nodosaurs [26].
Colbert [18: 11] identified an unusual rugose depression on the ventral surface of the left dentary of
MNA.V.175 as a possible pathologic feature (Fig. 6A); a similar depression is also present in an
equivalent position of the right dentary, but is not present in the referred dentary of MCZ VPRA-8797
(Fig. 7A–C).

There is a broad, flat facet on the medial surface of the dentary for the symphysis (Fig. 6B, sym),
although this is not ‘spout-shaped’ as in most other ornithischians [21,28], indicating that the articulation
between the lower jaws would have been V-shaped in dorsal view. The tooth row appears to extend to the
anterior end of the dentary, and there is no clear prementary facet on the dorsal surface. A small facet
ventrally on the anterior dentary may be for a prementary, suggesting that if it existed at all, the prementary
was very small. The dentary of *Emausaurus ernsti* (SGWG 85) is very similar, also possessing little
evidence for a prementary.  The maximum dorsoventral depth of the dentary ramus is at the posterior end
and the depth of the dentary at the symphysis is shallower than half the maximum depth of the dentary in
lateral view. The Meckelian groove on the medial surface (Figs. 6–7, Mg) does not extend anteriorly as
far as the symphyseal facet, in contrast to the condition in *Emausaurus ernsti* (SGWG 85). Small

[revised manuscript text omitted]

centrum in MCZ VPRA-8801 (Fig. 9J–L). The neural spine (Fig. 9G) is prominent and extends
posterodorsally beyond the posterior margins of the postzygapophyses, broadening slightly posteriorly, as
in *Scelidosaurus harrisonii* (NHMUK R1111; [11]: fig. 7). A midline ridge extends along the length of
the dorsal surface of the neural spine. Only the right postzygapophysis (Fig. 9G, I, poz) is preserved, and
it faces mostly ventrally and slightly laterally. The axis centrum of MCZ VPRA-8801 preserves portions
of the base of the neural arch (Fig. 9, na) that show a circular neural canal (Fig. 9, nc), and the atlas

[revised manuscript text omitted]

68. **Owen R. 1842.** Report on British fossil reptiles. *Report of the British Association for the Advancement of Science* 11: 60–204.

**69. Seeley HG. 1887.** On the classification of the fossil animals commonly named Dinosauria.
*Proceedings of the Royal Society of London* 43: 165–171.
**70. Nopcsa F. 1915.** Die Dinosaurier der Siebenbürgischen Landesteile Ungarns. *Mitteilungen aus*
*dem Jahrbuche der Königlich Ungarischen Geologischen Anstalt* 21: 1–26.
**71. Sereno PC. 1998.** A rationale for phylogenetic definitions, with application to the higher-level
taxonomy of Dinosauria. *Neues Jahrbuch für Geologie und Paläontologie Abhandlungen* 210:
41–83.
**72. Marsh AD. 2018.** Anatomy, systematics, and geochronology of *Dilophosaurus wetherilli*. In:
Thomson TJ, Harris J, Milner AR, Kirkland J, eds. Western Association of Vertebrate
Paleontologists Annual Meeting. *PaleoBios*, 35: 10–11.
**73. Bell CJ, Head JJ, Mead JI. 2004.** Synopsis of the herpetofauna from Porcupine Cave. pp. 117–
126 in Barnosky, A. D. (ed.), *Biodiversity Response to Climate Change in the Middle*
*Pleistocene: The Porcupine Cave Fauna from Colorado*. University of California Press,
Berkeley.
**74. Nesbitt SJ, Stocker MR. 2008.** The vertebrate assemblage of the Late Triassic Canjilon Quarry
(northern New Mexico, USA), and the importance of apomorphy-based assemblage
Comparisons. *Journal of Vertebrate Paleontology* 28: 1063–1072.
**75. Butler RJ, Galton PM, Porro LB, Chiappe LM, Henderson DM, Erickson GM. 2010.** Lower
limits of ornithischian dinosaur body size inferred from a new Upper Jurassic heterodontosaurid
from North America. *Proceedings of the Royal Society of London B* 277: 375–381.
**76. Norman DB, Crompton AW, Butler RJ, Porro LB, Charig AJ. 2011.** The Lower Jurassic
ornithischian dinosaur *Heterodontosaurus tucki* Crompton & Charig, 1962: cranial anatomy,
functional morphology, taxonomy, and relationships. *Zoological Journal of the Linnean Society*
163: 182–276.
**77. Kirkland JI, Alcalá L, Loewen MA, Espílez E, Mampel L, Wiersma JP. 2013.** The basal
nodosaurid ankylosaur *Europelta carbonensis* n. gen., n. sp. from the Lower Cretaceous (lower

Albian) Escucha Formation of northeastern Spain. *PLoS ONE* 8(12).

<https://doi.org/10.1371/journal.pone.0080405>

**78. Carpenter K, DiCroce T, Kinneer B, Simon R. 2013.** Pelvis of *Gargoyleosaurus* (Dinosauria:
Ankylosauria) and the origin and evolution of the ankylosaur pelvis. *PLoS ONE* 8(11).

<https://doi.org/10.1371/journal.pone.0079887>

**79. Maidment SCR, Norman DB, Wei G. 2006.** Re-description of the postcranial skeleton of the
middle Jurassic stegosaur *Huayangosaurus taibaii*. *Journal of Vertebrate Paleontology* 26(4):

944–956. [https://doi.org/10.1671/0272-4634\(2006\)26\[944:ROTPSO\]2.0.CO;2](https://doi.org/10.1671/0272-4634(2006)26[944:ROTPSO]2.0.CO;2)

**80. Maidment SCR, Barrett PM. 2011.** The locomotor musculature of basal ornithischian
dinosaurs. *Journal of Vertebrate Paleontology* 31: 1265–1291.

**81. Nesbitt SJ. 2011.** The Early Evolution of Archosaurs: Relationships and the Origin of Major
Clades. *Bulletin of the American Museum of Natural History* 352: 1–292.

<https://doi.org/10.1206/352.1>

[revised manuscript text omitted]

Appendix C

Responses to Reviewer Comments

Subject Editor comments to Author:

The reviewers are happy to see this paper, and are generally enthusiastic about its publication. One raises what should be carefully considered so as to dispel any doubts about "shingling": it would be important to show how this paper differs from the JVP one, and to repeat only necessary details (although this is sometimes difficult). A new phylogeny is probably needed, but this should not be difficult. A wider inclusion of taxa is also recommended. The other reviewer asks why you are figuring sub-optimal specimens rather than better ones, and if this because the better specimens were already figured in the JVP paper please make that clear.

Redundancies with Breeden & Rowe (2020) have been removed, and we have clarified in the manuscript that the specimens from TMM described in that paper are not re-described here. See comments below in response to Reviewer 2 regarding the exclusion of a phylogeny.

Finally, some clarity about the histological features may be needed. It is common in virtually all non-avian dinosaurs that fibro-lamellar bone should be expressed at least in early growth and a transition made either to FL tissue that is less vascularized OR to lamellar-zonal bone before skeletal growth effectively ceases and the EFS is deposited. Smaller taxa tend to grow more slowly, but of course any given section represents its ontogenetic age, so perhaps you could clarify? I don't mean at all to argue with your position, only to note that the outgroups to Dinosauria already grew very quickly and featured copious FL bone, yet they were not larger than *Scutellosaurus*. So the hypothesis that the larger, later Thyreophora evolved (not "developed," please) FL bone as a strategy for more rapid growth to larger size could be called into question. I will be interested in your perspective on this.

Padian et al. (2004) noted that in contrast to most dinosaurs, *Scutellosaurus* possesses lamellar-zonal bone throughout the cortex of the bones they examined. We might expect lamellar-zonal bone to be deposited during final stages of growth (during deposition of the EFS), but dinosaurs and their outgroups generally possess faster-growing, better vascularized fibrolamellar bone in earlier-deposited parts of the cortex. The point we are making here is that *Scutellosaurus* does *not* possess this earlier signature of rapid growth, unlike many dinosaurs. Padian et al. (2004) concluded that *Scutellosaurus* had slow growth throughout its life as a consequence of its small body size, but subsequent work on taxa including the small-bodied basal ornithischian *Lesothosaurus* has indicated that *Lesothosaurus* had typically dinosaurian fibrolamellar bone during early growth. This suggests that *Scutellosaurus* is unique in having slow growth throughout its life, and indeed this slow growth may be a synapomorphy of Thyreophora, because other studies we quote in this section of our paper have also suggested that stegosaurs and ankylosaurs have histological signatures indicative of slow growth relative to other dinosaurs. What we are trying to say here is that the *basal condition* for Thyreophora was slow growth (lamellar-zonal bone), but as thyreophorans got bigger, they grew faster than their smaller, earlier-diverging relatives. They still grew slower than other dinosaurs (they have more poorly vascularized bone), but *relative to basal thyreophorans*, they grew faster because they were bigger. We've attempted to nuance the text to explain this better. If the subject editor feels

we have either mischaracterized previous studies or haven't done a sufficient job of expressing what we mean here, we are happy to make further changes if advised.

Also, I noted that on line 23 of your MS page 46, you have an extra "closure"

Corrected.

Reviewer: 2

This study provided a detailed description of the early armoured dinosaur *Scutellosaurus lawleri* from the Kayenta Formation of Arizona. This will help for improving the early evolution of ornithischians. Both the description and figures are good. But there are still some things that need to make it clear before publication. Here are my comments as follows:

1. I notice that the authors have already provided a detailed description of *Scutellosaurus* published in JVP recently, although the latter only contains specimens from TMM. This MS cited many times of that paper, and I think it is a little redundant. So I believe it is more appropriate that this paper is a complementary description for *Scutellosaurus* that no need to describe all the bones, but just contains new data and comparisons. This will make the paper more concise.

Redundancies with Breeden & Rowe (2020) have been removed.

2. The authors mainly compared *Scutellosaurus* with basal thyreophorans. I hope the author adds more comparison with other ornithischians that have similar size or feature, such as *Eocursor*, *Agilisaurus*, *Hexinlusaurus*, *Jeholosaurus*, and also some primitive stegosaurs including *Huayangosaurus*. *Isaberrysaura* was original considered to a basal neornithischian, but re-put basal stegosaur in Han et al (2018). It should also be added for comparison. In addition, I notice that the dentary of *Scutellosaurus* is quite long and shallow dorsoventrally. The ratio of length to depth is also worth comparing with other ornithischians.

We have added comparisons to *Jeholosaurus*, *Huayangosaurus*, and *Isaberrysaura*. Comparisons to *Eocursor*, *Agilisaurus*, and *Hexinlusaurus* were already present in the manuscript, but we have added additional references where suggested in Reviewer 2's marked PDF.

3. In the comparison, I notice many other specimens do not have references. Do you check this by yourself or from references? If you check by yourself, please mention that in the "material and method" part.

We have added a paragraph clarifying which comparative taxa were examined firsthand and which were referenced from the literature.

4. Page 48. The authors state that among ornithischians only *Scutellosaurus* have lamellar-zonal bone. This seems unlikely. In Padian (2004), we can see that *Scutellosaurus* still has abundant vascular canals in the inner region of a tibia (Padian 2004, Figure 3D), unlike the condition in

modern crocodiles and lizards, but is similar to *Jeholosaurus* (Han et al 2020, bone histology of *Jeholosaurus*). Padian (2004) did mention that a radius shows lamellar-zonal pattern (Fig. 3A), but this is not the femur, tibia or humerus that we usually sampled. In fact, limb bones can show different bone patterns in different parts, the fibulae and ribs usually have less vascular canals that show parallel-fibered or lamellar bone.

This comment of the reviewer is difficult to reconcile with the information provided in the respective papers. Padian et al. (2004: 559), on the histology of three limb bones of *Scutellosaurus*, stated:

“In summary, there is very little if any typical fibro-lamellar bone in most of these elements; instead, the bone is parallel-fibered and not highly vascularized compared to the bones of most dinosaurs... Its “lamellar-zonal” structure is similar to that of crocodylians.”

They also noted that bone of the basal ornithopod *Orodromeus* is “more highly vascularized than *Scutellosaurus*...”.

Han et al. (2020), on the histology of the tibia and fibula of the basal ornithopod *Jeholosaurus*, stated that bone tissue in all growth stages they investigated was fibro-lamellar (Table 3, p. 6), and that the bone tissue throughout was highly vascularized.

Han et al. (2020) did not refer to the paper of Padian et al. (2004) or contrast the condition seen in *Jeholosaurus* with that in *Scutellosaurus*. Padian et al. (2004) showed an image of the tibia of *Scutellosaurus*, which can be compared directly with the tibia of *Jeholosaurus*. The difference in presentation of these images makes comparison a bit difficult, but it does appear that the *Jeholosaurus* tibia is more highly vascularized than that of *Scutellosaurus*, which accords with Padian et al’s findings that *Orodromeus* is also more highly vascularized than *Scutellosaurus*.

It would certainly be interesting to update the study of Padian et al. (2004) in the light of new specimens of *Scutellosaurus* and the vast number of histological studies that have been carried out on basal ornithopods and other small dinosaurs in the 17 years since the former was published. However, that is vastly beyond the scope of the present work. Given the information that is available in the literature, we stand by our original hypothesis that the evidence currently suggests that *Scutellosaurus* had slower growth rates than other dinosaurs. This isn’t particularly controversial as slow growth rates have been hypothesized for other thyreophorans. We have attempted to nuance the text to express these things better.

5. As the author provided new features of *Scutellosaurus*, a new phylogenetic analysis is necessary. I find that the authors have provided phylogenetic analysis in another paper (JVP online), but they did not mention it here. Furthermore, I strongly suggest that they can do a new phylogenetic analysis of basal ornithischians based on newly published data, such as Boyd 2015 or Han et al 2018, and add some new related taxa. The matrix of Butler et al (2010) has already changed more or less.

We argue that a phylogeny is beyond the scope of this paper and is not necessary because the phylogenetic position of *Scutellosaurus* as a thyreophoran is not controversial. In every matrix to which it has been added since the earliest phylogenetic analyses of ornithischian dinosaurs (e.g., Sereno, 1986), it has been recovered as an early-diverging thyreophoran. This includes analyses designed to focus on basal ornithischian relationships (e.g., Butler et al., 2008 and all matrices derived from that), basal dinosaur relationships (e.g., Baron et al., 2017), basal ornithopod relationships (e.g., Boyd et al., 2015; Han et al., 2018), ankylosaurs (e.g., Thompson et al., 2012), and stegosaurs (e.g., Maidment et al., 2008 and matrices derived from that). These analyses have used a variety of different character sets but result in the same phylogenetic position for *Scutellosaurus* as a non-eurypodan thyreophoran. Further, new observations on the anatomy of *Scutellosaurus* resulting from the work submitted here have already been incorporated into phylogenies by authors of this manuscript (e.g., Butler et al. 2008; Maidment et al. 2008; Raven & Maidment, 2017; Breeden & Rowe, 2020), some of which have then been incorporated into later ornithischian analyses. Lastly, the ornithischian data matrices Reviewer 2 suggests using to test the phylogenetic position of *Scutellosaurus* are not appropriate because they are not explicitly designed to test thyreophoran relationships, and its placement within Ornithischia as a thyreophoran is not in question. The most appropriate dataset would be a total-group thyreophoran dataset, which one of the authors (Tom Raven) currently has in preparation. That is a substantial piece of work on its own that will be published as a separate paper.

I have more comments in a marked version of the pdf. Please also see that. Thank you.

We have addressed the notes and comments in the marked PDF, perhaps the most significant of which is the addition of a figure showing the holotype manus (Figure 17 of the revised manuscript).

Reviewer: 3

Comments to the Author(s)

This is a well-written and highly detailed anatomical description of the key early dinosaur *Scutellosaurus*, and represents a very valuable addition to the literature that will no doubt be well-cited in the future. In its position at the base of the diverse and successful armoured dinosaur radiation, *Scutellosaurus* is a key taxon and the data contained in this manuscript will be invaluable for future research in taxonomy, cladistics, evolutionary dynamics and palaeobiology.

I have only a very few suggested changes for this manuscript prior to publication:

1. For several elements - the nasal, postorbital, squamosal and others, see annotated PDF - it is acknowledged in the text that better preserved and more complete elements exist for some specimens and indeed the text description is based on these best preserved examples. Yet they are not figured; instead, more poorly preserved specimens are figured. Why not figure the very best examples for each bone on which verbal descriptions are based?

As noted in our response to Review 2, redundancies with Breeden & Rowe (2020) have been removed, and we have also more explicitly stated that skeletal elements only present in specimens from TMM are fully described in that paper and are therefore not re-described here.

2. I do not think Figure 17 is ever cited in the main text and it does not appear between references to Figures 16 and 18.

We have added citations to Figure 17 (which is now Figure 18 with the addition of a new figure of the manus).

3. The authors make the very interesting observation on Page 50, Line 24 that different Early Jurassic localities featured different dinosaur faunas, in terms of composition of herbivores (sauropodomorphs, heterodontosaurids, thyreophorans). Any speculation on what was driving these variations?

We agree that this question is interesting, but it is beyond the scope of this study and warrants further study on its own. We refrain from speculating further here without additional data.

4. Some minor errors in the references cited (see PDF), please check these carefully prior to submission.

Corrected.

5. Also a few, very minor errors with figure captions, particularly abbreviations. Again, please check carefully. Depending on the final size of the figure, you may want to consider increasing the size of labels on the figures (abbreviations) as they are currently very small and can be difficult to discern. It would be very useful for Figure 8 to indicate which directions are anterior and posterior.

We have addressed the errors in figure captions and revised Figure 8 to include arrows; however, we have opted against increasing the font size of the labels.

Lastly, there are some minor grammatical errors and a few places where the meaning of sentences could be clarified - I have made suggested changes in the attached annotated PDF.

Corrected.

Appendix D

Responses to Editor and Reviewer Comments

Subject Editor comments to Author:

Dear Ben et al., thanks for your revisions on this. I tend to agree with the reviewer on the histology; we didn't sample a lot, but recall that small taxa grow more slowly than larger related taxa, and our specimen seemed quite typical of small ornithomirans, so I'm not sure this is a synapomorphy of anything, given the mixed tissue signal. Anyway, best wishes for revisions and I look forward to the finished product.

Reviewer comments to Author:

Reviewer: 2

Comments to the Author(s)

I still have some questions about the bone histology part. The original paper (Padian 2004) only mentioned the bone section of radius to be best characterized as lamellar-zonal. For the bone tissue of femur, he wrote: "Woven bone tissue of periosteal origin surrounds the numerous primary osteons, most of which are parallel and oriented longitudinally (P556), ----No LAGs are observed". This is more like fibrolamellar bone. The original paper did mention that "the bone is parallel-fibered and not highly vascularized compared to the bones of most dinosaurs". However, I am afraid this is also not an "autapomorphy" mentioned in their revised MS. For example, a noasaurid theropod *Masiakasaurus* also shows parallel-fibered bone type (see Lee and Connor, 2013, Bone histology confirms determinate growth and small body size in the noasaurid theropod *Masiakasaurus knopfleri*). In sum, I agree that *Scutellosaurus* may have slower growth rates than most dinosaurs but we need be careful to say it to be autapomorphic in dinosaurs.

Although we maintain that the apparent slow growth of *Scutellosaurus* seems unique among its close relatives, we have removed language referring to this phenomenon as apomorphic per the feedback from the Subject Editor and Reviewer 2.